# Actin dysregulation induces neuroendocrine plasticity and immune evasion: a vulnerability of small cell lung cancer

Yoojeong Seo [1,10], Shengzhe Zhang [1,10], Jinho Jang[1,10], Kyung-Pil Ko[1,10], Kee-Beom Kim[2,3,10], Yuanjian Huang[1,10], Dong-Wook Kim[2], Bongjun Kim [1], Gengyi Zou[1], Jie Zhang[1], Sohee Jun[1], Wonhong Chu[1], Nicole A. Kirk[2], Ye Eun Hwang[2], Young Ho Ban[4], Shilpa S. Dhar [5], Joseph M. Chan [6], Min Gyu Lee[7], Charles M. Rudin [6], Kwon-Sik Park [2] ✉ & Jae-Il Park [1,8,9] ✉

Small cell lung cancer (SCLC) is an aggressive malignancy with limited therapeutic options. Capping protein inhibiting regulator of actin dynamics (*CRACD*) that promotes actin polymerization, is frequently inactivated in SCLC. However, the role of CRACD loss in SCLC is unknown. Here we show that *CRACD* depletion drives neuroendocrine (NE) cell plasticity and immune evasion in SCLC. Mechanistically, CRACD inactivation disrupts actin organization, leading to suppression of Yap1-NOTCH signaling and subsequent NE gene upregulation. Simultaneously, CRACD loss drives EZH2-mediated histone methylation via nuclear actin disruption, leading to repression of MHC-I genes and depletion of CD8+ T cells. Consequently, CRACD-downregulated tumors exhibit increased cellular heterogeneity and escape from immune surveillance. Conversely, pharmacological inhibition of EZH2 restores MHC-I expression, reactivates antitumor immunity, and suppresses tumor growth. These findings identify CRACD as a tumor suppressor that constrains cell plasticity and immune evasion, highlighting the CRACD–EZH2–MHC-I axis as a potential therapeutic vulnerability in SCLC.

SCLC accounts for 13% of all lung cancers and remains a particularly lethal disease, with a 5-year survival rate of 7%. It is estimated to cause ~30,000 patient deaths annually in the United States[1,2]. Major contributing factors to the high mortality rate of SCLC patients include the high prevalence of metastasis at the time of diagnosis, which limits therapeutic options, and nearly universal disease relapse associated with resistance to further therapies[3,4].

Notably, immune checkpoint blockade (ICB) approaches designed to target tumors expressing neoantigens are effective in only ~13% of patients with SCLC - a small subset, given that the high mutation burden of SCLC tumors should be sufficient to trigger a

[1]Department of Experimental Radiation Oncology, Division of Radiation Oncology, The University of Texas MD Anderson Cancer Center, Houston, TX, USA. [2]Department of Microbiology, Immunology, and Cancer Biology, University of Virginia School of Medicine, Charlottesville, VA, USA. [3]BK21 FOUR KNU Creative BioResearch Group, School of Life Sciences, Kyungpook National University, Daegu, Republic of Korea. [4]Hamatovascular Biology Center, Robert M. Berne Cardiovascular Research Center, University of Virginia School of Medicine, Charlottesville, VA, USA. [5]Department of Gastrointestinal Medical Oncology, The University of Texas MD Anderson Cancer Center, Houston, TX, USA. [6]Department of Medicine, Thoracic Oncology Service, Memorial Sloan Kettering Cancer Center, New York, NY, USA. [7]Department of Molecular and Cellular Oncology, The University of Texas MD Anderson Cancer Center, Houston, TX, USA. [8]Graduate School of Biomedical Sciences, The University of Texas MD Anderson Cancer Center, Houston, TX, USA. [9]Program in Genetics and Epigenetics, The University of Texas MD Anderson Cancer Center, Houston, TX, USA. [10]These authors contributed equally: Yoojeong Seo, Shengzhe Zhang, Jinho Jang, Kyung-Pil Ko, Kee-Beom Kim, Yuanjian Huang. ✉e-mail: kp5an@virginia.edu; jaeil@mdanderson.org

robust immune response from cytotoxic T lymphocytes[5-7]. While it remains unclear what underlies the refractoriness of SCLC to ICB and how to stratify patient tumors by the degree of response to ICB, recent studies have explored emerging molecular subtypes of SCLC tumors, classified based on the actions of key lineage transcription factors (ASCL1, NEUROD1, and POU2F3) and inflammation[8-11]. However, the current classification system has not been robust enough to reliably predict immunotherapy response. Therefore, unveiling how SCLC cells evade immune surveillance and become resistant to immunotherapy is imperative to improve the durability of ICB in responding patients and to inform strategies to increase the fraction of patients benefiting from ICB.

Cell plasticity is defined as a change in cell fate, identity, or phenotype[12]. Tumor cell plasticity is implicated in tumor cell heterogeneity, therapy resistance, and metastasis[12-15]. NE cell plasticity has been observed in several cancers, including pancreatic, prostate, and lung cancers. Nonetheless, the underlying mechanisms of NE plasticity and tumor heterogeneity of SCLC remain elusive.

NOTCH signaling inactivation has been recognized as a key driver of NE differentiation in SCLC, whereas its activation promotes transitions toward non-NE states, contributing to tumor heterogeneity[16]. NOTCH1 suppression is often mediated by DLL3, a direct ASCL1 target and an endogenous NOTCH inhibitor[17]. In parallel, MHC-I downregulation has emerged as a central mechanism of immune evasion in SCLC, often characterized by widespread epigenetic silencing of antigen presentation pathways[18]. LSD1 has been shown to suppress MHC-I expression, and its inhibition restores antigen presentation and anti-tumor immunity in SCLC models[19].

We recently discovered a tumor suppressor gene called *CRACD*[20]. CRACD is ubiquitously expressed in epithelial cells and binds to and inhibits capping proteins (CAPZA and CAPZB), negative regulators of actin polymerization[20]. CRACD promotes actin polymerization, which is crucial for maintaining the cadherin-catenin-actin complex of epithelial cells. *CRACD* is recurrently mutated or transcriptionally downregulated in colorectal cancer cells, which results in a reduction of filamentous actin (F-actin) and disruption of the cadherin-catenin-actin complex[20]. These alterations by CRACD inactivation cause loss of epithelial cell integrity and decrease the cytoplasm-to-nucleus volume ratio; cells become 'small'. A pathological consequence of these aberrant changes is evident in the intestines, where CRACD inactivation hyperactivates WNT signaling via β-catenin release from the cadherin-catenin-actin complex and accelerates intestinal tumorigenesis[20].

*CRACD* is frequently inactivated in SCLC[21], which led us to hypothesize that CRACD is a tumor suppressor of SCLC. Here, we show that loss of CRACD promotes actin polymerization, drives neuroendocrine plasticity and immune evasion in SCLC. CRACD inactivation leads to MHC-I silencing and T cell exclusion through EZH2-mediated epigenetic repression, defining a distinct molecular signature associated with SCLC immune escape.

## Results

### CRACD loss converts preneoplastic *Rb1*, *Trp53* KO cells into SCLC-like cells

*CRACD* is mutated in 11-16% of SCLC patient tumors and cell lines, ranking after *RB1* and *TP53* but more frequently than *RBL2*, *CREBBP*, and *EP300* among validated tumor suppressor genes (Supplementary Fig. S1a-d)[21-24]. Additionally, *CRACD* mRNA expression is downregulated in SCLC tumors compared to normal lung tissues (Supplementary Fig. S1e). Therefore, we hypothesized that CRACD loss-of-function (LOF) contributes to SCLC tumorigenesis. To test this, we determined whether *Cracd* knockout (KO) is sufficient to promote the transformation of preneoplastic precursor cells of SCLC (preSCs). The preSCs were derived from early-stage NE lesions developed in an *Rb1* and *Trp53* double KO (dKO) mouse model of SCLC. Upon an oncogenic

event, such as *L-Myc* amplification or *Crebbp/Ep300* loss, preSCs progress to an invasive and fully malignant tumor[22,25,26]. Using CRISPR/Cas9-mediated gene editing as previously performed[20], we targeted the exon 2 of *Cracd* in preSCs. *Cracd* KO preSCs readily transformed into aggregates and spheres, characteristic of SCLC cells in culture, and formed subcutaneous tumors in an allograft model significantly faster than *Cracd* wild-type (WT) preSCs (Supplementary Fig. S2a-e).

Since *Cracd* KO induces SCLC-like morphological changes in preSC cells (Supplementary Fig. S2b), we investigated whether CRACD depletion is sufficient to drive cell plasticity by single-cell RNA sequencing (scRNA-seq) of preSC allograft tumors derived from preSC cells (*Cracd* WT or KO) (Supplementary Fig. S2f, S2g-j. Compared to *Cracd* WT, *Cracd* KO preSC tumors exhibited marked differences in the cell cluster proportion (Supplementary Fig. S2k) with upregulation of NE markers (*ChgA*, *Neurod1*, *Syp*, and *Uchl1*) and Mki67, a cell proliferation marker (Supplementary Fig. S2l). Cell lineage trajectory analysis using RNA velocity (scVelo)[27] and Dynamo[28] indicates that the root cell clusters, i.e., cellular origins (cell clusters 2 and 6), were increased in *Cracd* KO preSC tumors compared to *Cracd* WT (Supplementary Fig. S2m, n, Supplementary Movie 1). preSC allograft tumors comprised highly proliferative ('High prolif') and relatively less proliferative ('Low prolif') cells. Compared to *Cracd* WT preSC tumors, *Cracd* KO tumors showed increased cell numbers in root cell clusters in both less (cluster 2) and highly (cluster 6) proliferative cells and decreased cell numbers in differentiation cell clusters (Supplementary Fig. S2o), indicating the cell plasticity in CRPR2 tumors. These results suggest that CRACD depletion is sufficient to drive the cell plasticity of preneoplastic SCLC cells into SCLC-like cells.

### CRACD depletion accelerates SCLC tumorigenesis in vivo

Using GEMMs, we determined the impact of CRACD LOF on SCLC tumorigenesis. We employed a GEMM in which *Rb1*<sup>fl/fl</sup>, *Trp53*<sup>fl/fl</sup>, and *Rbl2*<sup>fl/fl</sup> alleles (RPR2) were conditionally deleted on the background of *Cracd* WT alleles or germline *Cracd* KO (*Cracd*, *Rb1*, *Trp53*, and *Rbl2* quadruple KO [CRPR2])[29,30]. CRPR2 mice showed marked increases in tumor burden and number (Fig. 1a-d) and mitotic index of SCLC tumors compared to those of RPR2 mice (Fig. 1e), indicating that *Cracd* KO accelerates SCLC tumor development in vivo. These results suggest that CRACD plays a tumor-suppressive role in SCLC tumorigenesis.

### *Cracd* loss promotes SCLC cell plasticity

To investigate the mechanisms by which CRACD loss accelerates SCLC tumorigenesis, we performed scRNA-seq of SCLC tumors isolated from the lung tissues of RPR2 and CRPR2 mice (Fig. 1f). The two datasets (RPR2 and CRPR2) were integrated and annotated for each cell type (Supplementary Fig. S3a-c). Epithelial tumor cell clusters were selected by unsupervised sub-clustering (Supplementary Fig. S3c-h, and Supplementary Data 3). Cell clusters 2, 3-13, and 15 were present in both RPR2 and CRPR2 tumors, while clusters 1 and 14 were unique to CRPR2. Compared to RPR2, CRPR2 tumors exhibited increased cell numbers in clusters 4, 6, and 7, whereas cluster 8 was reduced (Fig. 1g, and Supplementary Fig. S3i). Both RPR2 and CRPR2 tumors consisted of NE (*Ascl1* and *Calca* positive) and non-NE (*Ascl1* and *Calca* negative) tumor cells (Fig. 1h). Clusters 6-10, 12, 13, and 15 (NE cells) displayed higher expression of NE markers than clusters 1, 3-5 (non-NE cells) (Fig. 1h). In CRPR2 tumors, NE genes (*Ascl1* and *Calca*) were upregulated compared to RPR2, mirroring the NE gene upregulation in *Cracd* KO lung adenocarcinoma (LUAD)[31].

We conducted a comparative analysis of signaling pathways associated with SCLC tumorigenesis: NOTCH (*Hes1, Dll1, Jag1, Notch1/2/3*), MYC (*Myc, Mycl, Ndrg1*), WNT (*Ccnd1, Axin2, Wnt4, Wnt5a, Wnt7*), and EMT (*Zeb1/2*). NOTCH signaling was more active in non-NE cells of both RPR2 and CRPR2 tumors, while CRPR2's non-NE cells displayed marked activation of YAP1 and NOTCH signaling. The MYC pathway

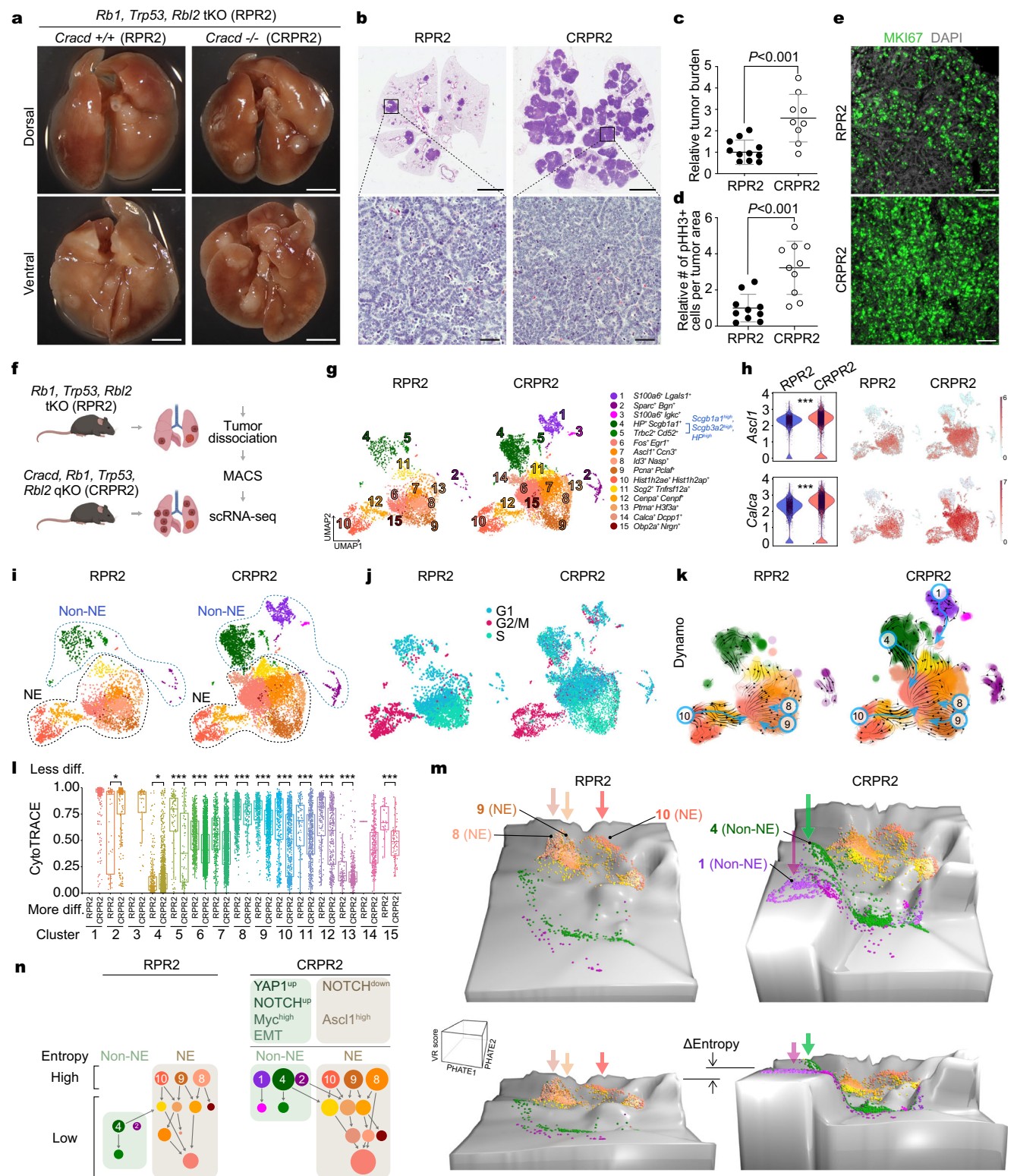

was also activated in non-NE cells of CRPR2 tumors. WNT signaling showed higher scores in NE cells compared to non-NE cells, while non-canonical WNT ligands (Wnt5a and Wnt7) were upregulated in non-NE cells of CRPR2 tumors compared to RPR2. Similarly, EMT genes (*Zeb1* and *Zeb2*) were upregulated in the non-NE cells of CRPR2 tumors compared to RPR2 (Fig. 1i, and Supplementary Fig. S4). We also examined cell proliferation in each cell cluster, finding that NE clusters in both RPR2 and CRPR2 tumors were highly proliferative (S or G2/M phases), while non-NE cells in RPR2 were less proliferative (G1 phase).

However, non-NE cells in CRPR2 displayed hyperproliferation (Fig. 1j), consistent with the accelerated proliferation of *Cracd* KO preSC cells (Supplementary Fig. S2a–c).

Given that *Cracd* KO induces preneoplastic cell plasticity (Supplementary Fig. S1) and accelerates SCLC tumorigenesis (Supplementary Fig. S2), we assessed its impact on tumor cell plasticity by analyzing cell lineage trajectories. While scVelo did not reveal significant differences between RPR2 and CRPR2 tumors (Supplementary Fig. S3j), the Dynamo algorithm that predicts cell fate transitions based

**Fig. 1 | Transformation of preneoplastic SCLC cells into SCLC-like cells by CRACD depletion. a–d** Analysis of autochthonous mouse models: RPR2 (*Rb1*, *Trp53*, *Rbl2* triple KO [tKO]) vs. CRPR2 (*Cracd*, *Rb1*, *Trp53*, *Rbl2* quadruple KO [qKO]). Representative images of whole lungs (RPR2 vs. CRPR2) (**a**) and hematoxylin and eosin–stained lung sections (**b**). Tumor burden (**c**) and proliferative cell quantification based on pHH3+ cells per tumor area (**d**) in RPR2 and CRPR2 mice. Each dot represents an independent biological sample (*n* = 8 mice per group). Scale bars: 5 mm (A and B [upper]), 40 μm (B [lower]). *P* values were calculated using two-sided Student's *t*-test; error bars: SD. **e** Immunostaining of MKI67 in RPR2 and CRPR2 tumors. Scale bars: 40 μm. Representative images (*n* ≥ 3). **f** Workflow for scRNA-seq of RPR2 and CRPR2 tumors six months after Ad-CMV-Cre infection. **g** UMAPs of cell types within RPR2 and CRPR2 tumor cell subsets. **h** Violin (left) and feature (right) plots visualizing *Ascl1* and *Calca* expression between RPR2 and

CRPR2 datasets. *P* values (***: <0.001) were calculated using two-sided Student's *t*-test. **i** UMAP plots of tumor cells from RPR2 and CRPR2 tumors, colored by cluster identity. **j** UMAPs visualizing cell cycle status. **k** UMAPs for predicted cell fates and the most probable path of cell-state transitions, analyzed by using the Dynamo package. **l** Boxplots of the cell differentiation potential of each cell cluster based on the CytoTRACE score analysis; diff.: cell differentiation. Boxes show the median (center line) and interquartile range (25th–75th percentiles), whiskers extend to 1.5 × IQR, and points represent individual cells. *P* values (*P* < 0.05; **P < 0.001) were calculated using two-sided Student's *t*-tests. **m** Waddington's landscape-like visualization of cell plastic potential. PHATE maps were 3D rendered based on VR scores. Arrows indicate cellular origins with higher cell plastic potential. **n** Illustration of cell lineages of RPR2 and CRPR2 tumors. Representative images are shown (n ≥ 3). Source data are provided as a Source Data file.

on differential geometry suggests that CRPR2 tumors displayed more complex cell lineage patterns than RPR2 tumors. In both tumors, NE clusters 8-10 were root cells in both RPR2 and CRPR2 tumors, but CRPR2 also identified non-NE clusters 1 and 4 as new root cells (Fig. 1k, Supplementary Movie 1). Partition-based graph abstraction further confirmed that CRACD loss increased cell lineage diversity (Supplementary Fig. S3k). We also determined the effect of *Cracd* KO on cell differentiation using CytoTRACE, which infers relative cell state (differentiation vs. de-differentiation)[32]. CRPR2 tumors exhibited higher overall cell differentiation than RPR2 (Fig. 1l). Cell clusters 1, 3, and 14 could not be compared due to their absence in RPR2. Root cell clusters in CRPR2 showed high CytoTRACE scores, i.e., lower cell differentiation states (Fig. 1l).

Next, we assessed the cell plastic potential (CPP) based on single-cell entropy[33]. Using this, we generated Waddington's landscape-like illustration by calculating valley-ridge (VR) scores, combining single-cell entropy with cell lineage trajectories on PHATE maps[34] (Fig. 1m, and Supplementary Fig. S3l). In RPR2 tumors, cell clusters 8-10 (NE cells) were located at the apexes and gave rise to differentiated cell clusters (Fig. 1m, left panels), as identified by Dynamo analysis as root cell clusters (Fig. 1k). However, in CRPR2 tumors, in addition to clusters 8-10, newly emerged clusters 1 and 4 (non-NE cells) were positioned at the apexes and acted as root cells (Fig. 1m, right panels). It was also observed that the cell clusters at the apexes in CRPR2 tumors displayed higher CPP than those in RPR2 tumors (Fig. 1m, n, 'ΔEntropy'). These findings suggest that CRACD LOF increases cell plastic potential and promotes cell plasticity.

### *Cracd* KO increases tumor cell heterogeneity with NOTCH signaling downregulation

Cell plasticity contributes to tumor cell heterogeneity[12,15]. Given the increased cell plasticity by *Cracd* KO (Fig. 1), we determined the impact of *Cracd* KO on SCLC tumor cell heterogeneity using spatial transcriptomics. We processed lung tumors (RPRP2 vs. CRPR2) for Xenium In Situ (Fig. 2a-c). To compare the heterogeneity of tumor cells in RPR2 and CRPR2 tumors, we examined the cell cluster compositions of RPR2 (4 tumors) and CRPR2 (36 tumors) (Fig. 2d). From a total of 33 cell clusters, 4 tumors of RPR2 were composed of 4 to 7 different cell clusters. However, CRPR2 tumors exhibited a more complex composition than those in RPR2 (Fig. 2e). To spatially visualize this, we superimposed Xenium-derived cluster identities and scRNA-seq–derived cell-cycle assignments onto tumor sections. In RPR2 nodules, clusters occupied discrete, uniform domains, and proliferative (in S and G2/M cell cycle phases) cells were confined to specific regions. In contrast, CRPR2 nodules displayed an intermingling of 15–20 clusters throughout each lesion and a higher overall fraction of proliferative cells. Among these, NE clusters exhibited the highest mitotic index in both genotypes (Fig. 2f–h). Unlike RPR2 tumors showing a high expression of *Ascl1*, CRPR2 tumors exhibited various levels of *Ascl1* expression (T1: Ascl1-negative, T2: Ascl1-low, T2-9: Ascl1-high) (Fig. 2i), which was reproduced in

immunohistochemistry (IHC) for ASCL1 (Fig. 2j). These data show that *Cracd* KO induces heterogeneity in ASCL1 expression in CRPR2, which is in line with scRNA-seq results (Fig. 1, Supplementary Fig. S4).

NOTCH signaling inhibition upregulates *ASCL1*, resulting in NE cell lineage activation[35]. Compared to RPR2, CRPR2 tumors exhibited HES1 downregulation (Supplementary Fig. S5a). We recently reported that CRACD LOF induces NE cell plasticity in LUAD[31]. *Cracd* KO LUAD (*Cracd* KO *Kras*^G12D *Trp53* KO) also showed the downregulation of HES1 (Supplementary Fig. S5b, c). Actin-mediated mechanical force is indispensable for the NOTCH signal transduction[36–42]. As a capping protein inhibitor, CRACD is required for actin polymerization[20]. We confirmed that CRACD depletion disrupted the actin cytoskeleton of RPR2 cells (Fig. 2k). Consistent with the mechanical force-sensing mechanism that activates YAP1 through actin polymerization, YAP1 immunostaining revealed nuclear YAP in RPR2 tumors while CRPR2 tumors showed barely detectable levels of YAP1 (Fig. 2l). Furthermore, treatment of RPR2 cells with verteporfin, a YAP-TEAD inhibitor, resulted in a marked decrease in cleaved NOTCH1 (N1ICD) and its downstream target HES1 (Fig. 2m). We then examined the impact of CRACD depletion on the NOTCH signaling by analyzing the NOTCH1 receptor protein. Compared to RPR2 cells, CRPR2 cells exhibited significantly reduced expression of NOTCH1 protein (uncleaved and cleaved [transmembrane + N1ICD]), which was partially rescued by treatment with N-[N-(3,5-Difluorophenacetyl)-L-alanyl]-S-phenylglycine t-butyl ester (DAPT), a γ-secretase inhibitor (Supplementary Fig. S5d), implying that CRACD depletion inhibits NOTCH1 via NOTCH1 downregulation and cleavage reduction. Next, we tested whether CRACD depletion-induced NE cell plasticity is due to NOTCH signaling downregulation by conducting rescue experiments. To activate the NOTCH signaling, we ectopically expressed the NOTCH1 intracellular domain (N1ICD) in RPR2 or CRPR2 cells. Immunoblot assays showed the upregulation of NE markers (ASCL1, CHGA, and CALCA) and a neuronal progenitor cell lineage marker (ATOH1) in CRPR2 compared to RPR2 (Fig. 2n, lanes 1 vs. 3), which was blocked by N1ICD ectopic expression (Fig. 2n, lanes 3 vs. 4). These results suggest that CRACD LOF induces NE cell plasticity with increased tumor cell heterogeneity mainly via NOTCH signaling downregulation (Fig. 2o).

### Intratumoral CD8+ T cell depletion and MHC-I suppression in *Cracd* KO SCLC tumors

Given the crucial roles of immune cells in tumorigenesis[43,44], we next examined the impact of CRACD loss on the tumor microenvironment. Using scRNA-seq, we profiled immune cells in RPR2 and CRPR2 tumors isolated from GEMMs (Supplementary Fig. S6a–d). CRPR2 tumors barely harbored CD8+ T cells (6.86% [170 of 2477 cells]) compared to RPR2 tumors (65.06% [3484 of 5355 cells]) while showing a slightly higher ratio of naïve T cells to total cell numbers (26.52% [657 of 2477 cells] versus 20.24% [1084 of 5355 cells]) (Fig. 3a–d), which was also confirmed by immunostaining (Supplementary Fig. S6e). Notably, immunostaining also revealed the spatial exclusion of CD8+ T cells in CRPR2 tumors, including peripheral regions. The overall counts of

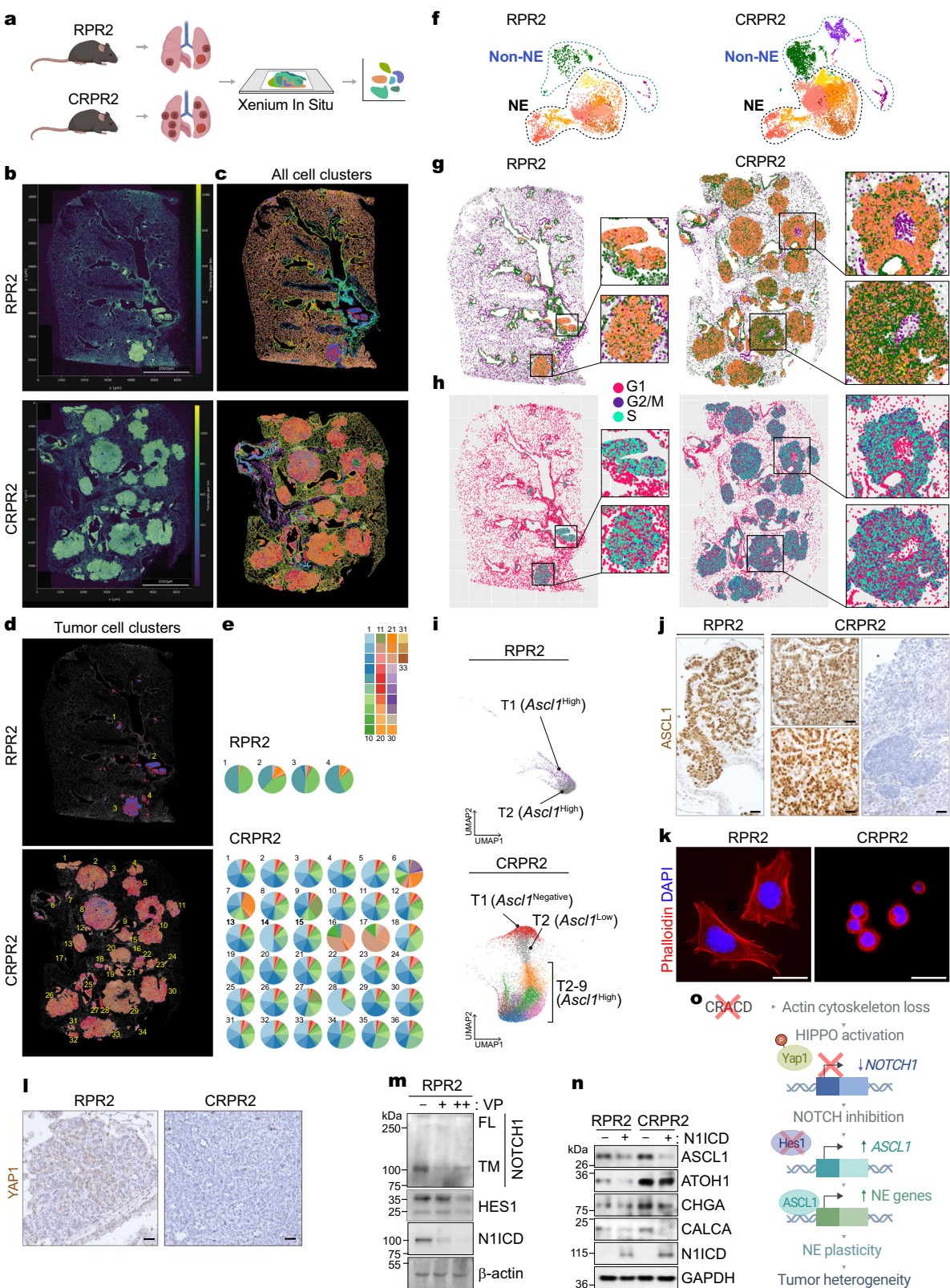

total T cells and apoptotic cells remained similar between RPR2 and CRPR2 tumors (Supplementary Fig. S6e–i). Consistent with this, spatial transcriptomic analysis using Xenium showed differential localization of T cells (*Cd3d, Cd3e, Cd3g, Cd8a, Cd8b1,* and *Cd4*) within tumor sections of RPR2 and CRPR2 mice (Supplementary Fig. S6j, k). Furthermore, the expression of T cell exhaustion markers and their ligands was not affected by *Cracd* KO in CRPR2 tumors compared to RPR2

tumors (Supplementary Fig. S6l, m), indicating that T cell exclusion rather than exhaustion underlies immune evasion in CRPR2 tumors. Additionally, CRPR2 tumors contained more regulatory T cells (Treg) than RPR2 tumors (Fig. 3c, d), with higher expression of genes associated with Treg anti-inflammatory functions, such as *Ahr, Ctla2, Tff1,* and *Tnfrsf18* (Fig. 3e, f). Moreover, compared to RPR2 tumors, CRPR2 tumors displayed a relatively higher number of monocytes (Fig. 3a, b).

**Fig. 2 | *Cracd* KO increases SCLC tumor cell heterogeneity. a–d** Workflow for spatial transcriptomics. Spatial transcriptomic results are shown with transcripts per bin, using a bin size of 20 μm (**b**). Scale bars = 2 mm. Cells were segmented and clustered by transcriptomes. 33 clusters were identified in each sample (**c**), and tumor cell clusters of RPR2 and CRPR2 were highlighted (**d**). Representative Xenium spatial transcriptomic results are shown. The experiment was independently repeated twice on distinct tumor samples with similar results. **e** Pie charts displaying the cell cluster composition for each tumor cell clone in RPR2[80] and CRPR2 (bottom). **f** UMAP projection of scRNA-seq data from RPR2 and CRPR2 tumors. Cells are colored by unsupervised clusters; NE and non-NE populations are delineated by dashed lines. **g** Spatial localization of NE and non-NE clusters in representative RPR2 and CRPR2 tumors. Cluster identities were inferred via correlation-based mapping of Xenium transcriptomic profiles to matched scRNA-seq clusters. **h** Spatial mapping of cell cycle states projected onto the Xenium coordinate space. G1 (magenta), S (aqua), and G2/M (purple). **i** UMAPs of tumors within RPR2 (up) and CRPR2 (down). UMAP coordinates profiling and tumor number annotations (T1-8) were performed using Xenium Explorer. **j** IHC of RPR2 vs. CRPR for ASCL1. Scale bars: 50 μm. Representative images (n ≥ 3). **k** IF staining of RPR2 and CRPR2 cells with phalloidin. Scale bars: 20 μm. **l** IHC of RPR2 vs. CRPR2 for YAP1. Scale bars: 50 μm. Representative images (n ≥ 3). **m** Immunoblot[57] of RPR2 cells treated with vehicle or verteporfin (VP) 0.5 (+) or 1.0 (++) μM. Representative immunoblot from three independent experiments. **n** IB of RPR2 vs. CRPR2 transduced with lentiviruses encoding N1ICD. Representative immunoblot from three independent experiments. **o** Working model: CRACD inactivation activates HIPPO signaling to phosphorylate and inhibit YAP1, leading to NOTCH1 downregulation and ASCL1-mediated NE lineage activation, thereby promoting NE cell plasticity and tumor heterogeneity. **a**, **o** were created with BioRender.com. Source data are provided as a Source Data file.

Given that myeloid-derived suppressor cells (MDSCs) inhibit T cell activation and proliferation[45,46], we also examined the impact of CRACD loss on MDSCs. Compared to RPR2, CRPR2 tumors showed an upregulation of MDSC marker gene expression in myeloid cells (Supplementary Fig. S6n, o). Consistent with the results from the autochthonous model, immune profiling of preSC-derived allograft tumors also displayed a decrease in CD8[+] T cells and an increase in myeloid cells in *Cracd* KO allograft tumors relative to *Cracd* WT tumors (Supplementary Fig. S7a–f).

The altered immune landscape in *Cracd* KO SCLC tumors (Fig. 2a, b) compelled us to determine the underlying mechanism of CRACD depletion-induced CD8[+] T cell loss. We examined the inferred intercellular communication networks between immune cells and SCLC tumor cells (RPR2 vs. CRPR2) using the CellChat package[47]. Overall, CRPR2 tumors showed fewer and weaker cellular interactions among different cell types than RPR2 tumors (Supplementary Fig. S7g). In the information flow maps, RPR2 tumors displayed strong cell-cell interaction between tumor cells and CD8[+] T cells, while CRPR2 tumors showed an interaction between tumor cells and B and myeloid cells (Supplementary Fig. S7h). Notably, the antigen processing and presentation–related pathways were significantly downregulated in CRPR2 tumors relative to RPR2 tumors, mostly between SCLC tumors and CD8[+] T cells (Fig. 3g). The information flow predicted by CellChat nominated differentially regulated pathways between RPR2 and CRPR2 tumors. According to the absolute values and fold changes of information flow, the most downregulated pathway in CRPR2 was the MHC-I pathway (Fig. 3h). The circle plots validated that the MHC-I pathway was barely detected in CRPR2 tumors but was prevalent in RPR2 tumors (Fig. 3i). Moreover, the GSEA of scRNA-seq datasets confirmed the downregulation of the gene sets associated with the MHC-I pathway (Fig. 3j). Additionally, *H2-Q1/2/4* and *H2-T3*, genes encoding the α chain of the mouse MHC-I complex were downregulated in CRPR2 tumors compared to RPR2 tumors (Fig. 3k), also validated by IHC for MHC-I (Fig. 3j). These data suggest that *Cracd* KO is associated with intratumoral CD8[+] T cell depletion and the MHC-I pathway suppression.

**CRACD depletion epigenetically suppresses the MHC-I pathway via EZH2 for immune evasion**
Next, we explored how CRACD depletion suppresses MHC-I gene expression. Beyond its role in the cytoskeleton, nuclear actin (N-actin) modulates gene expression, RNA splicing, translation, and DNA repair[48]. Since CRACD promotes actin polymerization[20], we examined whether CRACD is involved in N-actin dynamics. We visualized N-actin in RPR2 and CRPR2 cells using plasmids encoding N-actin chromobody[49]. RPR2 cells showed enrichment of N-actin, while CRPR2 cells displayed reduced N-actin levels (Fig. 4a).

N-actin is essential for epigenetic gene regulation[50,51]. N-actin depletion has been shown to promote EZH2-mediated gene repression[51–53]. Therefore, we hypothesized that EZH2 mediates *Cracd* KO-induced MHC-I transcriptional suppression. We compared the histone modifications between RPR2 and CRPR2 cells. Immunostaining of RPR2 and CRPR2 tumors showed decreased H3K27ac and increased H3K27me2 and H3K27me3, histone modifications induced by EZH2 methyltransferase in CRPR2 (Fig. 4b, c). Next, RPR2 and CRPR2 cell lines were subjected to Cleavage Under Targets and Release Using Nuclease (CUT&RUN) sequencing with anti-EZH2 antibody. Compared to RPR2 cells, EZH2's promoter occupancy on the transcriptional start sites (TSS) was overall elevated in CRPR2 cells (Fig. 4d). Moreover, the MHC-I genes (H2-D1, H2-Q1 - Q10) exhibited the enrichment of EZH2 on TSS (Fig. 4e, f).

To test whether MHC-I suppression in CRPR2 is EZH2-dependent, we treated CRPR2 cells with GSK343, an EZH2 inhibitor. GSK343 treatment was sufficient to de-repress MHC-I protein (Fig. 4g). Similarly, we treated CRACD-depleted murine (CRPR2) and human SCLC cells (NCI-H2081 carrying an endogenous frame-shift mutation in *CRACD* [Q168Tfs*17]) with GSK343 and assessed MHC-I gene expression. EZH2 inhibition restored the expression of MHC-I genes (murine: *H2-Q1/2/4, H2-T3*; human: *HLA-A/B/C*) in these CRACD-inactivated cells (Fig. 4h, i). Having observed NE cell plasticity induction via NOTCH signaling downregulation (Fig. 2, Supplementary Fig. S4), we also tested the potential interplay between NOTCH signaling and EZH2-mediated MHC-I suppression by ectopically expressing N1ICD. N1ICD overexpression did not affect MHC-I expression (Fig. 4j), suggesting that NOTCH signaling is not involved in the EZH2-repressed MHC-I pathway. These findings indicate that CRACD inactivation suppresses MHC-I expression through EZH2-mediated histone methylation (Fig. 4k).

Having determined that EZH2 blockade restores the MHC-I expression in CRACD-inactivated SCLC tumors (Fig. 4g–i), we hypothesized that EZH2 inhibitors suppress CRACD-inactivated SCLC tumorigenesis by reactivating MHC-I-based tumor antigen presentation. We assessed the impact of EZH2 inhibitors on the proliferation of RPR2 and CRPR2 cells in vitro. RPR2 (*Cracd* WT) and CRPR2 (*Cracd* KO) cells treated with GSK343 or tazemetostat, an FDA-approved EZH2 inhibitor, did not exhibit significant differences in growth inhibition between RPR2 and CRPR2 cells in vitro (Supplementary Fig. S8a). Next, we performed syngeneic transplantation of RPR2 or CRPR2 cells into C57BL/6 mice, followed by administration of GSK343 or tazemetostat. Compared to the control (vehicle only), EZH2 inhibitors significantly suppressed CRPR2 tumorigenesis (Fig. 4l, m). GSK343 had a more pronounced effect on SCLC tumor suppression than tazemetostat (Fig. 4l, m). Furthermore, tumor immunostaining showed that EZH2 inhibition reduced cell proliferation (MKI67), increased cell death (cleaved Caspase-3 [CC3]), and restored MHC-I expression in CRPR2 tumors (Fig. 4n). Fluorescence-activated cell sorting (FACS) analysis revealed that EZH2 inhibitors markedly increased the number of intratumoral CD8[+] T cells in CRPR2 tumors, with CD4[+] T cells being elevated only by GSK343 treatment (Fig. 4o, p, Supplementary

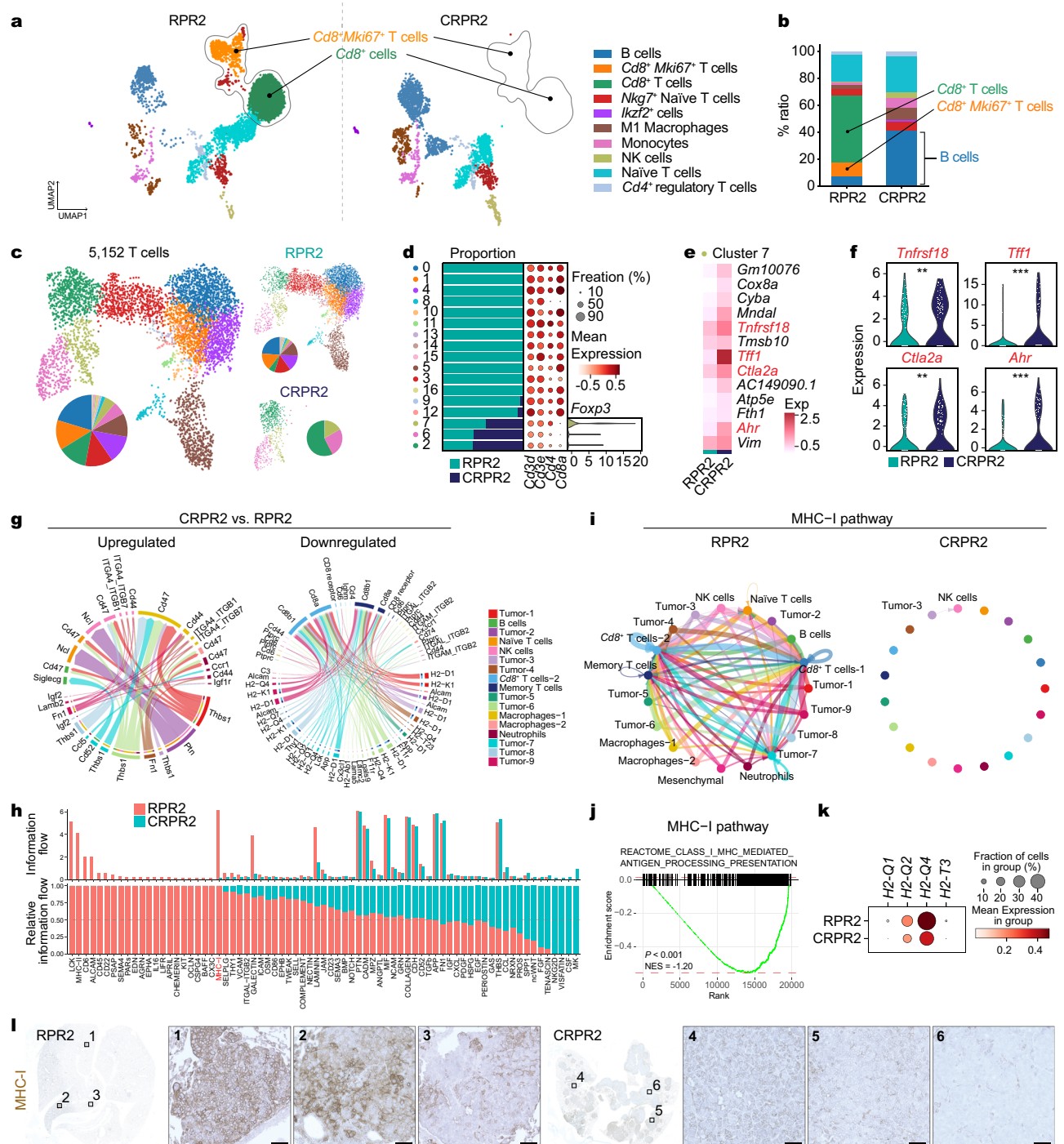

**Fig. 3 | Depletion of intratumoral CD8+ T cells in *Cracd* KO SCLC tumors.**
**a** UMAPs of different immune cell types. **b** Comparison of immune cell proportions between the RPR2 and CRPR2 datasets. **c–f** Single-cell transcriptomic analysis of T cells reveals a CRPR2-enriched inflammatory subpopulation. **c** UMAP visualization of 5,152 T cells from RPR2 and CRPR2 tumors, colored by cluster identity. Pie charts indicate the cluster composition within each sample. **d** Bar plot showing cluster proportions and dot plot displaying average expression of major T cell subtype markers across clusters. CD8+ T cell subclusters were selectively depleted in CRPR2, whereas cluster 7, enriched in CRPR2, exhibited high expression of *Foxp3*, indicative of regulatory T cells. **e** Heatmap of differentially expressed genes within cluster 7, comparing CRPR2 and RPR2 T cells. CRPR2 cluster 7 cells showed upregulation of immunoregulatory genes, including *Tnfrsf18*, *Tff1*, *Ctla2a*, and *Ahr*. **f** Violin plots showing significantly higher expression of these genes in CRPR2 cluster 7 cells compared to RPR2 (Wilcoxon rank-

sum test; **P < 0.01, ***P < 0.001). **g** Chord plots showing up- and down-regulated signaling pathways in the CRPR2 dataset compared to RPR2, analyzed using CellChat. Inner bar colors represent receiving cell clusters; chord thickness indicates ligand–receptor interaction strength. **h** Overall information flow (upper) and relative information flow (lower) of each signaling pathway in RPR2 and CRPR2 tumors, analyzed using CellChat. **i** Circle plots displaying the inferred network of the MHC-I signaling pathway in RPR2 (left) and CRPR2 tumors (right); the thickness of each line connecting the cell clusters indicates the interaction strength, analyzed using CellChat. **j** GSEA of gene sets associated with the MHC-I pathway in CRPR2 datasets compared to RPR2 scRNA-seq datasets; NES, normalized enrichment score. **k** Dot plot displaying the expression level of the MHC-I pathway-related genes in the RPR2 and CRPR2 datasets. **l** IHC of RPR2 and CRPR2 tumors for MHC-I; scale bars, 50 µm. Representative images are shown (n ≥ 3). Source data are provided as a Source Data file.

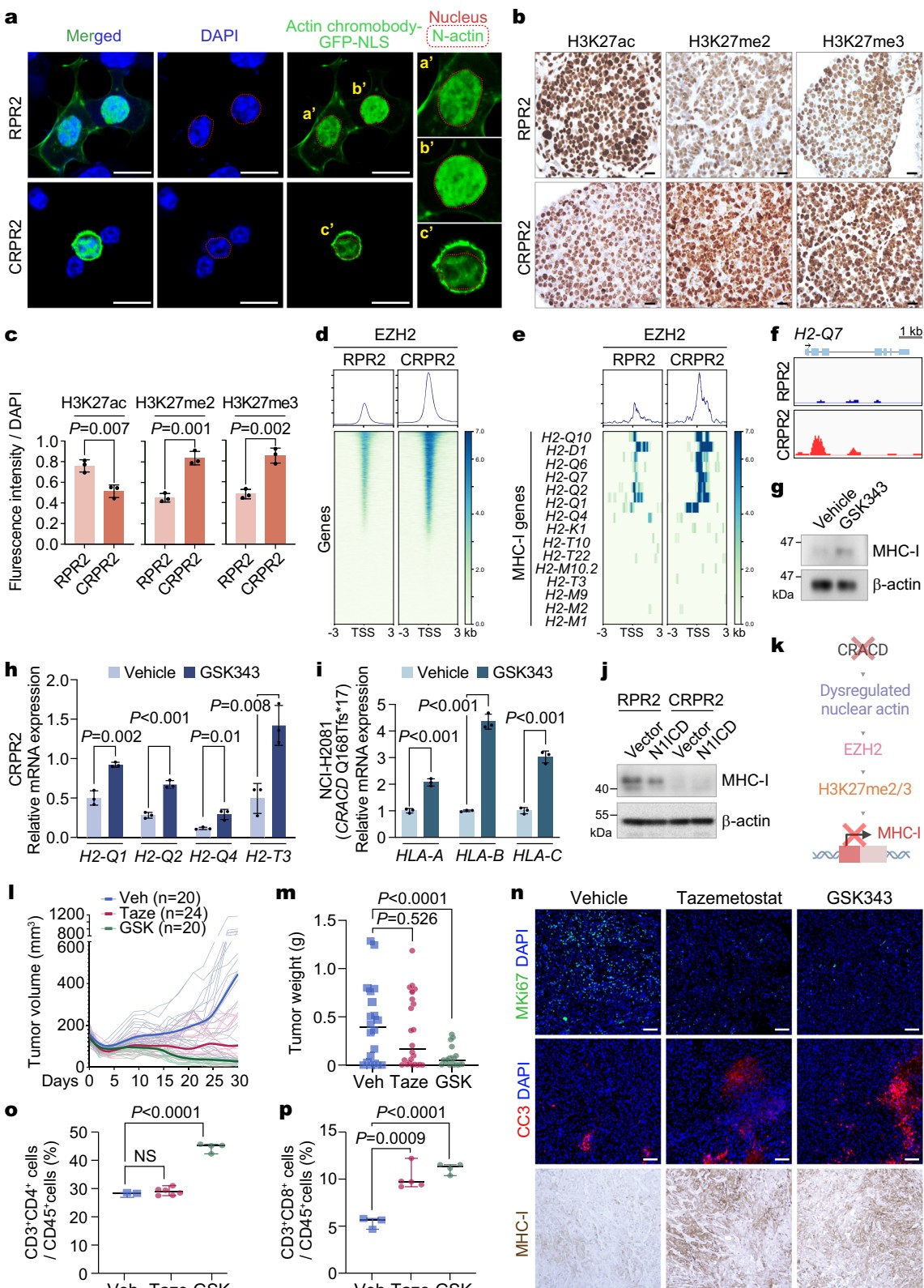

Fig. S8b, c). Notably, RPR2 cells rarely formed tumors in C57BL/6 mice within 30 days post-transplantation (Supplementary Fig. S8d, e).

To assess the functional status of intratumoral CD8[+] T cells, we conducted co-culture assays using CD8[+] T cells and CRPR2 tumor cells. Our results showed that EZH2 inhibition restored Perforin[+] CD8[+] T cell populations, indicating enhanced functional cytotoxic activity against CRPR2 tumor cells (Supplementary Fig. S8f). To further investigate the

role of CD8[+] T cells in tumor growth, we performed CD8 depletion experiments. Depletion of CD8[+] T cells in recipient C57BL/6 mice significantly increased the growth of RPR2 tumor cells. In contrast, the primary cells from CRPR2 tumors, which already exhibit low CD8[+] T cell infiltration, showed no significant difference in growth with or without CD8 depletion (Supplementary Fig. S8g). These findings suggest that the immune evasion phenotype of CRPR2 tumors is largely

**Fig. 4 | Immune evasion of CRPR2 tumors by EZH2-mediated MHC-I suppression. a** IF staining of RPR2 and CRPR2 cells transfected with Actin Chromobody-GFP-NLS plasmids; scale bars, 50 μm; a'-c', magnified images; red dot lines, nuclei. **b** IHC of SCLC tumors isolated from GEMMs (RPR2 vs. CRPR2) for histone modifications (H3K27ac, H3K27me2, and H3K27me3); scale bars, 20 μm. Representative images are shown for panels a-b (n ≥ 3). **c** Quantification of histone modifications (IF images) using ZEN software. Data are presented as mean values ± SD; two-sided Student's t-test. **d, e** Heatmaps showing EZH2 enrichment at transcription start sites (TSS) of global (**d**) and MHC-I genes (**e**) in RPR2 and CRPR2 cells by CUT&RUN-seq. **f** EZH2 occupancies on the *H2-Q7* promoter, visualized by IGV. **g** IB of CRPR2 cells treated with GSK343 (50 μM, 72 hrs) for MHC-I. Representative of three independent experiments. **h, i** RT-qPCR analysis of genes related to the mouse MHC-I pathway after 72 hr of treatment of the CRPR2 (**h**) and NCI-H2081 (**i**) cells with GSK343 (20 μmol/L). Data are presented as mean values ± SD. Exact two-sided *P* values are annotated on the plots (n = 3 independent biological replicates). **j** IB of RPR2 or CRPR2 cells transduced with lentiviruses encoding N1ICD for MHC-I. Representative immunoblot from three independent experiments. **k** Illustration of EZH2-mediated epigenetic suppression of the MHC-I genes by CRACD inactivation. **l** Tumor growth curves of subcutaneous CRPR2 treated with vehicle (Veh), tazemetostat (Taze; 200 mg/kg, n = 24), or GSK343 (GSK; 20 mg/kg, n = 20). Darker lines, median values of each group. **m** Tumor weights for each group: Veh (n = 20), Taze (n = 24), GSK343 (n = 20). Data are mean ± SD; two-sided Student's t-test; error bars, SD. **n** IF staining for MKI67 and Cleaved Caspase-3 (CC3) and IHC for MHC-I in CKP tumor sections. Scale bars, 100 μm. Representative images (n ≥ 3). **o, p** Quantification of CD4+CD3+ (**o**) and CD8+CD3+ (**p**) in CD45+ cells from CRPR2 tumors treated with EZH2 inhibitors. Data are illustrated as mean ± SD (n = 3 independent assays); two-sided Student's t-test. **k** was created with BioRender.com. Source data are provided as a Source Data file.

due to their ability to bypass CD8+ T cell-based immune surveillance. Beyond MHC-I regulation, we sought to determine whether EZH2 inhibition broadly alters oncogenic transcriptional programs in CRPR2 cells. Bulk RNA-seq followed by GSEA revealed that tazemetostat downregulates transcription of MYC targets, DNA repair, ribosome biogenesis, and RNA metabolic processes, while upregulating genes related to KRAS signaling and EMT pathways (Supplementary Fig. S9a-d). These data suggest that EZH2 inhibitors also exert tumor suppressive effects through global transcriptional reprogramming, in addition to enhancing tumor immunogenicity. These results indicate that CRACD inactivation induces EZH2-mediated suppression of MHC-I for immune evasion of SCLC tumor cells.

## Pathological relevance of CRACD and the MHC-I pathway in human SCLC

To determine the pathological relevance of the data from *Cracd KO* SCLC mice to human SCLC, we analyzed scRNA-seq datasets of 19 SCLC patient tumor samples and eight normal human lung samples from the previous studies[10,54] (Fig. 5a, Supplementary Data 7, 8). An unbiased pair-wise correlation analysis of tumor cells divided the SCLC tumor datasets into two major groups (MS1 [molecular subtype 1] and MS2) (Fig. 5b). The MS1 SCLC tumors were clinically associated with recurrence (2 of 3) and metastasis (1 of 3), whereas the MS2 is associated with primary tumors (6 of 16) and metastasis (7 of 16) (Fig. 5c). A copy number variation analysis showed relatively higher genomic instability in the MS2 than in the MS1 tumors (Supplementary Fig. S10a, b). According to the ANPY classification[9], the MS1 was mainly categorized as the ASCL1-type (Fig. 5d). CRACD expression was downregulated in MS1 compared to MS2 (Fig. 5e). Similarly, the scores of EZH2 target genes and NOTCH signaling were also notably reduced in MS1 (Fig. 5e, f).

Compared to MS2, the MS1 tumors expressed relatively lower levels of the genes encoding MHC-I and several of the antigen processing and presentation pathway components (*HLA-A, B, C, E, LMP2/LMP7*, and *TAP1/2*) (Fig. 5g), also confirmed by the GSEA results (Fig. 5h). Additionally, we observed positive correlations between *CRACD* and *HLA-A/E* expression in the TCGA datasets of SCLC bulk RNA-seq (Fig. 5i). Additionally, IHC analysis of human SCLC tissue microarrays confirmed that tumors with low CRACD expression showed reduced MHC-I and elevated H3K27me2/3 levels. In contrast, CRACD-high tumors exhibited higher MHC-I and lower H3K27me2/3 expression (Supplementary Fig. 11a, b), supporting the epigenetic silencing of the MHC-I pathway in CRACD-inactivated SCLC. Collectively, these data demonstrate that CRACD inactivation is pathologically associated with the downregulation of the tumor antigen processing and presentation pathway of human SCLC (Fig. 5j).

## Discussion

Since CRACD is often inactivated in SCLC, we determined the impact of CRACD LOF on SCLC tumorigenesis by using preneoplastic SCLC cells and GEMMs. Our results from preclinical models demonstrated that CRACD functions as a tumor suppressor of SCLC. We identified two significant outcomes of CRACD depletion in SCLC: NE cell plasticity and immune evasion.

Our data suggest that multiple signaling pathways mediate CRACD loss-driven NE cell plasticity in two distinct tumor cells (NE and non-NE). In CRPR2 tumors, the upregulation of NE genes in the NE cells is mainly due to the downregulation of NOTCH signaling. Mechanical pulling force generated by the actin cytoskeleton is required for NOTCH signaling activation via receptor endocytosis, ligand-receptor binding, and NOTCH cleavage[36–42]. However, in the condition of CRACD inactivation, the disrupted actin cytoskeleton suppresses NOTCH signaling, de-repressing *ASCL1* and activating its downstream NE cell lineage genes (Fig. 2, Supplementary Fig. S4). This is also confirmed by another result that N1ICD inhibited the NE gene upregulation induced by CRACD loss (Fig. 2n), reiterating that NOTCH signaling downregulation is crucial for NE gene upregulation in the NE cells. Conversely, non-NE cells of CRPR2 tumors displayed the activation of NOTCH, MYC, WNT, and EMT pathways (Supplementary Fig. 4). These findings are also consistent with the Julien Sage laboratory's report on the heterogeneity of NOTCH signaling activity and NE phenotype in SCLC[16]. In RPR2 SCLC mice, non-NE tumor cells showed high NOTCH signaling activity and are relatively less proliferative, whereas NOTCH-inactive NE tumor cells are highly proliferative[16], similar to our observation (Fig. 1i, j), which might be the reason why *Cracd*-depleted preSC cells displayed cell hyperproliferation in vitro (Supplementary Fig. S1a-c). In addition to our in vitro and in vivo data, the correlation between *CRACD* low and NOTCH signaling downregulation in patients' SCLC tumors (Fig. 5f) implies that CRACD or the actin pathway might be one of the key determinants positively modulating the NOTCH signaling beyond its role in maintaining the structural integrity of epithelial cells.

In intestinal epithelial and colorectal cancer cells, CRACD loss triggers the release of β-catenin from the cadherin-catenin-actin complex, inducing β-catenin-transactivated WNT target genes[20], including MYC, which might explain WNT and MYC activation in non-NE cells of CRPR2 tumors. Another question is how CRACD loss leads to two opposite outcomes in different cell types: NOTCH signaling inhibition in NE and activation in non-NE cells. Considering other capping protein inhibitors (CPIs), such as CARMILs, it is possible that, unlike NE cells, CRACD loss might be complemented by these CPIs in non-NE cells where NOTCH signaling is not downregulated. Conversely, in non-NE cells, WNT signaling likely activates NOTCH signaling, as previously demonstrated in different contexts[55].

CRACD depletion globally induces EZH2-mediated transcriptional suppression of genes, including ones encoding the MHC-I (Fig. 4). This epigenetic reprogramming renders tumor cells resistant to CD8+ cytotoxic T cells, contributing to the 'cold tumor' phenotype characterized by the absence of T cells in the tumor microenvironment (Fig. 3). Our data further suggest that aberrant N-actin-mediated

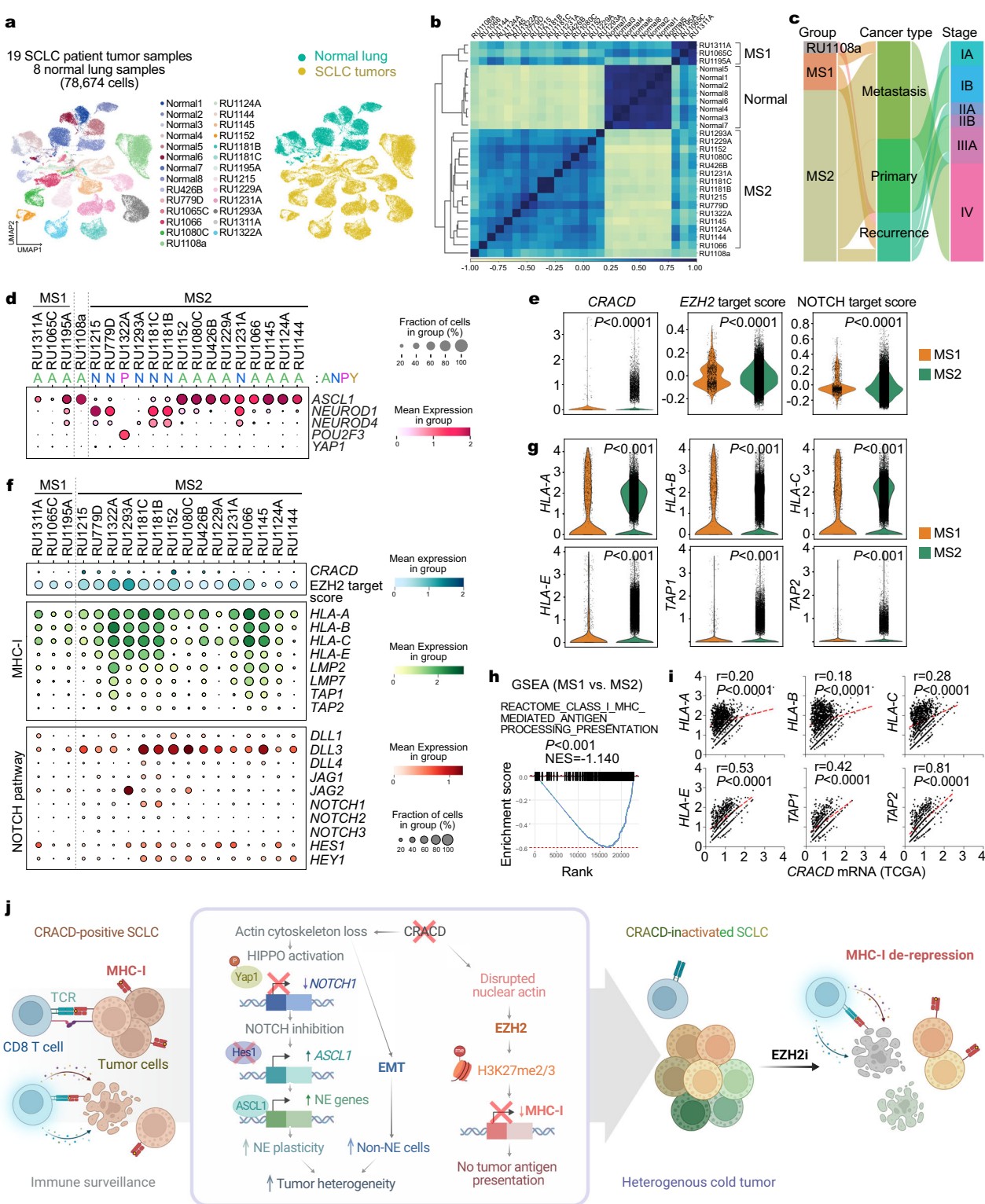

chromatin architecture resulting from CRACD loss is an underlying mechanism of the epigenetic changes. Emerging evidence indicates that N-actin is vital in organizing chromatin architecture[48,52,53,56]. The genetic ablation of *Actb* encoding β-actin increases genome-wide H3K27 methylation levels and EZH2's promoter occupancy[52,53,56]. CRACD loss reduces N-actin and increases H3K27 methylation on the promoters of the MHC-I genes (Fig. 4). We recently observed that the loss of E-cadherin also induces EZH2-mediated gene repression, leading to the development of diffuse-type gastric cancer[57]. Since CRACD loss also disrupts the E-cadherin-catenin-actin complex[20], it is highly

plausible that epithelial cell integrity loss might be functionally linked to EZH2-mediated transcriptional reprogramming. Moreover, our data indicate that overexpression of N1ICD does not restore MHC-I expression in CRPR2 cells (Fig. 4j), suggesting that EZH2-mediated gene repression is independent of NOTCH signaling. In addition to its role in restoring MHC-I expression, EZH2 inhibitors may have further tumor-suppressive effects on SCLC through broader transcriptional reprogramming, which includes the hyperactivation of ribosome biogenesis, epithelial-mesenchymal transition, and MYC targets (Supplementary Fig. S9a–d).

**Fig. 5 | Pathological relevance of the CRACD-EZH2-MHC-I axis in SCLC. a** UMAPs of SCLC tumor cells from 19 SCLC patient tumor samples (54,633 cells) and 8 normal lung samples (24,041 cells). Each dot represents a single cell, colored by a human sample ID (left) and SCLC vs. normal (right). **b** Correlation matrix plot showing pair-wise correlations among the human normal lung and 19 patient tumor samples. The dendrogram shows the distance of each dataset based on principal component analysis, and Pearson's correlation is displayed with a color spectrum. Groups of patients were unbiasedly categorized by dendrogram and correlation. **c** Sankey plot shows the correlation between SCLC subtypes (MS1 and MS2) and clinical information (cancer type and stage). **d** Dot plot showing NE marker gene expression in 19 SCLC patient samples. ANPY (*ASCL1*, *NEUROD1*, *POU2F3*, and *YAP1*)-based classification was noted at the top. **e, f** Violin (**e**) and dot (**f**) plots showing CRACD mRNA and EZH2/NOTCH-target scores. *P* values: two-sided Student's *t*-test. **g** Violin plots of MHC-I pathway gene expression in MS1 vs MS2 tumors; two-sided Student's *t*-test. **h** GSEA of gene sets associated with EZH2 targets and the MHC-I pathway in MS1 compared to MS2. **i** Correlation scatter plots for Pearson's correlation analysis (using GraphPad Prism) of *CRACD* and MHC-I genes (*HLA-A, B, C, E,* and *TAP1/2*) in SCLC patient tumor cells based on the TCGA bulk RNA-seq datasets. r, Pearson correlation coefficient; *P* values were calculated using two-sided Student's *t*-test. **j** Illustration of the impact of CRACD loss on SCLC tumorigenesis. CRACD-positive tumors maintain MHC-I–mediated immune activity ("hot tumors"), whereas CRACD-deficient tumors acquire NE plasticity and immune evasion. Loss of CRACD disrupts actin polymerization, activating HIPPO signaling and LATS1/2-mediated YAP1 inhibition, which suppresses NOTCH1 and induces ASCL1-driven NE gene expression. Concurrently, EZH2-dependent silencing of MHC-I genes converts tumors to immune "cold" states, accelerating tumor progression. **j** was partly created with BioRender.com. Source data are provided as a Source Data file.

Cancer immunotherapy has faced challenges due to primary and acquired resistance. Thus, identifying key determinants of sustained therapeutic benefit from ICB could inform strategies to overcome therapeutic resistance and personalize SCLC therapy. Through unsupervised clustering of tumor cells from the scRNA-seq datasets, we identified the distinct subtype (MS1) of human SCLC characterized by *CRACD* [low], EZH2-mediated gene repression, and MHC-I pathway suppression, distinguished from MS2, with *CRACD* [high] and a functional MHC-I pathway. Given the MHC-I pathway suppression in MS1, patients belonging to MS1 may not exhibit a favorable response to T cell-based ICB, making them non-responders. Restoring the MHC-I pathway, for example, by inhibiting EZH2, reverses the immune-cold phenotype commonly observed in human SCLC into hot tumors (Fig. 4l). Hence, EZH2 blockade may be a promising therapeutic strategy for patients with CRACD-inactivated SCLC. It is noteworthy that targeting other essential epigenetic regulators, such as the lysine demethylase LSD1, has also been shown to restore the MHC-I pathway and sensitize SCLC to ICB[19,58]. In addition to the ANPY classification, this study proposes another approach to stratify SCLC patients based on CRACD status, providing a potential predictive molecular signature for the effectiveness of T cell-based ICB therapies combined with EZH2 inhibitors. EZH2 inhibitors have shown therapeutic promise not only in SCLC but also in other cancers, including prostate and breast cancer models, further supporting the rationale for their clinical evaluation[59].

It remains unclear when and where CRACD inactivation occurs during tumorigenesis. This spatiotemporal information is necessary for a better understanding of the pathobiology of CRACD-inactivated SCLC tumorigenesis. Interestingly, MS1 (*CRACD* [low]) is only associated with recurrent (2 of 3) or metastatic (1 of 3) SCLC but with primary tumors (Fig. 5c), implying that CRACD LOF might take place at later stages or during therapies. Tumor cell plasticity contributes to therapy resistance and metastasis[13]. Similar observations have been reported in other malignancies, such as prostate cancer and pancreatic neuroendocrine tumors, where lineage plasticity is closely linked to therapy resistance and disease progression[60–62]. Therefore, the impact of CRACD loss-driven cell plasticity on SCLC therapy resistance and metastasis requires further investigation. Additionally, despite our intriguing results with EZH2 inhibitors (Fig. 4l–p), EZH2 monotherapy may not suffice in clinical trials. Therefore, future studies should explore combination therapy with other agents, including ICB. Our pathway analysis of scRNA-seq datasets showed a significant activation of the ribosome biogenesis pathway in CRPR2 tumors compared to RPR2. Targeting protein translation with omacetaxine mepesuccinate (homoharringtonine) may enhance treatment efficacy in SCLC, opening a new avenue for combinatorial therapy with EZH2 inhibitors. In addition to enhancing drug efficacy, identifying specific patients likely to respond well is crucial, which could be addressed by our finding that SCLC patients with *CRACD* [low] tumors may benefit from combining EZH2 inhibitors and immunotherapy.

In summary, our study provides new insights into the mechanisms of SCLC tumorigenesis by uncovering the unexpected role of CRACD, an actin regulator, in limiting cell plasticity and inhibiting tumor immune evasion. Additionally, it highlights the potential therapeutic application of EZH2 inhibitors in treating CRACD-inactivated SCLC tumor cells.

## Methods

The research reported here complies with all relevant ethical regulations. All mouse experiments were approved by the Institutional Animal Care and Use Committees (IACUC) of The University of Texas MD Anderson Cancer Center (protocol #00002414-RN00) and the University of Virginia (protocol #3967) and were performed in accordance with institutional guidelines and the Association for Assessment and Accreditation of Laboratory Animal Care (AAALAC) International standards.

### Mammalian cell culture

Human embryonic kidney 293 T (HEK293T) and NCI-H2081used in this study were purchased from the American Type Culture Collection (ATCC). The murine preSC cells have been previously described[25,63]. RPR2 and CRPR2 cell lines were established from the SCLC tumors isolated from each strain. HEK293T, preSC, RPR2, and CRPR2 cells were maintained in Dulbecco's Modified Eagle's Medium (DMEM) medium containing 10% fetal bovine serum (Thermo Fisher Scientific) and 1% penicillin and streptomycin (Thermo Fisher Scientific). NCI-H2081 was maintained in DMEM: F-12 medium (5% FBS, 1% penicillin/streptomycin, 0.005 mg/mL Insulin, 0.01 mg/mL Transferrin, 30 mmol/L Sodium selenite, 10 mmol/L Hydrocortisone, 10 mmol/Lb-estradiol, 2 mM L-glutamine). Cells were cultured at 37 °C in a humidified incubator supplied with 5% $CO_2$ air. Mycoplasma contamination was examined using the MycoAlert mycoplasma detection kit (Lonza). See Supplementary Data S1 for reagent information.

### CRISPR/Cas9 gene knockout

CRISPR/Cas9-mediated *Cracd* KO in preSC cells was performed according to Zhang laboratory's protocol[64]. Control sgRNA sequence target EGFP: 5'-GGGCG AGGAG CTGTT CACCG-3'; sgRNA sequence target *Cracd*: 5'-ACACA CGGCC ATTTT GGTCA-3'. The sgRNA sequence is based on our previous study[20].

### Virus production and transduction

HEK293T cells in a 10-cm dish were co-transfected with 5 μg of constructs, 5 μg of plasmid D8.2 (Plasmid #8455, Addgene), and 3 μg of plasmid VSVG (Plasmid #8454, Addgene). Cells were incubated at 37 °C, and the medium was replaced after 12 h. Virus-containing medium was collected 48 h after transfection and supplemented with 8 μg/mL polybrene to infect target cells in 6-well dishes. After 6 h, the

medium was changed. After 48 h, the infected cells were selected with 2 µg/mL puromycin.

## Plasmids

Nuclear Actin Chromobody®-TagGFP plasmid (Chromotek) was transfected using Lipofectamine 3000. For NOTCH signaling activation, N1ICD plasmids (Addgene #17623) were used for virus packaging and transduction.

## qRT-PCR

RNAs were extracted by TRIzol (Invitrogen) and used to synthesize cDNAs using the iScript cDNA synthesis kit (Biorad). qRT-PCR was performed using an Applied Biosystems 7500 Real-Time PCR machine with the primers listed in Supplementary Data S2. Target gene expression was normalized to that of mouse *Hprt1* and human *HPRT1*. Comparative $2^{-\Delta\Delta Ct}$ methods were used to quantify qRT-PCR results. See Supplementary Data S2 for primer information.

## Cell proliferation and viability assays

We counted the number of cells using a hematocytometer (Bio-Rad) on growth days according to the manufacturer's protocol. Cell proliferation was determined by crystal violet staining or Cell Counting Kit-8 (Dojindo Laboratories) according to the manufacturer's protocol. For crystal violet staining, plates were rinsed with Phosphate-buffered saline (PBS), fixed with 4% paraformaldehyde solution for 20 min, and stained with crystal violet solution (0.1% crystal violet, 10% methanol) for 20 min followed by rinsing with tap water.

## Immunoblotting

Whole-cell lysates of cells were prepared using radio-immunoprecipitation assay (RIPA) buffer with protease inhibitors for 30 min at 4 °C, followed by centrifugation (4 °C, $16,000 \times g$ for 15 min). Supernatants were denatured in 5 × Sodium dodecyl-sulfate (SDS) sample buffer (200 mmol/L Tris-HCl [pH 6.8], 40% glycerol, 8% SDS, 200 mmol/L dithiothreitol, and 0.08% bromophenol blue) at 95 °C for 5 min, followed by Sodium dodecyl-sulfate polyacrylamide gel electrophoresis (SDS-PAGE). We used 2% non-fat dry milk in Tris-buffered saline and Tween-20 (25 mmol/L Tris-HCl pH 8.0, 125 mmol/L NaCl, and 0.5% Tween-20) for immunoblot blocking and antibody incubation. SuperSignal West Pico and Femto reagents (Thermo Fisher Scientific) were used to detect horseradish peroxidase-conjugated secondary antibodies. Detailed information on the antibodies is shown in Supplementary Data S1.

## Immunofluorescence microscopy

Cells were fixed for 20 min in 4% paraformaldehyde and permeabilized with 0.1% Triton X-100 (in PBS) for 10 min. After three PBS washes, cells were blocked with 2% bovine serum albumin (BSA) for 30 min at ambient temperature. Cells were then incubated with antibodies diluted in 2% BSA at 4 °C overnight. After three PBS washes, the cells were incubated with 1 µg/mL Alexa fluorescence-conjugated secondary antibodies (Invitrogen) by shaking at ambient temperature in the dark for 1 h. Cells were washed three times with PBS in the dark and mounted in Prolong Gold Antifade Reagent (Invitrogen). Immunofluorescent staining was observed and analyzed using confocal or fluorescent microscopes (Zeiss) and ZEN software (Zeiss).

## Animals

Immunocompromised (BALB/c athymic nude, *Mus musculus*) mice and C57BL/6 mice were purchased from the Jackson Laboratory (Maine, USA). Compound transgenic mice *Rb1*^lox/lox *TrpS3*^lox/lox *Rbl2*^lox/lox (RPR2) mice have been previously described[63]. For SCLC tumor induction, the lungs of 10-week-old mice were infected with adenoviral Cre via intratracheal instillation as previously described[63,65]. Multiple cohorts of independent litters were analyzed to control for background effects, and both male and female mice were used in all experiments. Ad-Cracd-Cre particles were produced in the Vector Development Laboratory at Baylor College of Medicine. Mice were euthanized by $CO_2$ asphyxiation followed by cervical dislocation at the indicated time. Tumors were harvested from euthanized mice, fixed with 10% formalin, embedded in paraffin, and sectioned at 5-µm thickness. The sections were stained with hematoxylin and eosin for histological analysis. Mice were housed in a pathogen-free facility under controlled conditions with a 12-h light/dark cycle, ambient temperature of ~22 °C, and relative humidity of 40–70%, with ad libitum access to food and water. All mice were maintained in compliance with the guidelines of the Institutional Animal Care and Use Committee of the University of Texas MD Anderson Cancer Center and the University of Virginia School of Medicine. All animal procedures were performed based on the guidelines of the Association for the Assessment and Accreditation of Laboratory Animal Care and institutional (MD Anderson and the University of Virginia) approved protocols. This study was compliant with all relevant ethical regulations regarding animal research.

## Syngeneic models

C57BL/6 mice (4 months old) were purchased from the Jackson Laboratory. Mice were randomized and subcutaneously injected with 1 × 10^6 cells into both flanks. Mice were maintained in the Division of Laboratory Animal Resources facility at MD Anderson. Starting on day 4 after transplantation, mice were administered with tazemetostat (200 mg/kg; oral gavage) and GSK343 (20 mg/kg; intraperitoneal injection). Drug treatments were carried out approximately for 4 weeks, with administration every other day. Tumor volume was monitored and calculated by measuring with calipers every 2 days (volume = [length × width$^2$] / 2). The maximal tumor size permitted by the institutional animal care and use guidelines (≤1.5 cm in diameter or ≤10% of body weight) was not exceeded in any experiments. Mice were monitored daily, and animals showing signs of distress or tumor ulceration were euthanized according to IACUC-approved protocols. Tumor burden was calculated by measuring all tumor lesions within the lung to account for the complete tumor burden. On day 30, mice were euthanized (in CRPR2 tumors), tumors were photographed, and collected to proceed for paraffin-embedding and subsequent immunostaining or scRNA-seq. In the case of RPR2 tumors, drug treatment started when the tumors reached ~100 mm$^3$, which occurred around 12 days after cell injection. The mice were treated with drugs for 28 days and euthanized on day 40.

## CD8$^+$ T cell depletion models

CD8$^+$ T cell depletion was performed by intraperitoneal injection of 300 µg of anti-CD8 monoclonal antibody (clone 53-6/7, BioXcell) twice a week on days −2, 0, and 2, where day 0 corresponds to the day of tumor injection. Mice were euthanized on day 21.

## Co-culturing tumor cells with CD8$^+$ T cells

Mouse CD8$^+$ T cells were isolated from spleens according to the manufacturer's instructions. Isolated CD8$^+$ T cells were activated with anti-CD3 (1 µg/ml) and anti-CD28 (5 µg/ml) antibodies in the presence of 50 U/mL recombinant IL-2 (Peprotech) for 24 h. Tumor cells were seeded in 24-well plates at a density of 1 × 10^6 cells per well and allowed to adhere overnight. Activated CD8$^+$ T cells were then co-cultured with tumor cells at a 2:1 ratio (T cells to tumor cells).

## Mouse lung tissue collection

Lungs were harvested at a single fixed timepoint—6 months after intratracheal Cre administration—when RPR2 mice reliably reach the tumor burden and humane endpoints defined in our approved animal protocol[29,66]. This timing ensures adequate tumor development for assessing the impact of CRACD loss.

## Mouse lung tumor and allograft tumor preparation

Prior to processing, mouse SCLC and allograft tumors were decontaminated under the dissecting microscope by removing any normal and connective tissues. Then, tumors were transferred to a dry dish and minced into pieces with blades. The tissue was digested in Leibovitz's medium (Invitrogen) with 2 mg/mL Collagenase Type I (Worthington), 2 mg/mL Elastase (Worthington), and 2 mg/mL DNase I (Worthington) at 37 °C for 45 min. The tissue was triturated with a pipette every 15 minutes of digestion until homogeneous. The digestion was stopped with FBS (Invitrogen) to a final concentration of 20%. The cells were filtered with a 70 μm cell strainer (Falcon) and spun down at $4500 \times g$ for 1 min. The cell pellet was resuspended in red blood cell lysing buffer (Sigma) for 3 min, spun down at 5000 r/min for 1 min, and washed with 1 mL ice-cold Leibovitz's medium with 10% FBS. Cells were resuspended in 1 mL ice-cold Leibovitz's medium with 10% FBS and filtered with a cell strainer (20 μm). Dead cells were removed with a Dead Cell Removal Kit (Miltenyi Biotec) according to the manufacturer's instructions. Live ©cells were collected for 10X Genomics library preparation.

## Flow cytometry

Tumors from syngeneic models were harvested and processed into single-cell suspensions for flow cytometry analysis. Tumors were chopped using a blade and then placed into a solution containing collagenase A /DNase I (Sigma). The tissue suspension was incubated at 37 °C for 30 minutes to allow enzymatic digestion. After incubation, the cell suspension was passed through a 70 μm cell strainer (Falcon). The cells were then washed twice with PBS. Following the initial wash, the suspension was filtered through a FACS tube strainer (Falcon). The cells were washed twice more with FACS buffer (PBS with 0.5% BSA and 2 mM EDTA). The following antibodies were used for staining: PE-conjugated anti-mouse CD45 (Biolegend, dilution 1:100), Pacific Blue-conjugated anti-mouse CD4 (Biolegend, dilution 1:100), FITC-conjugated anti-mouse CD3 (Biolegend, dilution 1:50), and APC-conjugated anti-mouse CD8 (Biolegend, dilution 1:50). Cells were incubated with the antibodies for 30 minutes at 4 °C in the dark. Following incubation, cells were washed twice with FACS buffer and resuspended for acquisition. Flow cytometry was performed using an Attune flow cytometer, and data were analyzed using Flow Jo software.

## Bulk RNA sequencing

Total RNA was extracted using the RNeasy Mini Kit (Qiagen) according to the manufacturer's protocol. RNA integrity was confirmed prior to library construction. Poly(A)-enriched mRNA libraries were prepared using Novogene's standard workflow for eukaryotic mRNA (WOBI) and sequenced on an Illumina NovaSeq X Plus platform to generate paired-end 150 bp reads, yielding ~6 Gb of raw data per sample. Raw sequencing reads were processed for quality control, including removal of adapter sequences and low-quality reads. Clean reads were aligned to the mouse reference genome (GRCm38/mm10) using the STAR aligner, and gene-level quantification was performed using featureCounts. Differential gene expression was conducted using DESeq2. Genes with an adjusted $p$ value < 0.05 and $|\log_2$ fold change| > 1 were considered significantly differentially expressed.

## scRNA-seq library prep

Single-cell Gene Expression Library was prepared according to the guidelines for the Chromium Single Cell Gene Expression 3v3.1 kit (10X Genomics). Briefly, single cells, reverse transcription[67] reagents, Gel Beads containing barcoded oligonucleotides, and oil were loaded on a Chromium controller (10´ Genomics) to generate single-cell GEMS (Gel Beads-In-Emulsions), where full-length cDNA was synthesized and barcoded for each single cell. Subsequently, the GEMS were broken and cDNAs from each single cell were pooled, followed by cleanup

using Dynabeads MyOne Silane Beads and cDNA amplification by PCR. The amplified product was then fragmented to an optimal size before end-repair, A-tailing, and adaptor ligation. The final library was generated by amplification. The library was performed at the Single Cell Genomics Core at BCM.

## scRNA-seq - raw data processing, clustering, and annotation

The Cell Ranger was used for demultiplexing, barcoded processing, and gene counting. The R package Seurat[68] and the Python package Scanpy[69] were used for pre-processing and clustering of scRNA-seq data. UMAP was used for dimensional reduction, and cells were clustered in Seurat or Scanpy. Each cluster was annotated based on marker gene information (see Supplementary Data S3 and S4, the list of marker genes of each cell cluster). Datasets were pre-processed, normalized separately, and annotated based on their marker gene expression. Scanpy was used for human dataset preprocessing and integration. Each dataset was normalized separately and clustered by the "Leiden" algorithm[70]. Scanpy was used to concatenate the *Cracd* WT vs. KO dataset and preSC *Cracd* WT vs. KO samples. Cells with less than 100 genes expressed and more than 20% mitochondrial reads were removed. Genes expressed in less than 20 cells were removed. Gene expression for each cell was normalized and log-transformed. The percentages of mitochondrial reads were regressed before scaling the data. Dimensionality reduction and Leiden clustering (resolution 0.5 - 1) were carried out, and cell lineages were annotated based on algorithmically defined marker gene expression for each cluster (sc.tl.rank_genes_groups, method = 'wilcoxon'). The list of differentially expressed genes (DEGs) in *CRPR2 and preSC Cracd KO* was generated by comparing KO vs. WT (sc.tl.rank_genes_groups, groups = ['KO'], reference = 'WT', method = 'wilcoxon'). More information about the software and algorithms used in this study is shown in Supplementary Data S5.

## Cell lineage trajectory analysis

RNA velocity[71] was used to predict the future state of individual cells and cell lineage tracing. Cells were filtered, and dimensional reduction was performed following the default parameters using the scVelo and Scanpy packages. RNA velocity was calculated through the dynamical model and the negbin model, and cells were clustered using the "Leiden" algorithm. RNA velocity for all datasets was performed with the same parameters (n_neighbors = 10, n_pcs = 40). Velocity streams were analyzed and plotted using scVelo (dynamical model)[27] and Dynamo (negbin model)[28]. Velocity pseudotime analysis was done and plotted with the scVelo package[27] to show the cell state (differentiation vs. de-differentiation) of each cell. PAGA[72] analysis was performed and visualized with the scVelo package to predict developmental trajectories and explore the connectivity between different cell clusters.

## Proportion difference analysis

Differences in clusters from the two datasets were analyzed and plotted using the pandas package[73]. Each cell cluster from the integrated dataset was grouped, and cluster differences between the two datasets were compared.

## Cell plastic potential analysis

The cell plastic potential was computed following the protocol outlined by Qin et al.[34].

**Single-cell entropies.** Single-cell entropy was determined using the SCENT tool (v1.0.3). The scRNA-seq data, which had been normalized and logarithmized, were initially processed using Scanpy and subsequently converted into a Seurat object (v4.4.0)[68]. Mouse gene symbols were mapped to human Entrez Gene identifiers utilizing the Orthology.eg.db (v3.17.0) and org.Mm.eg.db (v3.17.0) databases. The single-cell entropy was then calculated using the CCAT (Correlation of

Connectome and Transcriptome) algorithm (CompCCAT(), ppiA=net17Jan16.m).

**RNA velocity lengths.** RNA velocity lengths for single cells were extracted from scVelo's dynamical modeling as previously described.

**Single-cell PHATE coordinates.** The PHATE embedding for single cells was generated using the PHATE Python package (v1.0.11)[74]. The normalized and logarithmized scRNA-seq data were input into the PHATE operator (phate_operator.fit_transform(adata.raw.X)), and the resulting PHATE coordinates were exported.

**Valley-Ridge (VR) scores.** The VR score was calculated as a weighted sum of two components: Valley and Ridge, with weights of 0.9 and 0.1, respectively. This computation was performed on a per sample and per cluster basis. The Valley component was defined as the median CCAT value for each sample-cluster combination. To compute the Ridge component, the inverse of the RNA velocity length was calculated and then scaled between 0 and 1. The cell centrality distance within each cluster was determined using the single-cell PHATE coordinates, with the Python function compute_distdeg() as defined by Qin et al.[34]. The knn parameter was optimized according to the size of each cluster. The Ridge component for each sample-cluster was then computed as the product of the median scaled inverse velocities and the scaled cell centrality distances.

**Waddington-like landscapes.** The Waddington-like landscapes were visualized using Houdini Indie (SideFX, v20.0.533). In these visualizations, the VR scores were plotted along the y-axis, while the single-cell PHATE coordinates were positioned on the xz plane.

## Spatial transcriptomics

For the Xenium In Situ experiment, a single FFPE block was prepared from RPR2 and CRPR2 samples and placed onto a Xenium slide. Alongside the 379 Mouse Tissue Atlassing gene panel, additional 100 genes were incorporated for further analysis. Raw data were processed using Xenium Explorer v3.0.0 for image analysis. Cell segmentation was performed using nuclear expansion algorithms implemented in the Xenium platform. Cells were annotated based on graph-based clustering in Xenium Explorer using cell-type marker genes. Gene expression was visualized by point and density maps overlaid on images of nuclei and cells. Transcript counts and metadata were stored within each segmented cell for subsequent analysis. To compare normalized gene expression, datasets from RPR2 and CRPR2 were converted into Xenium objects developed using the Seurat package. Spearman rank-based correlation analysis was performed between Xenium and scRNA-seq expression profiles, and only pairs with correlation coefficients greater than 0.5 were retained. For each Xenium cell, the corresponding scRNA-seq cell information from the highest-correlating pair was assigned. For cell heterogeneity analysis, we observed the enrichment pattern of clusters, which were determined based on graph-based clustering by Xenium Explorer, in each tumor cell subclone. The tumor cell subclones were defined based on their location displayed by Xenium Explorer.

## Gene set enrichment analysis (GSEA)

GSEA was done using the R package "fgsea"[75] based on the DEG list generated by Scanpy. The enrichment value was calculated and plotted with the fgsea package (permutation number = 2000). Multiple testing correction was applied using the Benjamini–Hochberg method to control the false discovery rate (FDR).

## Cell-cell communication analysis

For ligand-receptor interaction-based cell-cell communication analysis of scRNA-seq datasets, the 'CellChat'[47] package in R was used. The integrated dataset was processed using the Seurat package, then the clustered and annotated datasets were analyzed by CellChat with default parameters ($p$ value threshold = 0.05). Epithelial cells were used as a source group, and immune cells were used as target groups.

## Pathway score analysis

Scanpy with the 'scanpy.tl.score_genes' function was used for the pathway score analysis[69]. The analysis was performed with default parameters and the reference genes from the gene ontology biological process or the Kyoto Encyclopedia of Genes and Genomes database[76,77]. The gene list for the score analysis is shown in Supplementary Data S6.

## Data downloading of TCGA-LUSC

The bulk RNA-seq data of TCGA-LUSC patients were prepared by TCGAbiolinks[78]. Briefly, the count matrix was downloaded by GDCquery (project = "TCGA-LUSC", data.category = "Transcriptome Profiling", data.type = "Gene Expression Quantification", workflow.type = "STAR - Counts", experimental.strategy = 'RNA-Seq', sample.type=c('Primary Tumor', 'Solid Tissue Normal'). The clinical data were downloaded by GDCquery_clinic(project = "TCGA-LUSC", type = "Clinical"). Based on the sample barcodes (01 for 'Solid Tissue Normal', 11 for 'Primary Solid Tumor') and the 'primary_diagnosis' (only 'Squamous cell carcinoma, NOS' was included) in the clinical data, the count matrix was sorted in the order of Normal and SCLC. The human gene Ensembl IDs inside of the count matrix were converted to HUGO gene symbols by biomaRt.

## CUT&RUN

**CUT&RUN assays.** CUTANA ChIC/CUT&RUN Kit (EpiCypher, Cat. No. 14-1048) was used. In brief, $5 \times 10^5$ cells (RPR2 and CRPR2 cell lines) were pelleted at 600 g for 3 minutes at room temperature[67]. After resuspending the cells twice with 100 μL of washing buffer (pre-wash buffer, protease inhibitors, and 0.5 mM spermidine), the cells were resuspended in wash buffer, preparing them for binding with beads. Next, 100 μL of the cell suspension was added to 10 μL of concanavalin A beads in 8-strip tubes, and the bead-cell slurry was incubated for 10 min at RT. After a brief spin-down, the tubes were placed on a magnet to quickly discard the remaining supernatant. The tubes were then removed from the magnet, and 50 μL of cold antibody buffer (cell permeabilization buffer with 2 mM EDTA) was immediately added to each reaction. The mixtures were pipetted to resuspend and confirm ConA bead binding. Next, 2 μL of each primary antibody (H3K27ac, H3K27me2, H3K27me3, and EZH2 from Cell Signaling) was added to the respective reactions. For the positive and negative control reactions, 1 μL of H3K4me3 positive control antibody and 1 μL of IgG negative control antibody (provided by EpiCypher) were added. Additionally, 2 μL of K-MetStat Panel was added to the reactions designated for the positive and negative control antibodies. The reactions were gently vortexed to mix and incubated overnight on a nutator at 4 °C. After overnight incubation, the tubes were briefly spun, placed on a magnet to allow the slurry to clear, and the supernatant was removed. While keeping the tubes on the magnet, 200 μL of cold cell permeabilization buffer (wash buffer with 0.01% digitonin) was added to each reaction. Next, 2.5 μL of pAG-MNase was added to each reaction, followed by gentle vortexing and a 10 min incubation at RT. The tubes were then quickly spun and placed on the magnet to clear the slurry, and the supernatant was removed. While keeping the tubes on the magnet, 200 μL of cold cell permeabilization buffer was added directly onto the beads, and the supernatant was removed. The tubes were then removed from the magnet, and 50 μL of cold cell permeabilization buffer was immediately added to each reaction, followed by gentle vortexing to mix and disperse clumps by pipetting. Subsequently, 1 μL of 100 mM calcium chloride was added to each reaction, and the tubes were incubated on a nutator for 2 h at 4 °C. At the end of

the 2-h incubation, the tubes were quickly spun to collect the liquid, and 34 μL of stop buffer was added to terminate pAG-MNase cleavage activity. The tubes were then placed in a thermocycler set to 37 °C for 10 min. Afterward, the tubes were placed on a magnet, and the supernatants containing CUT&RUN DNA were transferred to new 8-strip tubes. To purify the DNA, 119 μL of SPRIselect beads were slowly added to each reaction, followed by a 5 min incubation at RT. The tubes were then placed on a magnet for 2-5 min at RT, the supernatant was removed, and the beads were washed twice with 180 μL of 85% ethanol. After washing, the tubes were removed from the magnet, and the beads were air-dried for 2–3 min at RT. Finally, 17 μL of 0.1 × TE buffer was added to each reaction to elute the DNA.

**Library preparation and sequencing.** Library preparation for CUT&RUN was performed using the NEBNext Ultra II DNA Library Prep Kit for Illumina (M0544S), incorporating Illumina barcodes with 12 cycles of amplification. The libraries were sequenced on the Illumina NovaSeq platform at Novogene USA, with a read length of 150 base pairs for paired-end reads, and a sequencing depth of 30 million read pairs. The original sequencing data generated by the NovaSeq platform was converted into raw reads through base calling. These raw reads were stored in FASTQ format files.

**Analysis.** Alignment was performed using Bowtie2 (version 2.4.2). The SAM file was preprocessed using Picard (version 3.2.0), which included sorting, marking, and removing duplicates. The resulting file was then converted to BAM format using Samtools (version 1.3) and subsequently to a bedgraph file using Bedtools (version 2.31.1). Further analysis, including the calculation and visualization of each region, was conducted using deepTools (version 3.5.5) and Python (version 3.9.0).

### Human scRNA-seq data analysis
The scRNA-seq data set of 19 human SCLC patient samples (Patient information is shown in Supplementary Data S7)[10] from the Human Tumor Atlas Network (HTAN, https://humantumoratlas.org/) was downloaded and analyzed according to the code provided in the original study. The scRNA-seq data set of the 8 normal human lungs (GSE122960, Supplementary Data S8)[54] was extracted from the Gene Expression Omnibus (GEO) database and analyzed with Scanpy and Python. First, to match the gene names of our mouse CRPR2 dataset with those of human datasets, we converted mouse gene names into human gene names using the R package biomaRt, which converted 16,780 genes into human genes. The converted CRPR2 dataset and 27 human datasets were concatenated, normalized, and clustered in Scanpy. Batch effects were corrected using the "Harmony"[79] algorithm. Then, the dendrogram and correlation matrix heatmap were plotted with Scanpy. The dendrogram shows the distance of each dataset based on principal component analysis, and the correlation matrix heatmap shows Pearson correlation by a color spectrum.

### Copy number variation analysis
We performed copy number variations (CNVs) analysis from the gene expression data using the Python package infercnvpy (https://icbi-lab.github.io/infercnvpy/index.html#). We ran infercnvpy using the Normal group (8 human normal lung datasets) as a reference dataset. The gene ordering file containing the chromosomal start and end positions for each gene was generated from the human GRCh38 assembly. Chromosome heatmap and CNV scores in the UMAP were plotted with infercnvpy.

### Public sequencing database
All TCGA cancer patients' sequencing data referenced in this study were obtained from the TCGA database at cBioPortal Cancer Genomics (http://www.cbioportal.org). Cancer cell line sequencing data from Cancer Cell Line Encyclopedia (CCLE) were extracted from the cBio-Portal Cancer Genomics (http://www.cbioportal.org).

### Statistical analyses
GraphPad Prism 9.4 (Dogmatics) was used for statistical analyses. Student's *t*-test was used to compare two samples. *P* values < 0.05 were considered statistically significant. Error bars indicate the standard deviation (s.d.), otherwise described in the Figure legends. All sample numbers were estimated by power calculation based on preliminary data and previous studies to ensure sufficient statistical power to detect biologically relevant differences. Random allocation was not applicable to most cell line–based experiments. For in vivo studies, animals were randomly assigned to experimental groups, and both male and female mice were used to minimize bias. Blind testing was not performed since most data were quantifiable through statistical analyses.

### Reporting summary
Further information on research design is available in the Nature Portfolio Reporting Summary linked to this article.

## Data availability
The scRNA-seq data generated in this study have been deposited in the GEO database under accession code GSE218544. The CUT&RUN-seq data generated in this study have been deposited in the GEO database under accession code GSE280263. The Xenium spatial transcriptomics data generated in this study have been deposited in the GEO database under accession code GSE299069. The processed data supporting the findings of this study are available within the paper and its Supplementary Information. Source data are provided with this paper.

## Code availability
The code used to reproduce the analyses described in this manuscript can be accessed via GitHub (https://github.com/jaeilparklab/CRACD_SCLC) and is archived on Zenodo under (https://doi.org/10.5281/zenodo.17382119).

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

## Acknowledgements

This work was supported by the Cancer Prevention and Research Institute of Texas (RP200315 to J.-I.P.), the National Cancer Institute (K99 CA286761 to K.-P.K., R35 CA263816 to C.M.R.; U01 CA224293 to K.-S.P.; R01 CA193297, R01 CA278971, R03 CA279867, and R03 CA256207 to J.-I.P.; R01 CA278967 to J.-I.P. and K.-S.P.; R01 CA262324 to M.G.L.), and the Lung TRT Award from the University of Virginia Cancer Center (to K.-S.P). The core facilities at MD Anderson (DNA Sequencing and Genetically Engineered Mouse Facility) were supported by the National Cancer Institute Cancer Center Support Grant (P30 CA016672). The Research Histology Core (RRID:SCR_025470) at the University of Virginia was supported by the National Cancer Institute Cancer Center Support Grant (P30 CA044579). The core facilities at Baylor College of Medicine (Cytometry & Cell Sorting Core and Single Cell Genomics Core) were supported by CPRIT (RP180672, RP200504) and the National Institutes of Health (CA125123, RR024574). Graphical illustrations were created with BioRender.com.

## Author contributions

Y.S.: Methodology, investigation, software, analysis, data curation, writing (original draft), visualization; S.Z.: Methodology, investigation, software, analysis, data curation, writing (original draft), visualization; J.J.: Methodology, investigation, software, analysis, data curation, writing (original draft), visualization; K.-P.K.: Methodology, investigation, software, analysis, data curation, writing (original draft), visualization; K.B.K.: Methodology, analysis, investigation, data curation, writing (original draft); Y.H.: Investigation, software, analysis, writing (original draft); D.W.K.: Investigation, analysis; B.K.: Investigation; G.Z.: Investigation; J.Z.: Investigation; S.J.: Investigation; W.C.: Investigation; N.A.K.: Investigation; Y.E.H.: Investigation; Y.H.B.: Investigation; S.S.D.: Methodology; J.M.C.: Data curation; M.G.L.: Resources, methodology, analysis; C.M.R.: Resources, analysis, writing (review and editing); K.-S.P.: Conceptualization, methodology, analysis, writing (original draft, review, and editing), visualization, supervision, project administration, funding acquisition; J.-I.P.: Conceptualization, methodology, analysis, writing (original draft, review, and editing), visualization, supervision, project administration, funding acquisition.

## Competing interests

All authors declare no competing interests.
