## [Transparent Peer Review file · Nature Communications]

Actin Dysregulation Induces Neuroendocrine Plasticity and Immune Evasion: A Vulnerability of Small Cell Lung Cancer

Corresponding Author: Dr Jae-Il Park

Version 0:

Reviewer comments:

Reviewer #1

(Remarks to the Author)

This study investigates the role of CRACD as a tumor suppressor in small cell lung cancer (SCLC), emphasizing its impact on neuroendocrine plasticity, immune evasion, and the potential therapeutic targeting of EZH2. The findings suggest that CRACD depletion facilitates immune evasion through reduction of nuclear actin, EZH2-mediated histone methylation, suppression of MHC-I Genes and depletion of CD8+ T Cells. The downregulation of MHC-I results in a significant decrease in intratumoral CD8+ T cells, impairing immune surveillance and promoting immune evasion.

Major Comments

Experimental Design and Data Presentation

- **Figure Repetitions & Sample Size:** The manuscript lacks clarity regarding the number of in vivo and in vitro experimental repetitions, as well as the number of mice per experiment. Are the data shown representative or cumulative from multiple experiments? Explicit details on experimental replicates should be provided.
- **Figure 1d – Use of Nude Mice:** The rationale for using Nude mice needs to be clarified. Did the authors intend to exclude tumor-adaptive immune interactions? A repetition of the experiment in syngeneic wild-type (WT) mice would be more appropriate to establish the role of the adaptive immune system in CRACD depletion from the outset.
- **Excess Figures:** The number of figures is excessive. Consider moving the transplantation model from Figure 1 to the supplementary section and merging Figures 2 and 3 for a more streamlined presentation.

Spatial and Single-Cell Analysis:

- **Figure 4 – Choice of Spatial Transcriptomics (ST) over scRNA-seq:** The authors utilized Xenium for tumor heterogeneity analysis, despite the availability of single-cell RNA sequencing (scRNA-seq) data (Figure 3). Instead of treating scRNA-seq for plasticity and ST for heterogeneity separately, they should compare the two in terms of plasticity and heterogeneity.
- **Discrepancies in HES1 Expression:** A key inconsistency arises between Figure 4j and Supplementary Figure S4a, where CRACD depletion appears to inhibit HES1, while Figure 3d suggests that CRPR2 tumors express more HES1 than RPR2 tumors. How do the authors reconcile this contradiction?
- **Spatial Organization of Tumor Microenvironment:** The spatial analysis does not explore tumor cell organization. Given that Figure 5 suggests tumor escape from CD8+ T cells via MHC-I downregulation, it would be highly relevant to analyze the spatial distribution of T cell infiltrates in CRPR2 versus RPR2 tumors. Specifically: Do CRPR2 tumors generate lymphoid aggregates? How do T cells localize relative to cancer cells? Are CD8+ T cells excluded from tumor neighborhoods?

Immune Response and Therapeutic Implications

- **Figures 6o–p – T Cell Data Presentation:** The current approach of presenting T cell infiltration as a percentage of CD45+ cells may obscure effects on CD4+ T cells. T cell infiltration should instead be quantified as the absolute number of T cells per tumor weight or per cancer cell. Reanalysis with these metrics is likely to reveal additional insights into CD4+ T cell responses.
- **EZH2 Inhibitor Effects on T Cells:** The study would benefit from an expanded FACS analysis of T cell phenotypes following EZH2 inhibitor treatment, including expression of activation markers and intracellular cytokines. Potential synergistic effects between EZH2 inhibitors and immune checkpoint inhibitors (ICIs) would be clinically relevant and should be investigated.

• CD8 T Cell Depletion Experiment: To substantiate the connection between CD8+ T cells and CRPR2 tumor progression, the authors should conduct a CD8 depletion experiment and analyze its effect on tumor growth. This is particularly critical given that CRACD depletion has intrinsic tumor-promoting effects. If CRPR2 and RPR2 tumors show minimal differences in the absence of CD8+ T cells, it would further support the hypothesis that immune evasion drives CRPR2 tumor outgrowth.

Conclusion

The study presents compelling findings on CRACD's role in immune evasion and the therapeutic potential of EZH2 inhibition in SCLC. However, several key points require clarification, including experimental design details, spatial organization of the tumor microenvironment, and the immune response to EZH2 inhibition. Addressing these points will strengthen the manuscript's conclusions and provide a more comprehensive understanding of CRACD's function in SCLC.

Tsoumakidou Maria, MD PhD
Group Leader - Institute of Bioinnovation
BSRC "Alexander Fleming"
Associate Professor of Physiology
Medical School, National and Kapodistrian University of Athens

Reviewer #2

(Remarks to the Author)

This manuscript by Seo et al. presents a significant advancement in our understanding of small-cell lung cancer (SCLC) by elucidating the role of CRACD loss in promoting neuroendocrine (NE) plasticity and immune evasion. The study employs rigorous methodologies, including genetically engineered mouse models, single-cell RNA sequencing (scRNA-seq), spatial transcriptomics, and functional validation experiments, to support its claims. The proposed CRACD-EZH2-MHC-I axis as a potential therapeutic target is compelling and has strong translational implications. However, there are critical areas that require further clarification, methodological refinement, and additional data to enhance the manuscript's impact and scientific rigor.

This study reveals a previously underappreciated role of CRACD in actin polymerization, tumor cell plasticity, and immune evasion. It establishes that CRACD loss drives neuroendocrine plasticity by suppressing NOTCH signaling, increasing tumor heterogeneity, and facilitating immune escape via EZH2-mediated MHC-I suppression. The findings reported that EZH2 inhibition is a novel therapeutic strategy for CRACD-deficient SCLC. By integrating GEMMs, CRISPR-Cas9 knockout models, scRNA-seq, and spatial transcriptomics, the study provides a high-resolution, multi-dimensional analysis of tumor plasticity and heterogeneity. Functional assays further confirm the efficacy of EZH2 inhibitors in restoring MHC-I expression and reversing immune evasion. Notably, the study identifies a distinct SCLC molecular subtype characterized by CRACD loss, EZH2-driven transcriptional repression, and MHC-I downregulation, highlighting a potential biomarker and supporting the therapeutic relevance of EZH2 inhibition in cold SCLC tumors.

Major comments:

1. The study establishes a connection between CRACD loss and NOTCH signaling downregulation; however, the precise molecular mechanism remains unresolved. Does CRACD directly interact with NOTCH pathway components, or is this effect mediated through actin cytoskeletal disruption?
2. While MHC-I suppression is attributed to EZH2-mediated histone methylation, the potential involvement of additional epigenetic regulators (e.g., HDACs, DNMTs) remains unexplored. Chromatin accessibility assays (e.g., ATAC-seq) could provide further mechanistic clarity.
3. The interplay between NOTCH signaling downregulation and EZH2-driven immune evasion is briefly mentioned but not thoroughly investigated. Could NOTCH signaling directly regulate EZH2 recruitment?
4. The observed depletion of intratumoral CD8+ T cells in CRACD-deficient tumors may be influenced by additional immunosuppressive populations, such as myeloid-derived suppressor cells (MDSCs) and regulatory T cells (Tregs). A more comprehensive immune profiling using flow cytometry or single-cell analysis is required.
5. While the study suggests that restoring MHC-I expression counteracts immune evasion, it does not confirm whether this results in functional CD8+ T cell activation. In vivo, cytotoxicity assays and cytokine profiling would strengthen this conclusion.
6. EZH2 inhibition is shown to suppress tumor growth, but its effect on overall survival remains unexamined. Kaplan-Meier survival curves should be included to assess therapeutic impact.
7. Given SCLC's heterogeneity, other epigenetic vulnerabilities in CRACD-low tumors may exist. Identifying potential co-targets with EZH2 could refine therapeutic strategies.
8. The scRNA-seq findings are compelling, but additional validation at the protein level is essential. Western blot or immunofluorescence for key markers identified in human SCLC samples would strengthen the conclusions.
9. Certain statistical comparisons require greater transparency regarding methodology. The manuscript should clarify whether multiple testing corrections were applied in pathway analyses.
10. The study does not compare EZH2 inhibition with current immunotherapies, such as PD-1/PD-L1 blockade. Could combination therapy enhance therapeutic efficacy? Does it provide an added advantage as combination therapy or an alternative to existing therapies?

Minor comments:

1. To enhance translational impact, the graphical abstract should more explicitly emphasize the therapeutic relevance of the CRACD-EZH2-MHC-I axis.
2. The discussion would benefit from a broader integration of findings from similar studies on tumor plasticity in other

malignancies, such as prostate and pancreatic neuroendocrine tumors.

3. Additional references on EZH2 inhibitors in SCLC and other cancers would provide a more comprehensive contextualization of the therapeutic approach.

4. Certain sections, particularly within the results, contain dense and complex sentences that could be streamlined for clarity and conciseness.

5. The term "immune evasion" is used broadly throughout the manuscript. Distinguishing between specific mechanisms, such as T cell exclusion versus antigen presentation loss, would improve precision in interpretation.

In summary, this study provides a compelling contribution to SCLC research by identifying CRACD as a key tumor suppressor that constrains NE plasticity and immune evasion. The mechanistic insights into actin cytoskeletal regulation, NOTCH signaling suppression, and EZH2-driven epigenetic remodeling are highly valuable. However, the study lacks a clear mechanistic link between CRACD, NOTCH, and EZH2. Additionally, expanding immune profiling to better characterize the tumor microenvironment, functionally validating the impact of EZH2 inhibition on CD8+ T cell activation and cytotoxicity, and assessing survival outcomes while exploring combinatorial therapeutic strategies will further strengthen the findings. Addressing these critical points will enhance the manuscript's scientific rigor.

Reviewer #3

(Remarks to the Author)

This manuscript, written by Seo et al., describes the role of the actin de-capping inhibitor, CRACD, in small cell lung cancer (SCLC). CRACD promotes the actin polymerization, and is thereby involved in the plasticity and immune reactivity of SCLC. The authors conducted the single cell RNA sequencing (scRNA seq) analysis and the spatial analysis (Xenium in situ hybridization) for the mouse model of SCLC, in which RB1, TP52 and RBL2 were knocked out (RPR2 KO mice). Using this mouse model, they found that the CRCD transforms the pre-neoplastic cells into SCLC tumor-like cells, including the compaction of the cells. CRACD also induces neuroendocrine (NE) plasticity and increases the tumor cell heterogeneity via dysregulation of NOTCH1 pathway. Even more importantly, they observed that CARCS induced the EZH2-mediated H3K27 tri-methylation by the nuclear actin reduction. This epigenomic change suppresses the expression of the MHC-I genes. As a result, the recruitment of cytotoxic T cells (CTLs) are reduced. As an attempt to utilize these findings for developing a new drug treatment, the authors treated the CRCD KO-RPR2 mice with an EZH2 inhibitor. They found that the MHC-I expression and the immune surveillance were actually restored. Even though further analyses directly using human material may be needed, this study paves the first step towards the combination therapy of the ICB and EZH2 inhibitor to this difficult cancer species.

Overall, I think this is the well-written paper, supported by the solid and a wide variety of experimental evidence. The proposed perspective to utilize the EZH2 is timely and the one which a large number of the patients having SCLC are looking forward to.

Major points:

1 While I fully understand that the scope of this particular paper should lie in the study of the pre-clinical mouse model, I'd still like to request the authors to conduct the extensive analysis to demonstrate the relevance of the obtained results directly using the human material, at least, to some extent. The presented data is, after all, from the mouse model and does not always indicate the in vivo relevance. In fact, the molecular mechanisms underlying the lung cell lineage and the cancer development are supported to occasionally quite different from humans, even though superficial appearance may look similar.

2 Results of the Xenium analysis is not fully associated with those of the scRNA analysis. The spatial locations of the respective single cells should be presented. Also, I assume the data from the 379 gene Mouse Tissue Atlas Panel plus 100 gene custom should not be so much comprehensive. In addition to showing just the cellular composition or the tumor heterogeneity, please show which parts of the single cells profiles are directly represented in the Xenium data.

3 To me, the highlight of this paper is the use of an EZH2 inhibitor. On the other hand, I think the current EZH2 inhibitors should invoke substantial effects on the epigenome statuses of indirect target sites (as a secondary effect) or on indirect target genes other than EZH2. In addition to the MHC-I genes, the sites where the relevant changes were observed should be further scrutinized. Also, I wonder if some key cancer gene regulations may be also influenced?

Minor points:

4 I'm not totally sure how the obtained results should be interpreted in the context of the previous molecular subtypes (ASCL1, NeuroD1, POU2F3 and YAP1) and the neuroendocrine (NE) or non-neuroendocrine subtypes. Sometimes, those particular subtypes are associated with inflammatory subtypes, to which immune check point blockage (ICB) should be effective. However, as mentioned by the authors themselves, these classifications are not always consistent with clinical phenotypes (or drug responses) of the patients. Honestly, I'm not familiar with this mouse model. In fact, during the review, I firstly found a quite frequently employed model for this purpose. Therefore, I wonder what subtype this mouse is modeling (also related to the point 1). Please carefully discuss the limit of the present analysis.

5 Relatedly, in human cancers, the RBL2 mutation does not always co-occur with the CRACD mutation. I wonder how the neoplastic phenotype which appears in the mouse model without the RBL2 mutation should be further connected to the results obtained from the RPR2 mouse model. For this purpose, I'd like to see the spatial analysis of the preSC mice. At least, please include the comparison analysis on the spatial characteristics between the RPR2 and CRPR2 mice.

6 Innocently, I also wonder, in this mouse model, where in the lung and what cell type the preSC cells and LCSC cells are

originating from.

7 The interpretation of the Notch-Delta signaling status is not clear to me in the context of the present paper. This axis is pivotal to the transition from NE to non-NE subtypes, as often referenced (even though it may not be the case of SCLC; Yan Ting Shu et al, Nat Com 2022, for example). There, REST is also involved, which might be also involved with the EZH2 inhibition. In the single cell data, I wonder if the whole picture of the regulatory network may be represented for the possible feed-back or cross-talk of the pathways (which may be also, as least to some extent, represented by the Xenium data).

8 Discussion should include the recent papers which also describe the single cell features of SCLC (Leslie Duplaquet et al, Nature Cell Biology 2023 and Abbie S. Ireland et al, Cancer Cell 2022). Those papers may be also useful to give a more comprehensive view for what is occurring in the SCLC cells and how the epigenome perturbation may make effects (either beneficial or adverse). Also, possible involvement of Myc could be also discussed.

9 Please show the list of the genes used for the custom 100 probes for the Xenium analysis. I also wonder how the timing of the sacrifice was determined. The time course sampling should be also useful to validate the robustness of the analysis.

10 Please make sure that the Xenium data is also made publicly available in GEO. Also, for all the codes for the data process in GitHub, especially for the parts of the relatively new analyses, including the VR analysis, which I found intuitively very much helpful. (I have not visited the indicated URLs for those datasets using the reviewer token.)

Reviewer #4

(Remarks to the Author)

Seo et al. examined the impact of CRACD inactivation on small cell lung cancer (SCLC) tumor development, plasticity and immune evasion. The Authors examined Cracd/CRACD in SCLC in vitro and in vivo model systems, human primary SCLC samples with experimental and computational approaches and found that loss of Cracd/CRACD disrupted cytoplasmic and nuclear actin organization leading to dysregulation of NOTCH1 signaling and EZH2-mediated suppression of MHC-I pathways. Alterations to these processes in turn resulted in increased plasticity and immune evasion. The Authors' demonstration of cellular plasticity using single cell and spatial genomics was state-of-the-art and also contributed to their hypothesis. Functional validation of computational results using animal and in vitro models of SCLC was also comprehensive. The study also suggested EZH2 inhibition as a potential strategy to improve response to immune checkpoint blockade (ICB) therapy in SCLC. Overall, this is a very comprehensive and well-executed study with high translational relevance. To strengthen the study further, the authors should consider the following questions/issues:

Major issues:

1. Given germline constitutive knockout of Cracd in the qKO model, is it possible that the differences in the tumor immune microenvironment and even in the cancer cells themselves in the tKO vs. qKO models also stemmed from Cracd loss in the stromal, immune and other non-cancer cells in the mouse? Has the impact of Cracd loss in various immune and other stromal compartments such as T cells been documented in vitro and in vivo models in the Cracd KO control mouse?
2. The authors argued that CRACD inactivation, via actin dysregulation, inhibited NOTCH1 signaling, which in turn promoted neuroendocrine (NE) plasticity. Why would this phenotype not occur in the non-NE population with activated NOTCH1 signaling in the qKO tumors? In Figure 7j, the authors claimed that increased non-NE features in qKO tumors may occur via EMT activation, although correlations between the two were only observed. The authors should qualify these claims more or provide additional evidence of the CRACD inactivation-EMT-non-NE axis.
3. What correlations did the authors observe upon analysis of mouse tumor samples using the different methods (scRNA-seq, spatial transcriptomics), in terms of (1) SCLC plasticity/lineage identity and (2) the profile of immune cells within the microenvironment? In other words, are the differences in cellular composition observed with scRNA-seq reproduced in the spatial transcriptomic dataset? A deeper and full analysis of these combined datasets may be the focus of a separate paper altogether, but some superficial-level correlations is warranted for additional rigor in the current study.
4. Any difference in survival between the qKO vs tKO GEMMs? Given the changes in plasticity, did the authors observe differences in metastasis with qKO vs tKO GEMMs?

Minor issues:

1. Italicize in vitro and in vivo
2. There was no mention of Fig. S2 in the main text.
3. Provide in the main text brief descriptions of computational terms and analytical methods that are heavily referenced such as 'root cell', 'scVelo', 'Dynamo', 'entropy', and 'CellChat'.
4. Line 74: Potential mechanisms underlying SCLC plasticity have been explored in a number of studies such as Ireland et al. (2020). The authors should cite additional relevant literature here.
5. Given the central nature of NOTCH1 signaling and MHC-I downregulation to the mechanism of their current study in SCLC, the authors should provide a brief background in the INTRODUCTION.
6. Can the authors comment on the location of mutations in CRACD that are found in primary tumors? Do these mutations, especially those occurring at a high frequency, reside in functional domains of the protein?
7. There are two panels in Fig. S1 labeled with 'D'
8. Line 99: please cite the specific dataset on TCGA the authors used to extract transcriptomic bulk RNA-seq data for healthy tissues and SCLC tumors.
9. The authors should label the timepoint at which the lungs were collected & compared.
10. How did the authors select genes for the analysis of the different signaling pathways (NOTCH, Myc, EMT, etc.)? Why are

there only 2 genes for the EMT program?

11. The figure legend for Figure. 3d refers to violin plots but there were none.

12. Figure 4D: Why were there so few tKO tumors (4) compared to qKO tumors (36) in the spatial transcriptomics study? Could this discrepancy have an impact on the results?

13. What is the mutation frequency in CRACD in MS1 vs. MS2 tumors?

14. Does the mutational status of CRACD correlate with response to immunotherapy in patients with SCLC?

15. Line 469. Authors mean "not with primary tumors".

16. Any functional assays to evaluate cell motility or migration affected by actin dysregulation? Authors should demonstrate the impact of Cracd KO on cell motility/migration to confirm its role in actin dysregulation in the context of SCLC cells.

Reviewer #5

(Remarks to the Author)

Version 1:

Reviewer comments:

Reviewer #1

(Remarks to the Author)

All reviewer concerns have been addressed. The authors have made substantial clarifications and improvements to the paper, added new experiments and analyses where appropriate, and provided additional explanations to strengthen the rigor and clarity of their work. I believe that these changes have resulted in a manuscript that is more robust and more convincingly supportive of their conclusions.

Reviewer #2

(Remarks to the Author)

The revised manuscript titled "Actin Dysregulation Induces Neuroendocrine Plasticity and Immune Evasion: A Vulnerability of Small Cell Lung Cancer" provides compelling insights into the role of CRACD as a tumor suppressor in SCLC. The study demonstrates that loss of CRACD drives neuroendocrine plasticity, tumor heterogeneity, and immune evasion through EZH2-mediated suppression of MHC-I, thereby revealing a mechanistically grounded vulnerability in this aggressive cancer type.

I find that the authors have thoroughly addressed all of my previous comments and concerns. The mechanistic connections have been clarified, the immune profiling has been expanded, and additional functional validations, as well as methodological clarifications, have been included.

Reviewer #3

(Remarks to the Author)

First of all, I appreciate the dedicated efforts of the authors made for this revision. Owing to the extensive analyses and the deepened discussion, I believe this manuscript has been very much improved. Particularly, I appreciate the newly generated results from the tissue array analyses of the human samples. I also enjoyed reading the deepened discussion on the EZH2 inhibition. Thanks to these, I think the mouse data should have become nearer to the clinical indications in a more practical manner. I also believe the future papers will produce the spatial/single cell analyses of clinical samples of SCLC or HGNEC in the near future. I sincerely hope the authors should continue their efforts integrate them with their mouse model and further polish the deduced hypothesis in this study. I believe such an integration analysis should bring the patients of this difficult cancer type a better therapy on a yet novel concept.

Reviewer #4

(Remarks to the Author)

The revisions address my major concerns, including clarification of replicates, integration of scRNA-seq with spatial data, resolution of the HES1 issue, improved T cell analyses, and addition of a CD8 depletion experiment. These strengthen the manuscript considerably, and I now find it suitable for publication.

Responses to Comments

Reviewer 1

“This study investigates the role of CRACD as a tumor suppressor in small cell lung cancer (SCLC), emphasizing its impact on neuroendocrine plasticity, immune evasion, and the potential therapeutic targeting of EZH2. The findings suggest that CRACD depletion facilitates immune evasion through reduction of nuclear actin, EZH2-mediated histone methylation, suppression of MHC-I Genes and depletion of CD8+ T Cells. The downregulation of MHC-I results in a significant decrease in intratumoral CD8+ T cells, impairing immune surveillance and promoting immune evasion.”

Major points

Experimental Design and Data Presentation

A1. *“Figure Repetitions & Sample Size: The manuscript lacks clarity regarding the number of in vivo and in vitro experimental repetitions, as well as the number of mice per experiment. Are the data shown representative or cumulative from multiple experiments? Explicit details on experimental replicates should be provided.”*

Thank you for the comments. In the initially submitted manuscript, n numbers were included in the Figure legends or shown as individual data points in the plots. As suggested, we clarified “n” numbers in the Figure legends (highlighted in the revised manuscript).

A2. *“Figure 1d – Use of Nude Mice: The rationale for using Nude mice needs to be clarified. Did the authors intend to exclude tumor-adaptive immune interactions? A repetition of the experiment in syngeneic wild-type (WT) mice would be more appropriate to establish the role of the adaptive immune system in CRACD depletion from the outset.”*

Yes, we initially sought to determine the cell-autonomous impact of *Cracd* KO. For a distinct study, we employed both immunocompromised and immunocompetent murine models and consistently observed that *Cracd*-depleted preSCs exhibited enhanced tumorigenesis in both Nude and C57BL/7 mice (**Fig. R10**). These results are the subject of another manuscript nearing completion and, therefore, cannot be incorporated into the present revised manuscript. We maintain, however, that the data derived from our GEMMs or tumor cells (RPR2 vs. CRPR2) (**Fig. 1**) adequately support our proposed working model, offering stronger evidence than the preSC cell data. In accordance with prior suggestions, Figure 1 has been relocated to Supplementary Figure 2.

A3. *“Excess Figures: The number of figures is excessive. Consider moving the transplantation model from Figure 1 to the supplementary section and merging Figures 2 and 3 for a more streamlined presentation.”*

Thank you for the suggestion. This manuscript covers two distinct aspects of SCLC tumorigenesis: NE cell plasticity and immune evasion. We believe that Figure 1 is essential to show the impact of CRACD loss-of-function on SCLC tumor initiation and NE plasticity using preSC cells, which we previously generated. Such an early event cannot be addressed by using GEMMs. As suggested, we merged Figure 2 with Figure 3.

Immune Response and Therapeutic Implications

A4. *“Figure 4 – Choice of Spatial Transcriptomics (ST) over scRNA-seq: The authors utilized Xenium for tumor heterogeneity analysis, despite the availability of single-cell RNA sequencing (scRNA-seq) data (Figure 3). Instead of treating scRNA-seq for plasticity and ST for heterogeneity separately, they should compare the two in terms of plasticity and heterogeneity.”*

Figure R10. Tumor growth of *Cracd* KO preSC cells in immunocompromised or immunocompetent mice. PreSC cells (*Cracd* WT vs. KO; derived from C57BL/6 mice) were subcutaneously injected into Nude or C57BL/6 mice. Tumor volumes were measured at the endpoint when any dimension of allograft tumor reaches 1.5 cm or bigger. Tumor growth is defined by tumor volume/days taken to reach the endpoint and is plotted relative to the *Cracd* WT preSCs allograft in nude mice. n=10 per group, Student's *t*-test.

scRNA-seq for analyzing tumor heterogeneity. We respectfully disagree that *scRNA-seq* can assess “intratumoral” heterogeneity. While scRNA-seq, by itself, lacks spatial context within lung tissues and therefore cannot directly quantify intratumoral spatial heterogeneity, it remains a valuable method for characterizing overall “*cellular*” heterogeneity through gene expression-based multi-dimensional visualization and cell clustering. Although Visium and Visium HD offer spatial information, their utility is limited by their sub-optimal cell segmentation, which does not achieve single-cell resolution. Importantly, our study pioneers the calculation of tumor heterogeneity by leveraging molecular-level spatial transcriptomics.

Xenium In Situ for assessing cell plasticity. While Xenium In Situ, based on nucleic acid hybridization, holds theoretical promise for assessing cell plasticity—a capability well-established through scRNA-seq—its current application for the intricate analyses presented here, such as the calculation and visualization of cell plasticity potential (Fig. 3), remains limited. Although the field is actively exploring the integration of single-cell and spatial transcriptomics, these methods have yet to reach the analytical sophistication demonstrated in our manuscript. Furthermore, our Xenium In Situ experiment predates the availability of the expanded 5,000 gene panel. Utilizing a custom 479-gene set, our analysis could not achieve the comprehensive gene coverage and resolution afforded by scRNA-seq, where we typically analyze over 5,000 genes for detailed cell lineage and plasticity studies. Consequently, a comparable cell plasticity analysis using Xenium In Situ could be limited.

Visualizing scRNA-seq information on Xenium. Cell lineages. As suggested, we projected cell clusters and cell proliferation information from scRNA-seq onto Xenium data. Consistent with the scRNA-seq results, we located new non-NE cell clusters (purple) generated in CRPR2 tumors at the core of SCLC tumor nodules. These non-NE cells were surrounded by NE cells (Fig. R11A, B), recapitulating NE cell plasticity of CRPR2 tumors. Additionally, CRPR2 tumor cells are more proliferative than RPR2 tumor cells. Moreover, NE cells are relatively more mitotic than non-NE cells (Fig. R11C). These new results were added to the revised manuscript (Fig. 2f-h).

Figure R11. Spatial localization and characterization of tumor cell subpopulations in RPR2 and CRPR2 tumors. **A.** UMAP projection of scRNA-seq data from RPR2 and CRPR2 tumors. Cells are colored by unsupervised clusters; NE and non-NE populations are delineated by dashed lines. **B.** Spatial localization of NE and non-NE clusters in representative RPR2 and CRPR2 tumors. Cluster identities were assigned by correlation-based mapping of Xenium transcriptomic data to matched scRNA-seq clusters. **C.** Spatial mapping of cell cycle states projected onto Xenium coordinates. Cell cycle phase (inferred from matched scRNA-seq data) are shown as: G1 (magenta), S (aqua), and G2/M (purple).

A5. “Discrepancies in HES1 Expression: A key inconsistency arises between Figure 4j and Supplementary Figure S4a, where CRACD depletion appears to inhibit HES1, while Figure 3d suggests that CRPR2 tumors express more HES1 than RPR2 tumors. How do the authors reconcile this contradiction?”

The apparent discrepancy in HES1 expression between **Supplementary Fig. S4** and **Supplementary Fig. S5** reflects the heterogeneous nature of CRACD-deficient SCLC, which is one of the central findings of our study. In line with this, we recently observed that *Cracd* KO also results in suppression of HES1 expression in LUAD GEMMs¹. Moreover, the CRPR2 tumors exhibit transcriptional heterogeneity, including the emergence of non-neuroendocrine subpopulations with elevated HES1 expression (**Supplementary Fig. S4d**).

This HES1-high non-NE population likely represents a subset of tumor cells undergoing lineage transition, accompanied by active or reactivated Notch signaling. These findings align with prior studies showing that Notch-HES1 activation suppresses neuroendocrine features and promotes a non-NE fate in SCLC¹. This

interpretation is further supported by **Figure 1m**, where CRPR2 tumors exhibit high-entropy, multilineage trajectories including a NOTCH^{high} non-NE cluster, consistent with the HES1^{high} subpopulations in CRPR2 tumors. Therefore, the presence of HES1-positive non-NE cells in CRPR2 tumors supports the notion that CRACD loss facilitates divergent cell fates and contributes to intratumoral heterogeneity. Importantly, such heterogeneity is not merely a molecular observation but is closely linked to tumor plasticity, therapeutic resistance, and disease progression. Thus, rather than indicating a contradiction, the variable HES1 expression across models and tumor subpopulations highlights the context-dependent and dynamic consequences of CRACD inactivation in SCLC.

A6. *“Spatial Organization of Tumor Microenvironment: The spatial analysis does not explore tumor cell organization. Given that Figure 5 suggests tumor escape from CD8+ T cells via MHC-I downregulation, it would be highly relevant to analyze the spatial distribution of T cell infiltrates in CRPR2 versus RPR2 tumors. Specifically: Do CRPR2 tumors generate lymphoid aggregates How do T cells localize relative to cancer cells? Are CD8+ T cells excluded from tumor neighborhoods?”*

Thank you for the comments regarding tumor infiltration, lymphoid aggregation, and CD8+ T cell exclusion. In the initial manuscript, we assessed the spatial distribution of T cell infiltration in CRPR2 and RPR2 tumors. Immunofluorescence staining for CD3 and CD8 showed that CD8+ and CD3+ T cell infiltration was markedly reduced in the tumor core and periphery of CRPR2 tumors compared to RPR2 tumors (**Supplementary Fig. S6e-h**). To address the comments regarding lymphoid aggregates, we repeated the experiments. Consistently, staining results showed the spatial exclusion of T cells in CRPR2 tumors. Notably, unlike RPR2 tumors that occasionally showed lymphoid aggregate-like structures, CRPR2 tumors did not exhibit substantial formation of tertiary lymphoid structures (**Fig. R12**), consistent with immunostaining and scRNA-seq results. New results and additional description were added (**Supplementary Fig. S6i**).

Figure R12. Reduced CD8+ T cell infiltration in CRPR2 tumors compared to RPR2 tumors **A.** Representative immunostaining images of tumor sections from RPR2 and CRPR2 tumors. Top panels show low-magnification views with red boxes indicating regions of interest. Bottom panels show higher-magnification images of the indicated ROIs. Panels labeled (a) and (b) show enlarged views of selected areas, highlighting CD8+ T cell distribution. **B.** Quantification of CD8+ T cell density per unit area in tumor sections. Data represent mean \pm SEM from $n = 3$ tumors per group. Statistical significance was determined by Student's t-test.

A7. *“Figures 6o–p – T Cell Data Presentation: The current approach of presenting T cell infiltration as a percentage of CD45+ cells may obscure effects on CD4+ T cells. T cell infiltration should instead be quantified as the absolute number of T cells per tumor weight or per cancer cell. Reanalysis with these metrics is likely to reveal additional insights into CD4+ T cell responses.”*

As kindly suggested, we reanalyzed the data by quantifying CD4+ T cells based on the total number of FACS events and normalized them by tumor weight (mg) to obtain absolute cell counts (**Fig. R13**). This metric reflects true T cell infiltration regardless of variations in CD45+ cell proportions, directly addressing the potential bias raised. To ensure consistency, we used freshly isolated single-cell suspensions from half

of each tumor (the other half was processed for FFPE) and excluded tumors that were markedly large. Or small to minimize size-related variability. We selected tumors of intermediate and comparable size (mean weight 0.1 ~ 0.2 mg, or 0.5 mg for control). Notably, CD4+ T cell infiltration per tumor volume was increased in the GSK343-treated CRPR2 group compared to the vehicle group or tazemetostat groups. *These results support the conclusion that EZH2 inhibition promotes CD4+ T cell recruitment into CRPR2 tumors.* This discrepancy between GSK343 and tazemetostat may be partially explained by the differences in drug administration route and bioavailability. Tazemetostat was administered orally, which might have resulted in relatively lower local drug effect compared to intraperitoneally delivered GSK343. While tazemetostat reduced tumor burden, its immune-modulatory effects were less pronounced than GSK343, possibly due to reduced issue penetrance or pharmacokinetic limitations. These updated results were added (**Supplementary Fig. S8c**).

Figure R13. Quantification of CD4+ T cells per tumor weight in CRPR2 tumors treated with EZH2 inhibitors. Total number of CD4+ T cells was quantified from freshly isolated tumors and normalized by tumor weight (mg) to account for tumor size differences.

A8. “EZH2 Inhibitor Effects on T Cells: The study would benefit from an expanded FACS analysis of T cell phenotypes following EZH2 inhibitor treatment, including expression of activation markers and intracellular cytokines. Potential synergistic effects between EZH2 inhibitors and immune checkpoint inhibitors (ICIs) would be clinically relevant and should be investigated.”

Thank you for the comments.

FACS and cytokine profiling. As kindly suggested, we treated C57BL/6 mice with GSK343 and analyzed T cells by FACS. The impact of EZH2 inhibitors on T cell activation or cytotoxicity was barely detected (**Fig. R14**). Therefore, no further cytokine profiling using RNA-seq or scRNA-seq was performed.

EZH2 inhibitors with ICIs. We completely agree with this comment because it is challenging to envision that a single agent is sufficient to inhibit tumorigenesis (herein, CRACD-negative SCLC) without relapse, which leads us to consider combination treatment for clinical applications. In animal models, an EZH2 inhibitor itself was sufficient to suppress SCLC tumorigenesis (**Fig. 4**). Therefore, the additional or synergistic impact of ICIs on top of the effect of EZH2 inhibitors cannot be assessed in animal models. Moreover, among ICIs (PD1, PD-L1, and CTLA4), there is no substantial justification for the choice of ICIs. Respectfully, such experiments would be more suitable in clinical settings or at least using immunologically well-defined SCLC PDXs in humanized mouse models, which is beyond the scope of the submitted manuscript, dissecting the biology of SCLC tumor cell plasticity and immune evasion. We added the comments as future studies in the Discussion.

Figure R14. Flow cytometry analysis of T cells following EZH2 inhibitor treatment in C57BL/6 mice.

A. C57BL/6 mice were treated with GSK343, and T cell populations were analyzed by flow cytometry. Representative flow cytometry plots. **B.** Quantification of helper T cells (CD4+ T cells) cytotoxicity-related markers (Perforin, Granzyme B) in CD8+ T cells.

A9. “CD8 T Cell Depletion Experiment: To substantiate the connection between CD8+ T cells and CRPR2 tumor progression, the authors should conduct a CD8 depletion experiment and analyze its effect on tumor growth. This is particularly critical given that CRACD depletion has intrinsic tumor-promoting effects. If CRPR2 and RPR2 tumors show minimal differences in the absence of CD8+ T cells, it would further support the hypothesis that immune evasion drives CRPR2 tumor outgrowth.”

Thank you for the insightful comments. We agree that utilizing CD8+ T cell-depleted mice will be another strong supportive evidence. In the initially submitted manuscript, we observed that, unlike CRPR2, RPR2 cells **barely** developed tumors in immunocompetent mice (C57BL/6) (**Supplementary Fig. S8d-e**), reiterating the immune evasion of CRPR2 cells.

As kindly suggested, we performed CD8 T cell depletion experiments. RPR2 cells do not develop tumors in C57BL/6 mice (**Supplementary Fig. S8d-e**). Conversely, RPR2 cells developed into tumors in CD8+ T cell-depleted immunocompetent (C57BL/6) mice compared to mice injected with IgG (**Fig. R15a, b** lanes 1 and 2), indicating that RPR2 tumor growth is blocked by T cell-related immune surveillance. Moreover, CRPR2 tumor growth in immunocompetent mice did not show a statistically significant difference between the IgG control and CD8 T cell depletion groups (**Fig. R15b**, lanes 3 and 4), which reconfirms our finding that Cracd LOF-induced immune evasion is related to dysfunction of T cell-related immune surveillance.

As expected, the tumorigenicity of CRPR2 is higher than that of RPR2 in C57BL/6 mice (**Fig. R15**). This is well explained by the *intrinsic tumor-promoting effects* of CRACD LOF; CRPR2 cells grew faster than RPR2 cells in vitro or immunocompromised mice (**Fig. R16**). In addition to MHC-I suppression, CRPR2 cells display cell hyperproliferation (**Fig. 3**), cell plasticity (**Fig. 1, 2**), and an increase in cell heterogeneity (**Fig. 2**), which explains why CRPR2 tumors are still bigger than RPR2 tumors in CD8 T cell-depleted C57BL/6 mice.

Therefore, unlike the comment that “*If CRPR2 and RPR2 tumors show minimal differences in the absence of CD8+ T cells*”, it is expected that CRPR2 tumor cells still show a significant difference in cell or tumor growth compared to RPR2 in CD8 T cell-depleted mice.

New results were added (**Supplementary Fig. S8g**).

Figure R15. Tumor growth under CD8+ T cell depletion in C57BL/6 mice.

A. Experimental timeline depicting subcutaneous injection of RPR2 or CRPR2 cells into C57BL/6 mice followed by anti-CD8 antibody administration. **B.** Tumor weights at sacrifice. Each dot represents an individual tumor.

Figure R16. Tumor growth and representative tumor images of RPR2 and CRPR2 cells in immunodeficient nude mice.

A. Tumor growth curves of RPR2 (blue), and CRPR2 (red) tumors measured over 40 days following subcutaneous injection into nude mice. Arrows indicate the median time points of sacrifice for each group. **B.** Representative images of excised tumors at sacrifice from nude mice bearing RPR2 and CRPR2 tumors. Scale bar: 1cm.

A10. “Conclusion. *The study presents compelling findings on CRACD’s role in immune evasion and the therapeutic potential of EZH2 inhibition in SCLC. However, several key points require clarification, including experimental design details, spatial organization of the tumor microenvironment, and the immune response to EZH2 inhibition. Addressing these points will strengthen the manuscript’s conclusions and provide a more comprehensive understanding of CRACD’s function in SCLC.*”

We have revised the manuscript according to your comments.

As kindly suggested, we added new experimental results and comments to the revised manuscript with clarification. Again, we appreciate very constructive and insightful comments.

Reviewer 2

“This manuscript by Seo et al. presents a significant advancement in our understanding of small-cell lung cancer (SCLC) by elucidating the role of CRACD loss in promoting neuroendocrine (NE) plasticity and immune evasion. The study employs rigorous methodologies, including genetically engineered mouse models, single-cell RNA sequencing (scRNA-seq), spatial transcriptomics, and functional validation experiments, to support its claims. The proposed CRACD-EZH2-MHC-I axis as a potential therapeutic target is compelling and has strong translational implications. However, there are critical areas that require further clarification, methodological refinement, and additional data to enhance the manuscript’s impact and scientific rigor.

This study reveals a previously underappreciated role of CRACD in actin polymerization, tumor cell plasticity, and immune evasion. It establishes that CRACD loss drives neuroendocrine plasticity by suppressing NOTCH signaling, increasing tumor heterogeneity, and facilitating immune escape via EZH2-mediated MHC-I suppression. The findings reported that EZH2 inhibition is a novel therapeutic strategy for CRACD-deficient SCLC. By integrating GEMMs, CRISPR-Cas9 knockout models, scRNA-seq, and spatial transcriptomics, the study provides a high-resolution, multi-dimensional analysis of tumor plasticity and heterogeneity. Functional assays further confirm the efficacy of EZH2 inhibitors in restoring MHC-I expression and reversing immune evasion. Notably, the study identifies a distinct SCLC molecular subtype characterized by CRACD loss, EZH2-driven transcriptional repression, and MHC-I downregulation, highlighting a potential biomarker and supporting the therapeutic relevance of EZH2 inhibition in cold SCLC tumors.”

Major points

B1. “The study establishes a connection between CRACD loss and NOTCH signaling downregulation; however, the precise molecular mechanism remains unresolved. Does CRACD directly interact with NOTCH pathway components, or is this effect mediated through actin cytoskeletal disruption?”

Thank you for the valuable comments. We admit that the initially submitted manuscript did not clearly demonstrate how CRACD LOF downregulates NOTCH signaling. Thanks to the comments, we performed additional experiments. Our new data support an indirect regulatory mechanism, in which CRACD loss leads to reduced actin polymerization (**Fig. 2k**), thereby modulating upstream pathways that influence NOTCH activity. Of note, we detected no signaling components of the NOTCH signaling from tandem affinity purification and mass spectrometry (TAP-MS/MS)².

Mechanism. Briefly, CRACD loss-dysregulated actin polymerization activates the HIPPO signaling that senses mechanical stress³, which subsequently phosphorylates and inhibits YAP1. Then, the downregulation of YAP1-transactivated *NOTCH1*⁴ downregulates the NOTCH signaling, followed by reduced expression of NOTCH1 target genes (*HES1* and *REST*). Finally, NE genes suppressed by *HES1* and *REST* are de-repressed to induce NE cell plasticity (**Fig. R20**).

Supporting data. A. While RPR2 tumors (*Cracd* WT) displayed a distinct nuclear pattern of YAP1 protein expression, CRPR2 (*Cracd* KO) tumor cells barely showed the expression of YAP1 (both nuclear and cytosolic) (**Fig. R21**). We also treated RPR2 cells with verteporfin, a YAP1 inhibitor that disrupts the YAP-TEAD complex. Verteporfin reduced the cleaved Notch1 (N1ICD) protein and its downstream target *HES1* (**Fig. R22**). These results support that CRACD influences NOTCH signaling via the actin-YAP1 axis rather than through direct molecular interaction.

We have updated the discussion and added this data to the revised manuscript (**Fig. 2l-m, Supplementary Fig. S4**).

Figure R20. Illustration of the mechanism of how CRACD inactivation induces NE cell plasticity, tumor cell heterogeneity, and MHC-I suppression.

Figure R21. Immunohistochemistry of SCLC tumors (RPR2 vs. CRPR2) for YAP1. Scale bars: 50 μ m.

Figure R22. Immunoblot analysis of RPR2 cells treated with vehicle or verteporfin 0.5 (+) or 1.0 μ M (++). 72 hours after treatment, cells were harvested for immunoblotting.

B2. “While MHC-I suppression is attributed to EZH2-mediated histone methylation, the potential involvement of additional epigenetic regulators (e.g., HDACs, DNMTs) remains unexplored. Chromatin accessibility assays (e.g., ATAC-seq) could provide further mechanistic clarity.”

As kindly suggested, we performed additional experiments to assess whether MHC-I suppression in CRACD-deficient cells may also involve other epigenetic regulators. We treated CRPR2 cells with SAHA (Vorinostat, 2.5 μ M for 48 hours), an HDAC inhibitor that prevents chromatin compaction, and 5-aza-deoxycytidine (5-Aza-dC, Decitabine, 1 μ M for 72 hours), a DNA methyltransferase inhibitor that induces DNA hypomethylation, and assessed MHC-I expression by immunoblotting (**Fig. R24**). We observed no effects of such inhibitors on MHC-I de-repression.

We agree that a combined RNA-seq and ATAC-seq approach would yield a broader and less biased genome-wide view. Nevertheless, we respectfully disagree with the assertion that ATAC-seq directly elucidates the mechanisms of epigenetic regulation. While ATAC-seq effectively profiles chromatin accessibility, its primary output is information on the openness of genomic regions, rather than the detailed molecular processes underlying epigenetic gene expression.

Figure R24. Expression of MHC-I in CRPR2 treated with vehicle, SAHA (HDAC inhibitor), 4-Aza-dc (DNMT inhibitor), or the combination of SAHA and 5-Aza-dC. RPR2 cells were used as a positive control.

B3. “The interplay between NOTCH signaling downregulation and EZH2-driven immune evasion is briefly mentioned but not thoroughly investigated. Could NOTCH signaling directly regulate EZH2 recruitment?”

Thank you for the comments. β -Actin (*Actb*) KO induces EZH2-mediated gene repression⁵. Consistent with these studies, in the initially submitted manuscript, we proposed that nuclear actin dysregulation by CRACD LOF is a key event inducing EZH2-mediated suppression of genes (**Fig. 4**), including encoding the MHC-1 protein. As suggested, it is also possible that inactivation of NOTCH signaling might trigger EZH2-mediated gene repression. Several studies have shown EZH2’s regulatory mechanisms of the NOTCH signaling, but not vice versa. In our data, EZH2 inhibitors de-repress MHC-I (**Fig. 4g**). However, in CRPR2 cells, where NOTCH signaling is downregulated and EZH2 occupies the MHC-I promoter (**Fig. 4d-e**), NOTCH signaling activation by ectopic expression of N1ICD does not affect MHC-I expression (**Fig. 4j**). These data suggest that NOTCH signaling does not regulate the MHC-I gene promoter activation, excluding the involvement of NOTCH signaling in inhibiting EZH2’s promoter occupancy. Thus, CUT&RUN for EZH2 using CRPR2 cells (vector control vs. N1ICD) is unlikely to be informative. As suggested, the revised manuscript discussion now includes better clarification (highlighted; lines 366-370).

B4. “The observed depletion of intratumoral CD8+ T cells in CRACD-deficient tumors may be influenced by additional immunosuppressive populations, such as myeloid-derived suppressor cells (MDSCs) and regulatory T cells (Tregs). A more comprehensive immune profiling using flow cytometry or single-cell analysis is

required.”

Figure R25. Single-cell transcriptomic analysis of T cells reveals a CRPR2-enriched inflammatory subpopulation. (a) UMAP visualization of 5,152 T cells from RPR2 and CRPR2 tumors, colored by cluster identity. Pie charts indicate the cluster composition within each sample. (b) Bar plot showing cluster proportions and dot plot displaying average expression of major T cell subtype markers across clusters. Notably, CD8⁺ T cell subclusters were selectively depleted in CRPR2, whereas cluster 7, enriched in CRPR2, exhibited high expression of *Foxp3*, indicative of regulatory T cells. Inset violin plot illustrates elevated *Foxp3* expression in cluster 7. (c) Heatmap of differentially expressed genes within cluster 7, comparing CRPR2 and RPR2 T cells. CRPR2 cluster 7 cells showed upregulation of immunoregulatory genes including *Tnfrsf18*, *Tff1*, *Ctla2a*, and *Ahr*. (d) Violin plots showing significantly higher expression of *Tnfrsf18*, *Tff1*, *Ctla2a*, and *Ahr* in CRPR2 cluster 7 cells compared to RPR2. Statistical analysis was performed using the Wilcoxon rank-sum test; ***P* < 0.01, ****P* < 0.001.

Thank you for this insightful comment. To address the potential contribution of immunosuppressive populations to CD8⁺ T cell depletion, we analyzed Tregs (Cd4⁺ Ikkzf2⁺) and MDSC-like myeloid cells (Itgam⁺ Cd14⁺ Clec4e⁺ Arg2⁺) based on our scRNA-seq data. Treg populations were readily detectable and quantifiable, and CRACD-deficient tumors exhibited an increased abundance of these cells (**Fig. R25**). Of note, a distinct MDSC subcluster was not clearly separable, likely due to the limited number of immune cells and overlapping marker expression. These new results were added (**Fig. 3c-f**).

B5. “While the study suggests that restoring MHC-I expression counteracts immune evasion, it does not confirm whether this results in functional CD8⁺ T cell activation. In vivo, cytotoxicity assays and cytokine profiling would strengthen this conclusion.”

Although in vivo cytotoxicity assays were not feasible in our current setting, we performed alternative in vitro experiments to assess functional activation of CD8⁺ T cells. (**Fig. R26**).

CD8⁺ T cells were co-cultured with CRPR2 or RPR2 tumor cells, and intracellular expression of Perforin and Granzyme B was assessed by flow cytometry. As expected, CRPR2 tumors showed significantly reduced frequencies of Perforin⁺ CD8⁺ T cells, which were partially restored upon EZH2 inhibitor (GSK343 and Tazemetostat) treatment. These results directly demonstrate functional restoration of cytotoxic CD8⁺ T cell activity via MHC-I re-expression.

Although Granzyme B⁺ CD8⁺ T cells also showed an increasing trend with EZH2 inhibitor treatment, the difference did not reach statistical significance. This may reflect distinct activation thresholds and temporal kinetics between effector molecules—Perforin being induced earlier during activation, while Granzyme B may require stronger or prolonged stimulation. Additionally, the limited stimulatory context of the co-culture assay may have constrained full Granzyme B induction. Nonetheless, the recovery of Perforin⁺ CD8⁺ T cells provides functional evidence that MHC-I

Figure R26. Functional restoration of cytotoxic CD8⁺ T cells upon EZH2 inhibitor treatment in vitro.

A. Representative flow cytometry plots showing intracellular Perforin and Granzyme B expression in CD8⁺ T cells co-cultured with RPR2 and CRPR2 cells, with or without EZH2 inhibitor treatment. **B.** Quantification of Perforin⁺ and Granzyme B CD8⁺ T cell frequencies.

restoration effectively counteracts immune evasion in CRPR2 tumors. New results were added (**Supplementary Fig. 8f**).

B6. “EZH2 inhibition is shown to suppress tumor growth, but its effect on overall survival remains unexamined. Kaplan-Meier survival curves should be included to assess therapeutic impact.”

In this study, we used a subcutaneous tumor model to assess the anti-tumor efficacy of EZH2 inhibitors. In all our animal experiments, death is not an endpoint. Animals were euthanized according to IACUC guidelines when control tumors reached 2 cm in diameter. Therefore, overall survival could not be assessed in this setting. Nonetheless, we showed all individual data points/lines in our tumor quantification plots without excluding any data (i.e., potential outliers) (**Fig. 4l** and **Supplementary Fig. 8b-d**). We agree that Kaplan-Meier survival analysis would be informative in a systemic or orthotopic model. However, given the clinical availability of an EZH2 inhibitor (tazemetostat), KM plotting in clinical studies might be more informative than those from preclinical models, including syngeneic transplantation models.

B7. “Given SCLC’s heterogeneity, other epigenetic vulnerabilities in CRACD-low tumors may exist. Identifying potential co-targets with EZH2 could refine therapeutic strategies.”

Thank you for the comments. Aside from the results from preclinical models, considering the heterogeneity of CRACD inactivated SCLC, there may be additional epigenetic vulnerabilities that could serve as secondary weak points for overcoming therapy resistance or relapse.

Figure R27. GSEA plots showing KEGG_RIBOSOME pathway enrichment in CRPR2 versus RPR2 cells within clusters 5, 6, 7, 9, and 10. Only clusters with adjusted $p < 0.05$ are shown. NES and P -values are indicated.

For instance, while our study focused on EZH2 as a key effector of immune evasion, combinatorial targeting strategies involving EZH2 and other epigenetic regulators may further enhance therapeutic efficacy. As kindly suggested, we performed pathway analyses from scRNA-seq datasets of RPR2 vs. CRPR2 tumors. We did not limit our analysis for identifying “other epigenetic vulnerabilities” since there are only a few in FDA-approved drugs targeting epigenetic regulators (e.g., DNMTs, HDACs, and EZH2). Our analysis identified ribosome signaling pathways as significantly activated in CRPR2 compared to RPR2 (**Fig. R27**). Notably, several FDA-approved drugs, such as homoharringtonine (omacetaxine mepesuccinate), a protein translation inhibitor targeting the ribosome, are clinically available and have been used in hematologic malignancies⁶. While these findings suggest that ribosome-targeting agents may represent potential co-targets with EZH2 inhibition, this hypothesis requires further experimental validation and is beyond the current scope of this manuscript. We have briefly incorporated this perspective into the Discussion (highlighted; lines 394-402). Thank you for the insightful comment.

B8. “The scRNA-seq findings are compelling, but additional validation at the protein level is essential. Western blot or immunofluorescence for key markers identified in human SCLC samples would strengthen the conclusions.”

Thank you for the insightful comments. As suggested, we performed IHC on human SCLC samples to validate the expression of key markers (CRACD, EZH2-mediated histone modification [H3K27Me2/3], and MHC-I) identified from our studies. The IHC results confirmed differential expression of representative markers at the protein level, thereby supporting the transcriptomic findings. Briefly, CRACD downregulation is correlated with MHC-I downregulation, whereas it is inversely correlated with EZH2 histone markers (H3K27Me2/3) (**Fig. R28**). New results were added (**Supplementary Fig. 11a-b**).

Figure R28. Immunohistochemical staining of CRACD, MHC-I, H3K27Me2, and H3K27Me3 in human SCLC tissues. Tumor sections were grouped based on CRACD expression status, and corresponding changes in MHC-I and histone methylation marks were examined. Quantification was performed using H-score analysis. Statistical significance was assessed by Student's t-test. Data are presented as mean \pm SEM; $P < 0.05$ was considered significant.

B9. “Certain statistical comparisons require greater transparency regarding methodology. The manuscript should clarify whether multiple testing corrections were applied in pathway analyses.”

We thank the reviewer for this helpful comment. As suggested, we have clarified in the Methods section that multiple testing correction was applied in pathway enrichment analyses. Specifically, GSEA was performed using the *fgsea* R package with 2,000 permutations, and adjusted *p*-values (*padj*) were calculated using the Benjamini–Hochberg method to control the false discovery rate (FDR). This information is now explicitly stated in the revised Methods section under “Gene set enrichment analysis (GSEA)” (highlighted).

B10. “The study does not compare EZH2 inhibition with current immunotherapies, such as PD-1/PD-L1 blockade. Could combination therapy enhance therapeutic efficacy? Does it provide an added advantage as combination therapy or an alternative to existing therapies?”

EZH2i + ICIs. We agree that combination therapy with immune checkpoint inhibitors (ICIs) could be a rational strategy to enhance therapeutic efficacy, particularly given the likelihood of relapse with monotherapy in CRACD-deficient SCLC. In our current study, we did not carry out combination therapy for the following reasons. First, *EZH2 inhibition alone was sufficient to suppress tumor growth in animal models*. Therefore, evaluating the effect of combination therapy (EZH2i + ICIs) was *not feasible in our preclinical settings*. Second, the choice of specific ICIs lacks experimental justification in our mouse models. Third, although using human SCLC cells or PDXs is more pathologically relevant, especially to immunotherapies, humanized mice cannot fully recapitulate the human immune landscape.

Perspective. Yes, we strongly agree that combination therapy would enhance therapeutic efficacy in clinical applications, which will improve current immunotherapy strategies by detailed patient stratification, e.g., CRACD expression and MHC-I expression in addition to the expression of immune checkpoint molecules. The limited efficacy of monotherapy in overcoming resistance or relapse is a well-established challenge in clinical trials. In contrast, our preclinical models indicate that EZH2 inhibition alongside ICIs could offer a compelling solution, especially for patients whose tumors exhibit CRACD loss-of-function (LOF). Our study proposes a biomarker-guided strategy, integrating targeted EZH2 inhibition and immunotherapy (ICIs), with the potential to overcome the shortcomings of current ICI-based treatments. Such information was added to the Discussion. Thank you.

Minor points

B11. “To enhance translational impact, the graphical abstract should more explicitly emphasize the therapeutic relevance of the CRACD-EZH2-MHC-I axis.”

As suggested, we have revised the graphical hypothesis (Fig. R29, Fig. 5j).

Figure R29. Graphical abstract

B12. *“The discussion would benefit from a broader integration of findings from similar studies on tumor plasticity in other malignancies, such as prostate and pancreatic neuroendocrine tumors.”*

As suggested, we have expanded the discussion section to include relevant findings from studies on tumor plasticity in prostate^{7,8} and pancreatic neuroendocrine tumors to provide a broader context⁹. (highlighted)

B13. *“Additional references on EZH2 inhibitors in SCLC and other cancers would provide a more comprehensive contextualization of the therapeutic approach.”*

As recommended, we have included an additional reference¹⁰ (Highlighted)

B14. *“Certain sections, particularly within the results, contain dense and complex sentences that could be streamlined for clarity and conciseness.”*

The writing editor (Christine F. Wogan) at MD Anderson Cancer Center revised the results for better legibility.

B15. *“The term “immune evasion” is used broadly throughout the manuscript. Distinguishing between specific mechanisms, such as T cell exclusion versus antigen presentation loss, would improve precision in interpretation.”*

As kindly suggested, we have refined the description of immune evasion throughout the manuscript by specifying distinct mechanisms such as T cell exclusion and antigen presentation loss where appropriate (highlighted).

“In summary, this study provides a compelling contribution to SCLC research by identifying CRACD as a key tumor suppressor that constrains NE plasticity and immune evasion. The mechanistic insights into actin cytoskeletal regulation, NOTCH signaling suppression, and EZH2-driven epigenetic remodeling are highly valuable. However, the study lacks a clear mechanistic link between CRACD, NOTCH, and EZH2. Additionally, expanding immune profiling to better characterize the tumor microenvironment, functionally validating the impact of EZH2 inhibition on CD8+ T cell activation and cytotoxicity, and assessing survival outcomes while exploring combinatorial therapeutic strategies will further strengthen the findings. Addressing these critical points will enhance the manuscript’s scientific rigor.”

We appreciate all critical comments that improve the manuscript.

Reviewer 3

“This manuscript, written by Seo et al., describes the role of the actin de-capping inhibitor, CRACD, in small cell lung cancer (SCLC). CRACD promotes the actin polymerization, and is thereby involved in the plasticity and immune reactivity of SCLC. The authors conducted the single cell RNA sequencing (scRNA seq) analysis and the spatial analysis (Xenium in situ hybridization) for the mouse model of SCLC, in which RB1, TP52 and RBL2 were knocked out (RPR2 KO mice). Using this mouse model, they found that the CRCD transforms the pre-neoplastic cells into SCLC tumor-like cells, including the compaction of the cells. CRACD also induces neuroendocrine (NE) plasticity and increases the tumor cell heterogeneity via dysregulation of NOTCH1 pathway. Even more importantly, they observed that CARCS induced the EZH2-mediated H3K27 trimethylation by the nuclear actin reduction. This epigenomic change suppresses the expression of the MHC-I genes. As a result, the recruitment of cytotoxic T cells (CTLs) are reduced. As an attempt to utilize these findings for developing a new drug treatment, the authors treated the CRCD KO-RPR2 mice with an EZH2 inhibitor. They found that the MHC-I expression and the immune surveillance were actually restored. Even though further analyses directly using human material may be needed, this study paves the first step towards the combination therapy of the ICB and EZH2 inhibitor to this difficult cancer species. Overall, I think this is the well-written paper, supported by the solid and a wide variety of experimental evidence. The proposed perspective to utilize the EZH2 is timely and the one which a large number of the patients having SCLC are looking forward to.”

Major points

C1. *“While I fully understand that the scope of this particular paper should lie in the study of the pre-clinical mouse model, I’d still like to request the authors to conduct the extensive analysis to demonstrate the relevance of the obtained results directly using the human material, at least, to some extent. The presented data is, after all, from the mouse model and does not always indicate the in vivo relevance. In fact, the molecular mechanisms underlying the lung cell lineage and the cancer development are supported to occasionally quite different from humans, even though superficial appearance may look similar.”*

Thank you for the constructive comments. We agree that analyzing more human patient samples will strengthen our findings. Unfortunately, we had to admit that dissecting “the molecular mechanisms underlying the lung cell lineage and the cancer development using human samples” is barely feasible. Using SCLC PDOs might be the only option, which has not been greatly successful. Moreover, the scRNA-seq datasets from human SCLC tumor samples are very limited. As shown in Figure 5, we performed extensive analysis using the previous scRNA-seq datasets of SCLC patients’ tumors (n=19), showing the relevance of CRACD status to MHC-I downregulation and NOTCH signaling pathways.

As kindly suggested, we conducted additional analysis by IHC of SCLC tumor microarray and observed that CRACD downregulation is correlated to the loss of MHC-I, the elevated H3K27me2/3. Conversely, in CRACD-high SCLC tumors, MHC-I expression was elevated, while H3K2me2/3 levels were relatively reduced (**Fig. R31**). New results were added (**Supplementary Fig. 11a-b**).

Figure R31. Immunohistochemical staining of CRACD, MHC-I, H3K27Me2, and H3K27Me3 in human SCLC tissues. Tumor sections were grouped based on CRACD expression status, and corresponding changes in MHC-I and histone methylation marks were examined. Quantification was performed using H-score analysis. Statistical significance was assessed by Student's t-test. Data are presented as mean \pm SEM; $P < 0.05$ was considered significant.

C2. “Results of the Xenium analysis is not fully associated with those of the scRNA analysis. The spatial locations of the respective single cells should be presented. Also, I assume the data from the 379 gene Mouse Tissue Atlas Panel plus 100 gene custom should not be so much comprehensive. In addition to showing just the cellular composition or the tumor heterogeneity, please show which parts of the single cells profiles are directly represented in the Xenium data.”

Thank you for the comments. As suggested, we visualized the scRNA-seq information (Figure 3) on the Xenium results.

Xenium In Situ before 5K genesets availability. We initially planned to combine scRNA-seq with Xenium In Situ to complement each other. Our Xenium experiments were performed with the customized 479 genesets before the release of 5K genesets. With the limited number of genes, we were only able to acquire cell cluster-related information, which was used for analyzing tumor cell heterogeneity (**Fig. 2**).

Visualizing scRNA-seq information on Xenium. Cell lineages. As suggested, we projected cell clusters and cell proliferation information from scRNA-seq onto Xenium data. Consistent with the scRNA-seq results, we located new non-NE cell clusters (purple) generated in CRPR2 tumors at the core of SCLC tumor nodules. These non-NE cells were surrounded by NE cells (**Fig. R32A, B**), recapitulating NE cell plasticity of CRPR2 tumors. Additionally, CRPR2 tumor cells are more proliferative than RPR2 tumor cells. Moreover, NE cells are relatively more mitotic than non-NE cells (**Fig. R32C**). **Immune niche.** We also quantified the T cells infiltrated into SCLC tumors (RPR2 vs. CRPR2). In line with the results from scRNA-seq and immunostaining-based immune profiling (**Supplementary Fig. S6**), T cells were rarely detected in CRPR2 tumors compared to RPR2 tumors (**Fig. R32D, E**). These new results were added to the revised manuscript (**Fig.2f-h, Supplementary Fig. S6i-k**).

It should be noted that *Cracd* KO itself is sufficient to upregulate NE and SCLC markers in lung organoids and GEMMs¹ (**Fig. R32**). Therefore, regardless of RBL2 LOF, it is highly likely that CRACD loss induces NE cell plasticity, promoting SCLC tumorigenesis in a cell-autonomous manner. Similarly, we recently found that *Cracd* KO upregulates NE genes in GEMMs and lung organoids¹.

Figure R32. Spatial localization and characterization of tumor cell subpopulations and T cells in RPR2 and CRPR2 tumors. **A.** UMAP projection of scRNA-seq data from RPR2 and CRPR2 tumors. Cells are colored by unsupervised clusters; NE and non-NE populations are delineated by dashed lines. **B.** Spatial localization of NE and non-NE clusters in representative RPR2 and CRPR2 tumors. Cluster identities were assigned by correlation-based mapping of Xenium transcriptomic data to matched scRNA-seq clusters. **C.** Spatial mapping of cell cycle states projected onto Xenium coordinates. Cell cycle phase (inferred from matched scRNA-seq data) are shown as: G1 (magenta), S (aqua), and G2/M (purple). **D.** Spatial localization of T cells (Cd3d, Cd3e, Cd3g, Cd8a, Cd8b1, Cd4) within RPR2 and CRPR2 tumors, identified by Xenium transcriptomic signatures. **E.** Quantification of T cells density per unit area from Xenium spatial data.

C3. “To me, the highlight of this paper is the use of an EZH2 inhibitor. On the other hand, I think the current EZH2 inhibitors should invoke substantial effects on the epigenome statuses of indirect target sites (as a secondary effect) or on indirect target genes other than EZH2. In addition to the MHC-I genes, the sites where the relevant changes were observed should be further scrutinized. Also, I wonder if some key cancer gene regulations may be also influenced?”

Despite the promising outcome of EZH2 blockade in preclinical models, we understand the possible off-target and/or secondary effects of EZH2 inhibitors beyond MHC-I. Also, EZH2 has been shown to promote tumorigenesis by methylating non-histone proteins^{11,12}. To address this, we performed bulk RNA-seq of CRPR2 cells treated with EZH2 inhibitors ($n=3$ per group). Of note, compared to scRNA-seq, bulk RNA-seq provides relatively deeper sequencing reads. It was reproducible to see the de-expression of MHC-I transcripts in CRPR2 cells treated with EZH2 inhibitors, along with the upregulation of KRAS signaling and EMT, and the downregulation of MYC targets, DNA repair, DNA replication, ribosome biogenesis, and RNA metabolic processes from GSEA (**Fig. R33**). Thus, in addition to EZH2i, using Kras inhibitors, PPARi, or HHT could be considered. New results were added (**Supplementary Fig. S9a-d**). Respectfully, we believe that further investigation of additional EZH2 targets and their inhibitors is out of the scope of this manuscript.

It is noteworthy that, despite the additional impact of EZH2 inhibitors on such signaling pathways, we strongly believe that the anti-tumorigenic effect of EZH2 inhibitors is mainly due to the changes related to immunosurveillance, including antigen processing and presentation. This is because, unlike RPR2 cells, CRPR2 cells develop tumors relatively well in immunocompetent mice (C57BL/6) (**Supplementary Fig. 8d-e**), which is markedly suppressed by EZH2 inhibitors (**Fig.4I-n**).

Minor points

C4. “I’m not totally sure how the obtained results should be interpreted in the context of the previous molecular subtypes (ASCL1, NeuroD1, POU2F3 and YAP1) and the neuroendocrine (NE) or non-neuroendocrine subtypes. Sometimes, those particular subtypes are associated with inflammatory subtypes, to which immune check point blockage (ICB) should be effective. However, as mentioned by the authors themselves, these classifications are not always consistent with clinical phenotypes (or drug responses) of the patients. Honestly,

I'm not familiar with this mouse model. In fact, during the review. I firstly found a quite frequently employed model for this purpose. Therefore, I wonder what subtype this mouse is modeling (also related to the point 1). Please carefully discuss the limit of the present analysis."

We thank the reviewer for raising this important point.

Subtypes. ANPY subtypes were first introduced based on the "bulk" RNA-seq¹³⁻¹⁵. Later, the inflammatory subtype was claimed based on the PDXs¹⁵. Thus, directly comparing the information from bulk RNA-seq with scRNA-seq (tumor cells only in this study) might not be informative. For example, despite the popularity of ANPY subtyping, scRNA-seq datasets do not show such distinct subtypes (except for the POUF1 subtype) (**Fig. 5d**).

GEMMs. RPR2 GEMMs have been extensively used^{16,17}, displaying the feature of A subtype SCLC (ASCL1). Our scRNA-seq analysis of human SCLC tumors revealed that CRACD-low tumors (MS1) belong to the ASCL1 subtype, while CRACD-high tumors (MS2) encompass all four ANPY subtypes (**Fig. 5d**), which supports our rationale for using RPR2 to study CRACD inactivation. We have addressed this point in the revised manuscript.

Tumor cell heterogeneity. Most SCLC preSCs and GEMMs presented in this study exhibit the NE features. Not surprisingly, many groups, including us, have identified dynamic cell plasticity, a conversion between non-NE cells to NE cells in SCLC^{18,19}, along with EMT gene expression as well as Myc expression. Our results showed such cell lineages and cell plasticity at single-cell levels for the first time (**Fig. 2f-h**). Intriguingly, our recent study also showed that *Cracd* KO induces NE cell plasticity of lung adenocarcinoma¹. It is noteworthy that the outcome of CRACD LOF is the loss of epithelial integrity, which is similar to EMT.

Together, despite the conventional SCLC subtypes and classical pathological classification, single-cell analysis indicates dynamic and heterogeneous features of SCLC. *Nonetheless, we fully appreciated the limitation of the RPR2 preclinical model and its possible incompatibility with human tumors.* As kindly suggested, such information and limitations were further discussed (highlighted). Thank you.

C5. *"Relatedly, in human cancers, the RBL2 mutation does not always co-occur with the CRACD mutation. I wonder how the neoplastic phenotype which appears in the mouse model without the RBL2 mutation should be further connected to the results obtained from the RPR2 mouse model. For this purpose, I'd like to see the spatial analysis of the preSC mice. At least, please include the comparison analysis on the spatial characteristics between the RPR2 and CRPR2 mice."*

Thank you for another insightful comment. *Rbl2* KO is often used for GEMMs since it accelerates SCLC tumorigenesis. Without *Rbl2* KO, it takes around 12 months to develop SCLC tumors in RP mice. In our understanding, the reviewer's comment is to assess the impact of *Cracd* KO on SCLC tumorigenesis in the absence of *Rbl2* KO by running Xenium in preSC mice. Unfortunately, we do not have a budget (14K) for additional Xenium of preSC tumors (subcutaneous transplantation), which could still be inferior to GEMMs. Alternatively, as suggested, we further compared Xenium data with scRNA-seq data (RPR2 vs. CRPR2). Please see the response to C3.

C6. *"Innocently, I also wonder, in this mouse model, where in the lung and what cell type the preSC cells and LCSC cells are originating from."*

The primary cells of origin for SCLC are known to be lung neuroendocrine (NE) cells²⁰. This cell type accounts for approximately 0.5% of all lung epithelial cells and expresses a number of neuroendocrine markers²¹, including ASCL1, CGRP, SYP, and CHGA. PreSC cells and mouse SCLC cells continue to express the same markers, which help define them as neuroendocrine cells. Nonetheless, we do not entirely exclude any engagement of NE cell plasticity in generating NE tumor cells from non-NE cells, as we recently reported¹.

C7. *"The interpretation of the Notch-Delta signaling status is not clear to me in the context of the present paper. This axis is pivotal to the transition from NE to non-NE subtypes, as often referenced (even though it*

may not be the case of SCLC; Yan Ting Shu et al, Nat Com 2022, for example). There, REST is also involved, which might be also involved with the EZH2 inhibition. In the single cell data, I wonder if the whole picture of the regulatory network may be represented for the possible feed-back or cross-talk of the pathways (which may be also, as least to some extent, represented by the Xenium data).”

Thank you for sharing another valuable insight and literature. Briefly, our findings are somewhat consistent with the Julien Sage lab’s report, i.e., the YAP-NOTCH-HES1/REST axis inhibits NE gene expression⁴ and findings from the Trudy Oliver lab., MYC-driven non-NE cell plasticity with YAP and NOTCH signaling activation^{4,19}.

YAP-NOTCH-REST axis in inhibiting NE cell plasticity. In addition to the results in this study, our recent paper¹ and a manuscript in revision showed that CRACD LOF consistently inhibits the NOTCH signaling, leading to the activation of the secretory cell lineages, including mucinous or neuroendocrine ones, depending on the context. Yang Ting Shu et al. showed that YAP1-transactivated NOTCH2 directly activates HES1 and REST to suppress NE genes⁴.

Mechanism. Briefly, CRACD loss-dysregulated actin polymerization activates the HIPPO signaling that senses mechanical stress³, which subsequently phosphorylates and inhibits YAP1. Then, the downregulation of YAP1-transactivated NOTCH⁴ downregulates the NOTCH signaling, followed by reduced expression of NOTCH1 target genes, HES1 and REST. Finally, NE genes suppressed by HES1 and REST are de-repressed to induce NE cell plasticity (Fig. R35).

Supporting data. A. While RPR2 tumors (Cracd WT) displayed a distinct nuclear pattern of YAP1 protein expression, CRPR2 (Cracd KO) tumor cells barely showed the expression of YAP1 (both nuclear and cytosolic) (Fig. R36). We also treated RPR2 cells with verteporfin, a YAP1 inhibitor that disrupts the YAP-TEAD complex. Verteporfin reduced the cleaved Notch1 (N1ICD) protein and its downstream target HES1 (Fig. R37). These results support that CRACD influences NOTCH signaling via the actin-YAP1 axis. We have updated the discussion and added these data to the revised manuscript (Fig. 2, Supplementary Fig. S4).

Figure R35. Illustration of the mechanism of how CRACD inactivation induces NE cell plasticity, tumor cell heterogeneity, and MHC-I suppression.

Figure R36. Immunohistochemistry of SCLC tumors (RPR2 vs. CRPR2) for YAP1. Scale bars: 50 μ m.

Figure R37. Immunoblot analysis of RPR2 cells treated with vehicle or verteporfin 0.5 (+) or 1.0 μ M (++) 72 hours after treatment, cells were harvested for immunoblotting.

Visualizing the YAP-NOTCH-REST axis in scRNA-seq.

CRACD is a capping protein inhibitor, promoting actin polymerization². Thus, as kindly pointed out, it is highly likely that CRACD LOF-reduced actin polymerization (F-actin) (**Fig. 4h**) inhibits the HIPPO signaling and subsequently activates YAP1-mediated *NOTCHs* expression, leading to the de-repression of NE genes. This may explain how CRACD LOF induces NE cell plasticity of the NE cell clusters. Indeed, compared to RPR2, CRPR2 tumors exhibited an increase in the number and cell plasticity of non-NE tumor cells (**Fig. 2d-e**). This might be due to the activation of the YAP1-NOTCH2-REST axis, as described in Yang Ting Shu et al⁴. In the initially submitted manuscript, we observed the marked upregulation of *Hes1*, *Dll1*, *Jag1*, *Notch1*, *Notch2*, and *Notch3* in the non-NE cell clusters of CRPR2 compared to RPR2 (**Supplementary Fig. S4d**). Similarly, *Yap1*, *Taz*, *Tead1/4*, and *Rest* were upregulated in non-NE cells of CRPR2 tumors compared to RPR2 (**Supplementary Fig. S4b-c, R39**). Thus, increased non-NE cell clusters in CRPR2 tumors can be well explained by the activation of the YAP-NOTCH-REST axis.

Unfortunately, given that our customized geneset of Xenium did not include those genes, we could not combine scRNA-seq with Xenium to spatially visualize the YAP-NOTCH-REST axis.

Figure 3 (from the initially submitted manuscript)

Figure R39. Feature plots of scRNA-seq datasets of RPR2 and CRPR2 tumors. **A.** Upregulation of the Yap1/Taz-REST axis in non-NE cell clusters of CRPR2 tumors. **B.** Upregulation of *Ezh2* mainly in NE cell clusters of CRPR2 tumors, compared to RPR2.

YAP-NOTCH-REST axis in EZH2-mediated MHC-I suppression of *Cracd* KO tumors. Given that REST recruits EZH2 (PRC2) to repress gene expression via H3K27me2/3²², it is possible that EZH2-induced silencing of MHC-I could also be due to YAP-NOTCH-REST axis activation, at least in non-NE cell clusters. However, we found that *Ezh2* is mainly expressed in NE cells but not non-NE cells, where NE markers are suppressed with HES1/REST activation (**Fig. R39B**). Therefore, it is implausible that the REST-EZH2 axis is engaged in suppressing NE gene expression.

New results were added (**Supplementary Fig. S4**), and further discussion was made (highlighted).

C8. “Discussion should include the recent papers which also describe the single cell features of SCLC (Leslie Duplaquet et al, *Nature Cell Biology* 2023 and Abbie S. Ireland et al, *Cancer Cell* 2022). Those papers may be also useful to give a more comprehensive view for what is occurring in the SCLC cells and how the epigenome perturbation may make effects (either beneficial or adverse). Also, possible involvement of *Myc* could be also discussed.”

As kindly recommended, two references were added. *Myc* was discussed (highlighted). Of note, compared to RPR2, CRPR2 tumors showed *Myc* upregulation in non-NE tumor cells (**Supplementary Fig. S4d**), in line with the findings of the Trudy Oliver group’s paper¹⁹ showing non-NE cell transition by MYC.

C9. *“Please show the list of the genes used for the custom 100 probes for the Xenium analysis. I also wonder how the timing of the sacrifice was determined. The time course sampling should be also useful to validate the robustness of the analysis.”*

The list of genes used for the custom 100 probes in the Xenium analysis has been included in **Supplementary Table 9**. The 6-month post-Cre infection time point was selected because RPR2 (Rb1/Trp53/Rbl2-mutant) mice commonly reach endpoints defined by approved animal protocols and tumor progression at this stage. This timing also allows a robust evaluation of the additional impact of Cracd loss on tumor formation. This standard time point has been widely adopted in prior studies using this GEMM¹⁶. While we agree that time-course sampling could further strengthen the analysis, our current study focused on characterizing the tumor microenvironment at a fully developed stage. Accordingly, samples at earlier time points were not collected. It will certainly be followed up in our future animal experiment, where the Cre-infected lungs are collected at multiple time points and analyzed using Xenium.

C10. *“Please make sure that the Xenium data is also made publicly available in GEO. Also, for all the codes for the data process in GitHub, especially for the parts of the relatively new analyses, including the VR analysis, which I found intuitively very much helpful. (I have not visited the indicated URLs for those datasets using the reviewer token.)”*

We thank the reviewer for this suggestion and have updated the Data and Code availability statements accordingly:

Data availability

All datasets generated in this study have been deposited in the Gene Expression Omnibus (GEO) under the following accession numbers (reviewer tokens provided):

- scRNA-seq: GSE218544 (token: efsxisoijvwzhuh)
- CUT&RUN-seq: GSE280263 (token: uzqpeauszlatteb)
- Xenium spatial transcriptomics: GSE299069 (token: qvkjqkquhtynjkv)

Code availability

All custom scripts and pipelines used for data processing and analysis—including the VR analysis—are publicly available at:

https://github.com/jaeilparklab/CRACD_SCLC_scRNAseq

We appreciate very constructive comments.

Reviewer 4

“Seo et al. examined the impact of CRACD inactivation on small cell lung cancer (SCLC) tumor development, plasticity and immune evasion. The Authors examined Cracd/CRACD in SCLC in vitro and in vivo model systems, human primary SCLC samples with experimental and computational approaches and found that loss of Cracd/CRACD disrupted cytoplasmic and nuclear actin organization leading to dysregulation of NOTCH1 signaling and EZH2-mediated suppression of MHC-I pathways. Alterations to these processes in turn resulted in increased plasticity and immune evasion. The Authors’ demonstration of cellular plasticity using single cell and spatial genomics was state-of-the-art and also contributed to their hypothesis. Functional validation of computational results using animal and in vitro models of SCLC was also comprehensive. The study also suggested EZH2 inhibition as a potential strategy to improve response to immune checkpoint blockade (ICB) therapy in SCLC. Overall, this is a very comprehensive and well-executed study with high translational relevance. To strengthen the study further, the authors should consider the following questions/issues:”

Major comments

D1. “Given germline constitutive knockout of Cracd in the qKO model, is it possible that the differences in the tumor immune microenvironment and even in the cancer cells themselves in the tKO vs. qKO models also stemmed from Cracd loss in the stromal, immune and other non-cancer cells in the mouse? Has the impact of Cracd loss in various immune and other stromal compartments such as T cells been documented in vitro and in vivo models in the Cracd KO control mouse?”

Thank you for the insightful comments. We agree that the differences observed between tKO and qKO models could be influenced not only by tumor-intrinsic factors but also by changes in the stromal and/or immune compartments.

Cracd KO in tumor cells vs. non-tumor cells. To address this potential issue from germline KO models, we performed *syngeneic transplantation* of SCLC tumor cells (RPR2 [Cracd WT] vs. CRPR2 [Cracd KO]; derived from GEMMs [C57BL/6]; **Fig. 1a**) into C57BL/6 mice (Cracd WT) (**Fig. 4, Supplementary Fig. S8**). In this experimental setting, only tumor cells carry the Cracd KO. Consistent with the results from GEMMs (**Fig. 3**), tumor development from CRPR2 cells was significant compared to RPR2 (**Fig. 4I, Supplementary Fig. 8d-e**), excluding potential pitfalls in using Cracd germline KO mice. We recently demonstrated the NE cell plasticity of LUAD cells by using conditional Cracd KO¹. In this study, we utilized both Cracd KO GEMMs and somatic engineering by intratracheal instillation of adenoviruses encoding Cas9-sgCracd-Cre and observed similar results, a conversion of LUAD into NE tumor cells.

Impact of Cracd KO on tumor niche. We identified the CRACD gene as a tumor suppressor and established Cracd germline KO mice². In mice, Cracd KO is sufficient to induce intestinal hyperplasia², mucinous cell plasticity of the intestine (unpublished), NE cell plasticity of LUAD¹, and hyperpigmentation of the skin (unpublished). Cracd KO mice are viable without any discernible phenotypes. It should be noted that CRACD, as a capping protein inhibitor, promotes actin polymerization and is generally expressed in epithelial cells. In our recent study¹, we analyzed Cracd WT vs. KO lung tissues for scRNA-seq datasets. We found no outstanding differences in cell clusters and ratios of immune cells, fibroblasts, and endothelial cells in Cracd KO mice compared to WT¹.

Therefore, the impact of Cracd KO on the tumor niche, including immune cells, might be marginal or insufficient to be associated with NE plasticity. We added such information in the Discussion as an alternative interpretation as well as limitations.

D2. “The authors argued that CRACD inactivation, via actin dysregulation, inhibited NOTCH1 signaling, which in turn promoted neuroendocrine (NE) plasticity. Why would this phenotype not occur in the non-NE population with activated NOTCH1 signaling in the qKO tumors? In Figure 7j, the authors claimed that increased non-NE features in qKO tumors may occur via EMT activation, although correlations between the two were only observed. The authors should qualify these claims more or provide additional evidence of the CRACD inactivation-EMT-non-NE axis.”

Thank you for insightful comments. To some extent, our results align with the Julien Sage lab's report, i.e., the YAP-NOTCH-HES1/REST axis inhibits NE gene expression⁴ and the Trudy Oliver lab's finding., MYC-driven non-NE cell plasticity with YAP and NOTCH signaling activation^{4,19}.

YAP-NOTCH-REST axis in inhibiting NE cell plasticity. In addition to the results in this study, our recent paper¹ and a manuscript in revision showed that CRACD LOF consistently inhibits the NOTCH signaling, leading to the activation of the secretory cell lineages, including mucinous or neuroendocrine ones, depending on the context. Yang Ting Shu et al. showed that YAP1-transactivated *NOTCH2* directly activates *HES1* and *REST* to suppress NE genes⁴.

Mechanism. Briefly, CRACD loss-dysregulated actin polymerization activates the HIPPO signaling that senses mechanical stress^{23,24}, which subsequently phosphorylates and inhibits YAP1. Then, the downregulation of YAP1-transactivated *NOTCH1*²⁵ downregulates the NOTCH signaling, followed by reduced expression of NOTCH1 target genes, *HES1* and *REST*. Finally, NE genes suppressed by HES1 and REST are de-repressed to induce NE cell plasticity (**Fig. R41**).

Supporting data. A. While RPR2 tumors (*Cracd* WT) displayed a distinct nuclear pattern of YAP1 protein expression, CRPR2 (*Cracd* KO) tumor cells barely showed the expression of YAP1 (both nuclear and cytosolic) (**Fig. R42**). We also treated RPR2 cells with verteporfin, a YAP1 inhibitor that disrupts the YAP-TEAD complex. Verteporfin reduced the cleaved Notch1 (N1ICD) protein and its downstream target HES1 (**Fig. R43**). These results support that CRACD influences NOTCH signaling via the actin-YAP1 axis. We have updated the discussion and added these data to the revised manuscript (**Fig. 2l-m, Supplementary Fig. S4**).

Figure R41. Illustration of the mechanism of how CRACD inactivation induces NE cell plasticity, tumor cell heterogeneity, and MHC-I suppression.

Figure R42. Immunohistochemistry of SCLC tumors (RPR2 vs. CRPR2) for YAP1. Scale bars: 50 μm.

Figure R43. Immunoblot analysis of RPR2 cells treated with vehicle or verteporfin 0.5 (+) or 1.0 μM (++) 72 hours after treatment, cells were harvested for immunoblotting.

CRACD is a capping protein inhibitor, promoting actin polymerization². Thus, as pointed out, it is highly likely that CRACD LOF reduces actin polymerization (F-actin) (**Fig. 2k**), which inhibits the HIPPO signaling and subsequently activates YAP1-mediated *NOTCHs* expression, leading to the de-repression of NE genes. This may explain how CRACD LOF induces NE cell plasticity of the NE cell clusters. Indeed, compared to RPR2, CRPR2 tumors exhibited an increased number and cell plasticity of non-NE tumor cells (**Fig. 2f-h**). This might be due to the activation of the YAP1-NOTCH2-REST axis, as described in Yang Ting Shu et al^{26,27}. In the initially submitted manuscript, we observed the marked upregulation of *Hes1*, *Dll1*, *Jag1*, *Notch1*, *Notch2*, and *Notch3* in the non-NE cell clusters of CRPR2 compared to RPR2 (**Supplementary Fig. S3d**). Additionally, we observed the upregulation of *Yap1*, *Taz*, *Tead1/4*, and *Rest* in non-NE cells of CRPR2 tumors compared to RPR2 (**Supplementary Fig. 4b, Fig. R44**). Thus, increased non-NE cell clusters in CRPR2 tumors can be well explained by the activation of the YAP-NOTCH-REST axis. Unfortunately, given that our customized geneset of Xenium did not include those genes, we could not combine scRNA-seq with Xenium in the context of the YAP-NOTCH-REST axis.

Figure 3 (from the initially submitted manuscript)

Figure R44. Feature plots of scRNA-seq datasets of RPR2 and CRPR2 tumors. **A.** Upregulation of the Yap1/Taz-REST axis in non-NE cell clusters of CRPR2 tumors. **B.** Upregulation of Ezh2 mainly in NE cell clusters of CRPR2 tumors, compared to RPR2.

Distinct impact of *Cracd* KO on non-NE and NE cell clusters. As insightfully pointed out, it is somewhat difficult to digest why *Cracd* KO affects both non-NE and NE cell clusters, which is quite perplexing in several aspects. Until scRNA-seq-based analyses, such complicated questions could not be addressed. To challenge this, we examined cell lineage trajectories of RPR2 vs. CRPR2 tumors using the scRNA-seq datasets. First, we found that #10 NE cell cluster is the actively proliferating cell cluster in both RPR2 and CRPR2 tumors (**Fig. 1j**). Intriguingly, cell lineage trajectory analyses using multiple packages showed that, unlike RPR2, *Cracd* KO (CRPR2) increases non-NE (cell clusters #1, 4, 2), which become NE cell clusters (**Fig. 1k-n**). This explains why *Cracd* KO SCLC tumor cells show an increase in both non-NE and NE cell clusters. This is also consistent with the MYC-driven non-NE cell plasticity¹⁹. Our experimental data is further supported by analyzing tumor cell heterogeneity. Unlike such a cell lineage, spatial transcriptomics also confirmed the increased cell plasticity and tumor cell heterogeneity in *Cracd* KO SCLC tumors (CRPR2) (**Fig. 3i, j, and 4**). Next, we checked the expression of all components and target genes of the actin-HIPPO, HIPPO, YAP, and NOTCH pathways from the scRNA-seq datasets of RPR2 and CRPR2 tumors. As mentioned above, the YAP-

Figure R45. Illustration of expression of signaling pathways and target genes. The expression of each pathway component or target gene was visualized based on the scRNA-seq datasets of RPR2 and CRPR2 tumors.

NOTCH signaling axis is activated in only the non-NE cells but not NE cells, represented by specific expression of *Yap1*, *Taz*, *Notchs*, *Hes1*, and *Rest* in non-NE cells (**Fig. R44**). While genes encoding proteins involved in actin-regulated HIPPO signaling did not show differences in their expression, *Ajuba* expression is only detected in non-NE cells (**Fig. R45**). Ajuba protein directly binds to and inhibits LATS1/2, activating YAP. Given its role in inhibiting HIPPO signaling, Ajuba's specific expression in non-NE cells might explain why *Cracd* KO leads to two distinct outcomes (HIPPO activation in NE cells and YAP activation in non-NE cells). Nonetheless, we cannot exclude the involvement of other signaling or TFs because the expression level of *Ajuba* was not markedly higher compared to YAP and NOTCH signaling activation.

These new results and discussion were added to the revised manuscript.

D3. *“What correlations did the authors observe upon analysis of mouse tumor samples using the different methods (scRNA-seq, spatial transcriptomics), in terms of (1) SCLC plasticity/lineage identity and (2) the profile of immune cells within the microenvironment? In other words, are the differences in cellular composition observed with scRNA-seq reproduced in the spatial transcriptomic dataset? A deeper and full analysis of these combined datasets may be the focus of a separate paper altogether, but some superficial-level correlations is warranted for additional rigor in the current study.”*

Xenium In Situ before 5K genesets availability. We agree to these critical comments. As pointed out, we initially planned to combine scRNA-seq with Xenium In Situ to complement each other. Our Xenium experiments were performed with the customized 479 genesets before the release of 5K genesets. With the limited number of genes, we were only able to acquire cell cluster-related information, which was used for analyzing tumor cell heterogeneity (Fig. 2f-h).

Visualizing scRNA-seq information on Xenium. Cell lineages. As suggested, we projected cell clusters and cell proliferation information from scRNA-seq onto Xenium data. Consistent with the scRNA-seq results, we located new non-NE cell clusters (purple) generated in CRPR2 tumors at the core of SCLC tumor nodules. These non-NE cells were surrounded by NE cells (Fig. R46A, B), recapitulating NE cell plasticity of CRPR2 tumors. Additionally, CRPR2 tumor cells are more proliferative than RPR2 tumor cells. Moreover, NE cells are relatively more mitotic than non-NE cells (Fig. R46C). **Immune niche.** We also quantified the T cells infiltrated into SCLC tumors (RPR2 vs. CRPR2). In line with the results from scRNA-seq and immunostaining-based immune profiling (Supplementary Fig. S6), T cells were rarely detected in CRPR2 tumors compared to RPR2 tumors (Fig. R46D). These new results were added to the revised manuscript (Fig. 2 f-h, Supplementary Fig. 6n-o).

Figure R46. Spatial localization and characterization of tumor cell subpopulations and T cells in RPR2 and CRPR2 tumors. **A.** UMAP projection of scRNA-seq data from RPR2 and CRPR2 tumors. Cells are colored by unsupervised clusters; NE and non-NE populations are delineated by dashed lines. **B.** Spatial localization of NE and non-NE clusters in representative RPR2 and CRPR2 tumors. Cluster identities were assigned by correlation-based mapping of Xenium transcriptomic data to matched scRNA-seq clusters. **C.** Spatial mapping of cell cycle states projected onto Xenium coordinates. Cell cycle phase (inferred from matched scRNA-seq data) are shown as: G1 (magenta), S (aqua), and G2/M (purple). **D.** Spatial localization of T cells (Cd3d, Cd3e, Cd3g, Cd8a, Cd8b1, Cd4) within RPR2 and CRPR2 tumors, identified by Xenium transcriptomic signatures. **E.** Quantification of T cells density per unit area from Xenium spatial data.

D4. “Any difference in survival between the qKO vs tKO GEMMs? Given the changes in plasticity, did the authors observe differences in metastasis with qKO vs tKO GEMMs?”

As noted in our response to Reviewer 3 (Comment C9), we used the same time point, six months post-Cre infection, for our analysis. At this stage, we compared tumor burden between qKO vs. tKO GEMMs, although survival analysis was not performed. While not known, it is reasonably assumed that the significantly higher tumor burden in the qKO GEMMs compromises lung function faster and shortens their survival compared to tKO GEMMs. At the six-month time points, we have not found any palpable metastasis in the liver and adrenal glands.

Minor comments

D5. “*Italicize in vitro and in vivo*”
 Edited.

D6. “There was no mention of Fig. S2 in the main text.”

It was mentioned in line 112. We edited the manuscript for better legibility.

D7. “Provide in the main text brief descriptions of computational terms and analytical methods that are heavily referenced such as ‘root cell’, ‘scVelo’, ‘Dynamo’, ‘entropy’, and ‘CellChat’.”

All jargons were spread out with references.

D8. “Line 74: Potential mechanisms underlying SCLC plasticity have been explored in a number of studies such as Ireland et al. (2020). The authors should cite additional relevant literature here.”

As kindly suggested, additional references^{18,19,26} were cited.

D9. “Given the central nature of NOTCH1 signaling and MHC-I downregulation to the mechanism of their current study in SCLC, the authors should provide a brief background in the INTRODUCTION.”

Added as shown below.

“Recent studies have identified NOTCH signaling as a critical regulator of NE plasticity in SCLC. The NOTCH signaling inactivation promotes NE differentiation, whereas its activation drives transitions toward non-NE states, contributing to tumor heterogeneity in SCLC¹⁸. A key mechanism of NOTCH1 suppression in SCLC is the upregulation of DLL3 (Delta-like ligand 3), a direct target of ASCL1, which serves as a negative regulator of the NOTCH pathway²⁸.

In parallel, MHC-I downregulation has emerged as a key mechanism of cancer immune evasion, limiting CTL recognition and reducing the efficacy of ICB therapies. Unlike NSCLC, where genetic alterations frequently drive immune escape, SCLC is characterized by profound defects in antigen processing and presentation pathways²⁹, leading to widespread MHC-I suppression.

Recent evidence has implicated LSD1 (KDM1A) as a key epigenetic regulator of MHC-I repression in SCLC³⁰. LSD1 suppresses antigen presentation through histone modifications that silence the transcription of key MHC-I components³¹. Conversely, pharmacological inhibition of LSD1 using GSK-LSD1 or ORY-1001 has been shown to restore MHC-I expression, enhance antigen presentation, and reinvigorate anti-tumor immunity in preclinical SCLC models³⁰.”

D6. “Can the authors comment on the location of mutations in CRACD that are found in primary tumors? Do these mutations, especially those occurring at a high frequency, reside in functional domains of the protein?”

In the initial submission, we have shown the distribution of CRACD mutations identified in primary SCLC tumors (**Supplementary Fig. 1**). Frame-shift mutations at AA 165 are most frequent in SCLC, which is also observed in other cancer types. Thus, the truncated CRACD mutant likely loses its function (loss-of-function), as previously tested². Additional point mutations were also identified, while no specific mutational hot spots other than nonsense mutations were found. In another manuscript (in preparation), we assessed the role of the various CRACD mutants by ectopic expression of CRACD mutants in CRACD-null cells. CRACD is a capping protein inhibitor, promoting actin polymerization (increased F-actin

Figure R47. Assessing the impact of CRACD point mutations on actin polymerization. MKN45 cells carrying a nonsense mutation in the CRACD alleles were transduced with lentiviruses encoding mCherry-tagged CRACD wild-type (WT) or other point mutants. F-actin was visualized by phalloidin staining.

visualized by phalloidin). However, CRACD point mutations failed to do so (**Fig. R47**). There are no specific functional domains in the CRACD protein except for two capping protein-interacting motifs we previously identified². There is a Glu-rich domain (213–333 amino acids). It is noteworthy that, in addition to genetic alterations, the expression of *CRACD* mRNA is also downregulated in SCLC. We are investigating those point mutations in the context of the CRACD protein structure and dysfunction in another study.

D7. “There are two panels in Fig. S1 labeled with ‘D’”
It was revised. Thank you.

D8. “Line 99: please cite the specific dataset on TCGA the authors used to extract transcriptomic bulk RNA-seq data for healthy tissues and SCLC tumors.”

Bulk transcriptomic profiles of healthy lung tissues and SCLC tumors were downloaded from the TCGA repository (TCGA-SCLC cohort) and processed as described.

Reference: Cancer Genome Atlas Research Network. Comprehensive genomic characterization of squamous cell lung cancers. *Nature* 489, 519–525 (2012)³².

D9. “The authors should label the timepoint at which the lungs were collected & compared.”
The detailed information was added in the Methods. Thank you.

D10. “How did the authors select genes for the analysis of the different signaling pathways (NOTCH, Myc, EMT, etc.)? Why are there only 2 genes for the EMT program?”

Pathway analyses. Our apologies for misleading. For unbiased analysis, we do not pick and choose genes for pathway analysis. We performed GSEA, fGSEA, or pathway analysis using the well-accepted and publicly available genesets. In those figures, due to the limited space, we only showed a few representative genes for signaling or pathways. For clarification, we added the pathway analysis instead of several genes (**Supplementary Fig. S4**).

EMT scoring. For clarification, we included the EMT pathway scores and EMT-related genes (**Fig. R48**). The results remained the same.

Figure R48. Feature plots of scRNA-seq datasets of RPR2 and CRPR2 tumors. **A.** Upregulation of the Yap1/Taz-REST axis in non-NE cell clusters of CRPR2 tumors. **B.** Upregulation of Ezh2 mainly in NE cell clusters of CRPR2 tumors, compared to RPR2.

D11. “The figure legend for Figure. 3d refers to violin plots but there were none.”
Legends were edited accordingly (feature plots only).

D12. “Figure 4D: Why were there so few *t*KO tumors (4) compared to *q*KO tumors (36) in the spatial transcriptomics study? Could this discrepancy have an impact on the results?”

As observed and described in Figure 2, *Cracd* KO significantly accelerates SCLC tumorigenesis. For Xenium In Situ analysis, we used the same FFPE blocks from Figure 3 showing relatively more tumors in CRPR2 than RPR2 (six months post Cre). This is also consistent with additional finding that immune evasion of CRPR2 tumors by suppressing MHC-I (**Fig. 3I**). Therefore, such a discrepancy in tumor development (RPR2 vs. CRPR2) is well in line with overall results.

D13. “What is the mutation frequency in *CRACD* in MS1 vs. MS2 tumors?”

Thank you for the insightful comment. Based on the public datasets, CRACD mutation frequency was 12.9% (32/249) in SCLC patient tumors and 15.7% (16/102) in SCLC cell lines (**Supplementary Fig. S1a-b**). Unfortunately, the exome-seq datasets of those patients were not available, therefore, we were unable to directly compare mutation frequency between MS1 and MS2. However, transcriptomic analysis revealed that CRACD expression was consistently lower in MS1 compared to MS2. To better address this comment, we also performed IHC of SCLC TMA for CRACD and found consistent results (**Fig. R28**). It should be noted that CRACD LOF includes genetic alterations, transcriptional downregulation, and post-translational downregulation. We apologize for misleading. We revised the manuscript by adding new results and editing.

Figure R28. Immunohistochemical staining of CRACD, MHC-I, H3K27Me2, and H3K27Me3 in human SCLC tissues. Tumor sections were grouped based on CRACD expression status, and corresponding changes in MHC-I and histone methylation marks were examined. Quantification was performed using H-score analysis. Statistical significance was assessed by Student's t-test. Data are presented as mean \pm SEM; $P < 0.05$ was considered significant.

D14. “Does the mutational status of CRACD correlate with response to immunotherapy in patients with SCLC?”

This is another very important perspective. However, addressing this comment itself would be another study that needs large cohorts. Moreover, SCLC datasets combined with immunotherapy response are not available to the best of our knowledge. Additionally, considering several options for immune checkpoint inhibitors (anti-PD1, PD-L1, and CTLA-4) as well as patient stratification. Respectfully, this could be beyond the scope of the submitted manuscript.

D15. “Line 469. Authors mean “not with primary tumors”

Revised.

D16. “Any functional assays to evaluate cell motility or migration affected by actin dysregulation? Authors should demonstrate the impact of *Cracd* KO on cell motility/migration to confirm its role in actin dysregulation in the context of SCLC cells.”

As suggested, we performed wound healing and transwell cell migration assays using RPR2 and CRPR2 cells. CRPR2 cells showed increased cell migration and invasion (**Fig. R49**). Of note is that the *Cracd* KO colorectal cancer mouse model showed the invasive phenotype in vitro, as examined by pathologists².

Figure R49. Increased motility and invasiveness of CRPR2 cells. **A.** Representative images of wound healing assays in RPR2 and CRPR2 cells at 0 and 2 days. The wound closure area was quantified and plotted on the right. **B.** Transwell migration and invasion assays of RPR2 and CRPR2 cells. Representative images of migrated cells stained with crystal violet.

Again, we appreciate all insightful and constructive comments.

Reviewer 5

“I co-reviewed this manuscript with one of the reviewers who provided the listed reports. This is part of the Nature Communications initiative to facilitate training in peer review and to provide appropriate recognition for Early Career Researchers who co-review manuscripts.”

References

- 1 Kim, B. *et al.* CRACD loss induces neuroendocrine cell plasticity of lung adenocarcinoma. *Cell Reports* **43** (2024).
- 2 Jung, Y.-S. *et al.* Deregulation of CRAD-controlled cytoskeleton initiates mucinous colorectal cancer via β -catenin. *Nature cell biology* **20**, 1303-1314 (2018).
- 3 Seo, J. & Kim, J. Regulation of Hippo signaling by actin remodeling. *BMB reports* **51**, 151 (2018).
- 4 Shue, Y. T. *et al.* A conserved YAP/Notch/REST network controls the neuroendocrine cell fate in the lungs. *Nature Communications* **13**, 2690 (2022).
- 5 Mahmood, S. R. *et al.* β -actin dependent chromatin remodeling mediates compartment level changes in 3D genome architecture. *Nature Communications* **12**, 5240 (2021). <https://doi.org/10.1038/s41467-021-25596-2>
- 6 Gandhi, V., Plunkett, W. & Cortes, J. E. Omacetaxine: a protein translation inhibitor for treatment of chronic myelogenous leukemia. *Clin Cancer Res* **20**, 1735-1740 (2014). <https://doi.org/10.1158/1078-0432.Ccr-13-1283>
- 7 Ku, S. Y. *et al.* Rb1 and Trp53 cooperate to suppress prostate cancer lineage plasticity, metastasis, and antiandrogen resistance. *Science* **355**, 78-83 (2017).
- 8 Nouri, M. *et al.* Therapy-induced developmental reprogramming of prostate cancer cells and acquired therapy resistance. *Oncotarget* **8**, 18949 (2017).
- 9 Farrell, A. S. *et al.* MYC regulates ductal-neuroendocrine lineage plasticity in pancreatic ductal adenocarcinoma associated with poor outcome and chemoresistance. *Nature communications* **8**, 1728 (2017).
- 10 Porazzi, P. *et al.* EZH1/EZH2 inhibition enhances adoptive T cell immunotherapy against multiple cancer models. *Cancer Cell* **43**, 537-551. e537 (2025).
- 11 Zoma, M. *et al.* EZH2-induced lysine K362 methylation enhances TMPRSS2-ERG oncogenic activity in prostate cancer. *Nature Communications* **12**, 4147 (2021).
- 12 Kim, E. *et al.* Phosphorylation of EZH2 activates STAT3 signaling via STAT3 methylation and promotes tumorigenicity of glioblastoma stem-like cells. *Cancer cell* **23**, 839-852 (2013).
- 13 Rudin, C. M. *et al.* Molecular subtypes of small cell lung cancer: a synthesis of human and mouse model data. *Nature Reviews Cancer* **19**, 289-297 (2019).
- 14 Gay, C. M. *et al.* Patterns of transcription factor programs and immune pathway activation define four major subtypes of SCLC with distinct therapeutic vulnerabilities. *Cancer cell* **39**, 346-360. e347 (2021).
- 15 Chan, J. M. *et al.* Signatures of plasticity, metastasis, and immunosuppression in an atlas of human small cell lung cancer. *Cancer cell* **39**, 1479-1496. e1418 (2021).
- 16 Schaffer, B. E. *et al.* Loss of p130 accelerates tumor development in a mouse model for human small-cell lung carcinoma. *Cancer research* **70**, 3877-3883 (2010).
- 17 Gazdar, A. F. *et al.* The comparative pathology of genetically engineered mouse models for neuroendocrine carcinomas of the lung. *Journal of thoracic oncology* **10**, 553-564 (2015).
- 18 Lim, J. S. *et al.* Intratumoural heterogeneity generated by Notch signalling promotes small-cell lung cancer. *Nature* **545**, 360-364 (2017).
- 19 Ireland, A. S. *et al.* MYC drives temporal evolution of small cell lung cancer subtypes by reprogramming neuroendocrine fate. *Cancer cell* **38**, 60-78. e12 (2020).
- 20 Park, K.-S. *et al.* Characterization of the cell of origin for small cell lung cancer. *Cell cycle* **10**, 2806-2815 (2011).
- 21 Branchfield, K. *et al.* Pulmonary neuroendocrine cells function as airway sensors to control lung immune response. *Science* **351**, 707-710 (2016).
- 22 Dietrich, N. *et al.* REST-mediated recruitment of Polycomb repressor complexes in mammalian cells. *PLoS genetics* **8**, e1002494 (2012).
- 23 Reddy, P., Deguchi, M., Cheng, Y. & Hsueh, A. J. Actin cytoskeleton regulates Hippo signaling. *PloS one* **8**, e73763 (2013).
- 24 Matsui, Y. & Lai, Z.-C. Mutual regulation between Hippo signaling and actin cytoskeleton. *Protein & cell* **4**, 904-910 (2013).
- 25 Totaro, A. *et al.* YAP/TAZ link cell mechanics to Notch signalling to control epidermal stem cell fate. *Nature communications* **8**, 15206 (2017).
- 26 Wu, Q. *et al.* YAP drives fate conversion and chemoresistance of small cell lung cancer. *Science Advances* **7**, eabg1850 (2021).

- 27 Totaro, A., Panciera, T. & Piccolo, S. YAP/TAZ upstream signals and downstream responses. *Nature cell biology* **20**, 888-899 (2018).
- 28 Kim, J. W., Ko, J. H. & Sage, J. DLL3 regulates Notch signaling in small cell lung cancer. *IScience* **25** (2022).
- 29 Roper, N. *et al.* Notch signaling and efficacy of PD-1/PD-L1 blockade in relapsed small cell lung cancer. *Nature communications* **12**, 3880 (2021).
- 30 Nguyen, E. M. *et al.* Targeting LSD1 rescues MHC class I antigen presentation and overcomes PD-L1 blockade resistance in small cell lung cancer. *Journal of thoracic oncology: official publication of the International Association for the Study of Lung Cancer* **17**, 1014 (2022).
- 31 Hiatt, J. B. *et al.* Inhibition of LSD1 with bome demstat sensitizes small cell lung cancer to immune checkpoint blockade and T-cell killing. *Clinical Cancer Research* **28**, 4551-4564 (2022).
- 32 Hammerman, P. S. *et al.* Comprehensive genomic characterization of squamous cell lung cancers. *Nature* **489**, 519-525 (2012). <https://doi.org:10.1038/nature11404>

Responses to Comments

Reviewer 1

“This study investigates the role of CRACD as a tumor suppressor in small cell lung cancer (SCLC), emphasizing its impact on neuroendocrine plasticity, immune evasion, and the potential therapeutic targeting of EZH2. The findings suggest that CRACD depletion facilitates immune evasion through reduction of nuclear actin, EZH2-mediated histone methylation, suppression of MHC-I Genes and depletion of CD8+ T Cells. The downregulation of MHC-I results in a significant decrease in intratumoral CD8+ T cells, impairing immune surveillance and promoting immune evasion.”

Major points

Experimental Design and Data Presentation

A1. *“Figure Repetitions & Sample Size: The manuscript lacks clarity regarding the number of in vivo and in vitro experimental repetitions, as well as the number of mice per experiment. Are the data shown representative or cumulative from multiple experiments? Explicit details on experimental replicates should be provided.”*

Thank you for the comments. In the initially submitted manuscript, n numbers were included in the Figure legends or shown as individual data points in the plots. As suggested, we clarified “n” numbers in the Figure legends (highlighted in the revised manuscript).

A2. *“Figure 1d – Use of Nude Mice: The rationale for using Nude mice needs to be clarified. Did the authors intend to exclude tumor-adaptive immune interactions? A repetition of the experiment in syngeneic wild-type (WT) mice would be more appropriate to establish the role of the adaptive immune system in CRACD depletion from the outset.”*

Yes, we initially sought to determine the cell-autonomous impact of *Cracd* KO. For a distinct study, we employed both immunocompromised and immunocompetent murine models and consistently observed that *Cracd*-depleted preSCs exhibited enhanced tumorigenesis in both Nude and C57BL/7 mice (**Fig. R10**). These results are the subject of another manuscript nearing completion and, therefore, cannot be incorporated into the present revised manuscript. We maintain, however, that the data derived from our GEMMs or tumor cells (RPR2 vs. CRPR2) (**Fig. 1**) adequately support our proposed working model, offering stronger evidence than the preSC cell data. In accordance with prior suggestions, Figure 1 has been relocated to Supplementary Figure 2.

A3. *“Excess Figures: The number of figures is excessive. Consider moving the transplantation model from Figure 1 to the supplementary section and merging Figures 2 and 3 for a more streamlined presentation.”*

Thank you for the suggestion. This manuscript covers two distinct aspects of SCLC tumorigenesis: NE cell plasticity and immune evasion. We believe that Figure 1 is essential to show the impact of CRACD loss-of-function on SCLC tumor initiation and NE plasticity using preSC cells, which we previously generated. Such an early event cannot be addressed by using GEMMs. As suggested, we merged Figure 2 with Figure 3.

Immune Response and Therapeutic Implications

A4. *“Figure 4 – Choice of Spatial Transcriptomics (ST) over scRNA-seq: The authors utilized Xenium for tumor heterogeneity analysis, despite the availability of single-cell RNA sequencing (scRNA-seq) data (Figure 3). Instead of treating scRNA-seq for plasticity and ST for heterogeneity separately, they should compare the two in terms of plasticity and heterogeneity.”*

Figure R10. Tumor growth of *Cracd* KO preSC cells in immunocompromised or immunocompetent mice. PreSC cells (*Cracd* WT vs. KO; derived from C57BL/6 mice) were subcutaneously injected into Nude or C57BL/6 mice. Tumor volumes were measured at the endpoint when any dimension of allograft tumor reaches 1.5 cm or bigger. Tumor growth is defined by tumor volume/days taken to reach the endpoint and is plotted relative to the *Cracd* WT preSCs allograft in nude mice. n=10 per group, Student's *t*-test.

scRNA-seq for analyzing tumor heterogeneity. We respectfully disagree that *scRNA-seq* can assess “intratumoral” heterogeneity. While scRNA-seq, by itself, lacks spatial context within lung tissues and therefore cannot directly quantify intratumoral spatial heterogeneity, it remains a valuable method for characterizing overall “*cellular*” heterogeneity through gene expression-based multi-dimensional visualization and cell clustering. Although Visium and Visium HD offer spatial information, their utility is limited by their sub-optimal cell segmentation, which does not achieve single-cell resolution. Importantly, our study pioneers the calculation of tumor heterogeneity by leveraging molecular-level spatial transcriptomics.

Xenium In Situ for assessing cell plasticity. While Xenium In Situ, based on nucleic acid hybridization, holds theoretical promise for assessing cell plasticity—a capability well-established through scRNA-seq—its current application for the intricate analyses presented here, such as the calculation and visualization of cell plasticity potential (Fig. 3), remains limited. Although the field is actively exploring the integration of single-cell and spatial transcriptomics, these methods have yet to reach the analytical sophistication demonstrated in our manuscript. Furthermore, our Xenium In Situ experiment predates the availability of the expanded 5,000 gene panel. Utilizing a custom 479-gene set, our analysis could not achieve the comprehensive gene coverage and resolution afforded by scRNA-seq, where we typically analyze over 5,000 genes for detailed cell lineage and plasticity studies. Consequently, a comparable cell plasticity analysis using Xenium In Situ could be limited.

Visualizing scRNA-seq information on Xenium. Cell lineages. As suggested, we projected cell clusters and cell proliferation information from scRNA-seq onto Xenium data. Consistent with the scRNA-seq results, we located new non-NE cell clusters (purple) generated in CRPR2 tumors at the core of SCLC tumor nodules. These non-NE cells were surrounded by NE cells (Fig. R11A, B), recapitulating NE cell plasticity of CRPR2 tumors. Additionally, CRPR2 tumor cells are more proliferative than RPR2 tumor cells. Moreover, NE cells are relatively more mitotic than non-NE cells (Fig. R11C). These new results were added to the revised manuscript (Fig. 2f-h).

Figure R11. Spatial localization and characterization of tumor cell subpopulations in RPR2 and CRPR2 tumors. **A.** UMAP projection of scRNA-seq data from RPR2 and CRPR2 tumors. Cells are colored by unsupervised clusters; NE and non-NE populations are delineated by dashed lines. **B.** Spatial localization of NE and non-NE clusters in representative RPR2 and CRPR2 tumors. Cluster identities were assigned by correlation-based mapping of Xenium transcriptomic data to matched scRNA-seq clusters. **C.** Spatial mapping of cell cycle states projected onto Xenium coordinates. Cell cycle phase (inferred from matched scRNA-seq data) are shown as: G1 (magenta), S (aqua), and G2/M (purple).

A5. “Discrepancies in HES1 Expression: A key inconsistency arises between Figure 4j and Supplementary Figure S4a, where CRACD depletion appears to inhibit HES1, while Figure 3d suggests that CRPR2 tumors express more HES1 than RPR2 tumors. How do the authors reconcile this contradiction?”

The apparent discrepancy in HES1 expression between **Supplementary Fig. S4** and **Supplementary Fig. S5** reflects the heterogeneous nature of CRACD-deficient SCLC, which is one of the central findings of our study. In line with this, we recently observed that *Cracd* KO also results in suppression of HES1 expression in LUAD GEMMs¹. Moreover, the CRPR2 tumors exhibit transcriptional heterogeneity, including the emergence of non-neuroendocrine subpopulations with elevated HES1 expression (**Supplementary Fig. S4d**).

This HES1-high non-NE population likely represents a subset of tumor cells undergoing lineage transition, accompanied by active or reactivated Notch signaling. These findings align with prior studies showing that Notch-HES1 activation suppresses neuroendocrine features and promotes a non-NE fate in SCLC¹. This

interpretation is further supported by **Figure 1m**, where CRPR2 tumors exhibit high-entropy, multilineage trajectories including a NOTCH^{high} non-NE cluster, consistent with the HES1^{high} subpopulations in CRPR2 tumors. Therefore, the presence of HES1-positive non-NE cells in CRPR2 tumors supports the notion that CRACD loss facilitates divergent cell fates and contributes to intratumoral heterogeneity. Importantly, such heterogeneity is not merely a molecular observation but is closely linked to tumor plasticity, therapeutic resistance, and disease progression. Thus, rather than indicating a contradiction, the variable HES1 expression across models and tumor subpopulations highlights the context-dependent and dynamic consequences of CRACD inactivation in SCLC.

A6. *“Spatial Organization of Tumor Microenvironment: The spatial analysis does not explore tumor cell organization. Given that Figure 5 suggests tumor escape from CD8+ T cells via MHC-I downregulation, it would be highly relevant to analyze the spatial distribution of T cell infiltrates in CRPR2 versus RPR2 tumors. Specifically: Do CRPR2 tumors generate lymphoid aggregates How do T cells localize relative to cancer cells? Are CD8+ T cells excluded from tumor neighborhoods?”*

Thank you for the comments regarding tumor infiltration, lymphoid aggregation, and CD8+ T cell exclusion. In the initial manuscript, we assessed the spatial distribution of T cell infiltration in CRPR2 and RPR2 tumors. Immunofluorescence staining for CD3 and CD8 showed that CD8+ and CD3+ T cell infiltration was markedly reduced in the tumor core and periphery of CRPR2 tumors compared to RPR2 tumors (**Supplementary Fig. S6e-h**). To address the comments regarding lymphoid aggregates, we repeated the experiments. Consistently, staining results showed the spatial exclusion of T cells in CRPR2 tumors. Notably, unlike RPR2 tumors that occasionally showed lymphoid aggregate-like structures, CRPR2 tumors did not exhibit substantial formation of tertiary lymphoid structures (**Fig. R12**), consistent with immunostaining and scRNA-seq results. New results and additional description were added (**Supplementary Fig. S6i**).

Figure R12. Reduced CD8+ T cell infiltration in CRPR2 tumors compared to RPR2 tumors **A.** Representative immunostaining images of tumor sections from RPR2 and CRPR2 tumors. Top panels show low-magnification views with red boxes indicating regions of interest. Bottom panels show higher-magnification images of the indicated ROIs. Panels labeled (a) and (b) show enlarged views of selected areas, highlighting CD8+ T cell distribution. **B.** Quantification of CD8+ T cell density per unit area in tumor sections. Data represent mean \pm SEM from $n = 3$ tumors per group. Statistical significance was determined by Student's t-test.

A7. *“Figures 6o–p – T Cell Data Presentation: The current approach of presenting T cell infiltration as a percentage of CD45+ cells may obscure effects on CD4+ T cells. T cell infiltration should instead be quantified as the absolute number of T cells per tumor weight or per cancer cell. Reanalysis with these metrics is likely to reveal additional insights into CD4+ T cell responses.”*

As kindly suggested, we reanalyzed the data by quantifying CD4+ T cells based on the total number of FACS events and normalized them by tumor weight (mg) to obtain absolute cell counts (**Fig. R13**). This metric reflects true T cell infiltration regardless of variations in CD45+ cell proportions, directly addressing the potential bias raised. To ensure consistency, we used freshly isolated single-cell suspensions from half

of each tumor (the other half was processed for FFPE) and excluded tumors that were markedly large. Or small to minimize size-related variability. We selected tumors of intermediate and comparable size (mean weight 0.1 ~ 0.2 mg, or 0.5 mg for control). Notably, CD4+ T cell infiltration per tumor volume was increased in the GSK343-treated CRPR2 group compared to the vehicle group or tazemetostat groups. *These results support the conclusion that EZH2 inhibition promotes CD4+ T cell recruitment into CRPR2 tumors.* This discrepancy between GSK343 and tazemetostat may be partially explained by the differences in drug administration route and bioavailability. Tazemetostat was administered orally, which might have resulted in relatively lower local drug effect compared to intraperitoneally delivered GSK343. While tazemetostat reduced tumor burden, its immune-modulatory effects were less pronounced than GSK343, possibly due to reduced issue penetrance or pharmacokinetic limitations. These updated results were added (**Supplementary Fig. S8c**).

Figure R13. Quantification of CD4+ T cells per tumor weight in CRPR2 tumors treated with EZH2 inhibitors. Total number of CD4+ T cells was quantified from freshly isolated tumors and normalized by tumor weight (mg) to account for tumor size differences.

A8. “EZH2 Inhibitor Effects on T Cells: The study would benefit from an expanded FACS analysis of T cell phenotypes following EZH2 inhibitor treatment, including expression of activation markers and intracellular cytokines. Potential synergistic effects between EZH2 inhibitors and immune checkpoint inhibitors (ICIs) would be clinically relevant and should be investigated.”

Thank you for the comments.

FACS and cytokine profiling. As kindly suggested, we treated C57BL/6 mice with GSK343 and analyzed T cells by FACS. The impact of EZH2 inhibitors on T cell activation or cytotoxicity was barely detected (**Fig. R14**). Therefore, no further cytokine profiling using RNA-seq or scRNA-seq was performed.

EZH2 inhibitors with ICIs. We completely agree with this comment because it is challenging to envision that a single agent is sufficient to inhibit tumorigenesis (herein, CRACD-negative SCLC) without relapse, which leads us to consider combination treatment for clinical applications. In animal models, an EZH2 inhibitor itself was sufficient to suppress SCLC tumorigenesis (**Fig. 4**). Therefore, the additional or synergistic impact of ICIs on top of the effect of EZH2 inhibitors cannot be assessed in animal models. Moreover, among ICIs (PD1, PD-L1, and CTLA4), there is no substantial justification for the choice of ICIs. Respectfully, such experiments would be more suitable in clinical settings or at least using immunologically well-defined SCLC PDXs in humanized mouse models, which is beyond the scope of the submitted manuscript, dissecting the biology of SCLC tumor cell plasticity and immune evasion. We added the comments as future studies in the Discussion.

Figure R14. Flow cytometry analysis of T cells following EZH2 inhibitor treatment in C57BL/6 mice.

A. C57BL/6 mice were treated with GSK343, and T cell populations were analyzed by flow cytometry. Representative flow cytometry plots. **B.** Quantification of helper T cells (CD4+ T cells) cytotoxicity-related markers (Perforin, Granzyme B) in CD8+ T cells.

A9. “CD8 T Cell Depletion Experiment: To substantiate the connection between CD8+ T cells and CRPR2 tumor progression, the authors should conduct a CD8 depletion experiment and analyze its effect on tumor growth. This is particularly critical given that CRACD depletion has intrinsic tumor-promoting effects. If CRPR2 and RPR2 tumors show minimal differences in the absence of CD8+ T cells, it would further support the hypothesis that immune evasion drives CRPR2 tumor outgrowth.”

Thank you for the insightful comments. We agree that utilizing CD8+ T cell-depleted mice will be another strong supportive evidence. In the initially submitted manuscript, we observed that, unlike CRPR2, RPR2 cells **barely** developed tumors in immunocompetent mice (C57BL/6) (**Supplementary Fig. S8d-e**), reiterating the immune evasion of CRPR2 cells.

As kindly suggested, we performed CD8 T cell depletion experiments. RPR2 cells do not develop tumors in C57BL/6 mice (**Supplementary Fig. S8d-e**). Conversely, RPR2 cells developed into tumors in CD8+ T cell-depleted immunocompetent (C57BL/6) mice compared to mice injected with IgG (**Fig. R15a, b** lanes 1 and 2), indicating that RPR2 tumor growth is blocked by T cell-related immune surveillance. Moreover, CRPR2 tumor growth in immunocompetent mice did not show a statistically significant difference between the IgG control and CD8 T cell depletion groups (**Fig. R15b**, lanes 3 and 4), which reconfirms our finding that Cracd LOF-induced immune evasion is related to dysfunction of T cell-related immune surveillance.

As expected, the tumorigenicity of CRPR2 is higher than that of RPR2 in C57BL/6 mice (**Fig. R15**). This is well explained by the *intrinsic tumor-promoting effects* of CRACD LOF; CRPR2 cells grew faster than RPR2 cells in vitro or immunocompromised mice (**Fig. R16**). In addition to MHC-I suppression, CRPR2 cells display cell hyperproliferation (**Fig. 3**), cell plasticity (**Fig. 1, 2**), and an increase in cell heterogeneity (**Fig. 2**), which explains why CRPR2 tumors are still bigger than RPR2 tumors in CD8 T cell-depleted C57BL/6 mice.

Therefore, unlike the comment that “*If CRPR2 and RPR2 tumors show minimal differences in the absence of CD8+ T cells*”, it is expected that CRPR2 tumor cells still show a significant difference in cell or tumor growth compared to RPR2 in CD8 T cell-depleted mice.

New results were added (**Supplementary Fig. S8g**).

Figure R15. Tumor growth under CD8+ T cell depletion in C57BL/6 mice.

A. Experimental timeline depicting subcutaneous injection of RPR2 or CRPR2 cells into C57BL/6 mice followed by anti-CD8 antibody administration. **B.** Tumor weights at sacrifice. Each dot represents an individual tumor.

Figure R16. Tumor growth and representative tumor images of RPR2 and CRPR2 cells in immunodeficient nude mice.

A. Tumor growth curves of RPR2 (blue), and CRPR2 (red) tumors measured over 40 days following subcutaneous injection into nude mice. Arrows indicate the median time points of sacrifice for each group. **B.** Representative images of excised tumors at sacrifice from nude mice bearing RPR2 and CRPR2 tumors. Scale bar: 1cm.

A10. “Conclusion. *The study presents compelling findings on CRACD’s role in immune evasion and the therapeutic potential of EZH2 inhibition in SCLC. However, several key points require clarification, including experimental design details, spatial organization of the tumor microenvironment, and the immune response to EZH2 inhibition. Addressing these points will strengthen the manuscript’s conclusions and provide a more comprehensive understanding of CRACD’s function in SCLC.*”

We have revised the manuscript according to your comments.

As kindly suggested, we added new experimental results and comments to the revised manuscript with clarification. Again, we appreciate very constructive and insightful comments.

Reviewer 2

“This manuscript by Seo et al. presents a significant advancement in our understanding of small-cell lung cancer (SCLC) by elucidating the role of CRACD loss in promoting neuroendocrine (NE) plasticity and immune evasion. The study employs rigorous methodologies, including genetically engineered mouse models, single-cell RNA sequencing (scRNA-seq), spatial transcriptomics, and functional validation experiments, to support its claims. The proposed CRACD-EZH2-MHC-I axis as a potential therapeutic target is compelling and has strong translational implications. However, there are critical areas that require further clarification, methodological refinement, and additional data to enhance the manuscript’s impact and scientific rigor.

This study reveals a previously underappreciated role of CRACD in actin polymerization, tumor cell plasticity, and immune evasion. It establishes that CRACD loss drives neuroendocrine plasticity by suppressing NOTCH signaling, increasing tumor heterogeneity, and facilitating immune escape via EZH2-mediated MHC-I suppression. The findings reported that EZH2 inhibition is a novel therapeutic strategy for CRACD-deficient SCLC. By integrating GEMMs, CRISPR-Cas9 knockout models, scRNA-seq, and spatial transcriptomics, the study provides a high-resolution, multi-dimensional analysis of tumor plasticity and heterogeneity. Functional assays further confirm the efficacy of EZH2 inhibitors in restoring MHC-I expression and reversing immune evasion. Notably, the study identifies a distinct SCLC molecular subtype characterized by CRACD loss, EZH2-driven transcriptional repression, and MHC-I downregulation, highlighting a potential biomarker and supporting the therapeutic relevance of EZH2 inhibition in cold SCLC tumors.”

Major points

B1. “The study establishes a connection between CRACD loss and NOTCH signaling downregulation; however, the precise molecular mechanism remains unresolved. Does CRACD directly interact with NOTCH pathway components, or is this effect mediated through actin cytoskeletal disruption?”

Thank you for the valuable comments. We admit that the initially submitted manuscript did not clearly demonstrate how CRACD LOF downregulates NOTCH signaling. Thanks to the comments, we performed additional experiments. Our new data support an indirect regulatory mechanism, in which CRACD loss leads to reduced actin polymerization (**Fig. 2k**), thereby modulating upstream pathways that influence NOTCH activity. Of note, we detected no signaling components of the NOTCH signaling from tandem affinity purification and mass spectrometry (TAP-MS/MS) ².

Mechanism. Briefly, CRACD loss-dysregulated actin polymerization activates the HIPPO signaling that senses mechanical stress ³, which subsequently phosphorylates and inhibits YAP1. Then, the downregulation of YAP1-transactivated *NOTCH1* ⁴ downregulates the NOTCH signaling, followed by reduced expression of NOTCH1 target genes (*HES1* and *REST*). Finally, NE genes suppressed by *HES1* and *REST* are de-repressed to induce NE cell plasticity (**Fig. R20**).

Supporting data. A. While RPR2 tumors (*Cracd* WT) displayed a distinct nuclear pattern of YAP1 protein expression, CRPR2 (*Cracd* KO) tumor cells barely showed the expression of YAP1 (both nuclear and cytosolic) (**Fig. R21**). We also treated RPR2 cells with verteporfin, a YAP1 inhibitor that disrupts the YAP-TEAD complex. Verteporfin reduced the cleaved Notch1 (N1ICD) protein and its downstream target *HES1* (**Fig. R22**). These results support that CRACD influences NOTCH signaling via the actin-YAP1 axis rather than through direct molecular interaction.

We have updated the discussion and added this data to the revised manuscript (**Fig. 2l-m, Supplementary Fig. S4**).

Figure R20. Illustration of the mechanism of how CRACD inactivation induces NE cell plasticity, tumor cell heterogeneity, and MHC-I suppression.

Figure R21. Immunohistochemistry of SCLC tumors (RPR2 vs. CRPR2) for YAP1. Scale bars: 50 μ m.

Figure R22. Immunoblot analysis of RPR2 cells treated with vehicle or verteporfin 0.5 (+) or 1.0 μ M (++). 72 hours after treatment, cells were harvested for immunoblotting.

B2. “While MHC-I suppression is attributed to EZH2-mediated histone methylation, the potential involvement of additional epigenetic regulators (e.g., HDACs, DNMTs) remains unexplored. Chromatin accessibility assays (e.g., ATAC-seq) could provide further mechanistic clarity.”

As kindly suggested, we performed additional experiments to assess whether MHC-I suppression in CRACD-deficient cells may also involve other epigenetic regulators. We treated CRPR2 cells with SAHA (Vorinostat, 2.5 μ M for 48 hours), an HDAC inhibitor that prevents chromatin compaction, and 5-aza-deoxycytidine (5-Aza-dC, Decitabine, 1 μ M for 72 hours), a DNA methyltransferase inhibitor that induces DNA hypomethylation, and assessed MHC-I expression by immunoblotting (**Fig. R24**). We observed no effects of such inhibitors on MHC-I de-repression.

We agree that a combined RNA-seq and ATAC-seq approach would yield a broader and less biased genome-wide view. Nevertheless, we respectfully disagree with the assertion that ATAC-seq directly elucidates the mechanisms of epigenetic regulation. While ATAC-seq effectively profiles chromatin accessibility, its primary output is information on the openness of genomic regions, rather than the detailed molecular processes underlying epigenetic gene expression.

Figure R24. Expression of MHC-I in CRPR2 treated with vehicle, SAHA (HDAC inhibitor), 4-Aza-dc (DNMT inhibitor), or the combination of SAHA and 5-Aza-dC. RPR2 cells were used as a positive control.

B3. “The interplay between NOTCH signaling downregulation and EZH2-driven immune evasion is briefly mentioned but not thoroughly investigated. Could NOTCH signaling directly regulate EZH2 recruitment?”

Thank you for the comments. β -Actin (*Actb*) KO induces EZH2-mediated gene repression⁵. Consistent with these studies, in the initially submitted manuscript, we proposed that nuclear actin dysregulation by CRACD LOF is a key event inducing EZH2-mediated suppression of genes (**Fig. 4**), including encoding the MHC-1 protein. As suggested, it is also possible that inactivation of NOTCH signaling might trigger EZH2-mediated gene repression. Several studies have shown EZH2’s regulatory mechanisms of the NOTCH signaling, but not vice versa. In our data, EZH2 inhibitors de-repress MHC-I (**Fig. 4g**). However, in CRPR2 cells, where NOTCH signaling is downregulated and EZH2 occupies the MHC-I promoter (**Fig. 4d-e**), NOTCH signaling activation by ectopic expression of N1ICD does not affect MHC-I expression (**Fig. 4j**). These data suggest that NOTCH signaling does not regulate the MHC-I gene promoter activation, excluding the involvement of NOTCH signaling in inhibiting EZH2’s promoter occupancy. Thus, CUT&RUN for EZH2 using CRPR2 cells (vector control vs. N1ICD) is unlikely to be informative. As suggested, the revised manuscript discussion now includes better clarification (highlighted; lines 366-370).

B4. “The observed depletion of intratumoral CD8+ T cells in CRACD-deficient tumors may be influenced by additional immunosuppressive populations, such as myeloid-derived suppressor cells (MDSCs) and regulatory T cells (Tregs). A more comprehensive immune profiling using flow cytometry or single-cell analysis is

required.”

Figure R25. Single-cell transcriptomic analysis of T cells reveals a CRPR2-enriched inflammatory subpopulation. (a) UMAP visualization of 5,152 T cells from RPR2 and CRPR2 tumors, colored by cluster identity. Pie charts indicate the cluster composition within each sample. (b) Bar plot showing cluster proportions and dot plot displaying average expression of major T cell subtype markers across clusters. Notably, CD8⁺ T cell subclusters were selectively depleted in CRPR2, whereas cluster 7, enriched in CRPR2, exhibited high expression of *Foxp3*, indicative of regulatory T cells. Inset violin plot illustrates elevated *Foxp3* expression in cluster 7. (c) Heatmap of differentially expressed genes within cluster 7, comparing CRPR2 and RPR2 T cells. CRPR2 cluster 7 cells showed upregulation of immunoregulatory genes including *Tnfrsf18*, *Tff1*, *Ctla2a*, and *Ahr*. (d) Violin plots showing significantly higher expression of *Tnfrsf18*, *Tff1*, *Ctla2a*, and *Ahr* in CRPR2 cluster 7 cells compared to RPR2. Statistical analysis was performed using the Wilcoxon rank-sum test; ***P* < 0.01, ****P* < 0.001.

Thank you for this insightful comment. To address the potential contribution of immunosuppressive populations to CD8⁺ T cell depletion, we analyzed Tregs (Cd4⁺ Ikkzf2⁺) and MDSC-like myeloid cells (Itgam⁺ Cd14⁺ Clec4e⁺ Arg2⁺) based on our scRNA-seq data. Treg populations were readily detectable and quantifiable, and CRACD-deficient tumors exhibited an increased abundance of these cells (**Fig. R25**). Of note, a distinct MDSC subcluster was not clearly separable, likely due to the limited number of immune cells and overlapping marker expression. These new results were added (**Fig. 3c-f**).

B5. “While the study suggests that restoring MHC-I expression counteracts immune evasion, it does not confirm whether this results in functional CD8⁺ T cell activation. In vivo, cytotoxicity assays and cytokine profiling would strengthen this conclusion.”

Although in vivo cytotoxicity assays were not feasible in our current setting, we performed alternative in vitro experiments to assess functional activation of CD8⁺ T cells. (**Fig. R26**).

CD8⁺ T cells were co-cultured with CRPR2 or RPR2 tumor cells, and intracellular expression of Perforin and Granzyme B was assessed by flow cytometry. As expected, CRPR2 tumors showed significantly reduced frequencies of Perforin⁺ CD8⁺ T cells, which were partially restored upon EZH2 inhibitor (GSK343 and Tazemetostat) treatment. These results directly demonstrate functional restoration of cytotoxic CD8⁺ T cell activity via MHC-I re-expression.

Although Granzyme B⁺ CD8⁺ T cells also showed an increasing trend with EZH2 inhibitor treatment, the difference did not reach statistical significance. This may reflect distinct activation thresholds and temporal kinetics between effector molecules—Perforin being induced earlier during activation, while Granzyme B may require stronger or prolonged stimulation. Additionally, the limited stimulatory context of the co-culture assay may have constrained full Granzyme B induction. Nonetheless, the recovery of Perforin⁺ CD8⁺ T cells provides functional evidence that MHC-I

Figure R26. Functional restoration of cytotoxic CD8⁺ T cells upon EZH2 inhibitor treatment in vitro.

A. Representative flow cytometry plots showing intracellular Perforin and Granzyme B expression in CD8⁺ T cells co-cultured with RPR2 and CRPR2 cells, with or without EZH2 inhibitor treatment. **B.** Quantification of Perforin⁺ and Granzyme B CD8⁺ T cell frequencies.

restoration effectively counteracts immune evasion in CRPR2 tumors. New results were added (**Supplementary Fig. 8f**).

B6. “EZH2 inhibition is shown to suppress tumor growth, but its effect on overall survival remains unexamined. Kaplan-Meier survival curves should be included to assess therapeutic impact.”

In this study, we used a subcutaneous tumor model to assess the anti-tumor efficacy of EZH2 inhibitors. In all our animal experiments, death is not an endpoint. Animals were euthanized according to IACUC guidelines when control tumors reached 2 cm in diameter. Therefore, overall survival could not be assessed in this setting. Nonetheless, we showed all individual data points/lines in our tumor quantification plots without excluding any data (i.e., potential outliers) (**Fig. 4l** and **Supplementary Fig. 8b-d**). We agree that Kaplan-Meier survival analysis would be informative in a systemic or orthotopic model. However, given the clinical availability of an EZH2 inhibitor (tazemetostat), KM plotting in clinical studies might be more informative than those from preclinical models, including syngeneic transplantation models.

B7. “Given SCLC’s heterogeneity, other epigenetic vulnerabilities in CRACD-low tumors may exist. Identifying potential co-targets with EZH2 could refine therapeutic strategies.”

Thank you for the comments. Aside from the results from preclinical models, considering the heterogeneity of CRACD inactivated SCLC, there may be additional epigenetic vulnerabilities that could serve as secondary weak points for overcoming therapy resistance or relapse.

Figure R27. GSEA plots showing KEGG_RIBOSOME pathway enrichment in CRPR2 versus RPR2 cells within clusters 5, 6, 7, 9, and 10. Only clusters with adjusted $p < 0.05$ are shown. NES and P -values are indicated.

For instance, while our study focused on EZH2 as a key effector of immune evasion, combinatorial targeting strategies involving EZH2 and other epigenetic regulators may further enhance therapeutic efficacy. As kindly suggested, we performed pathway analyses from scRNA-seq datasets of RPR2 vs. CRPR2 tumors. We did not limit our analysis for identifying “other epigenetic vulnerabilities” since there are only a few in FDA-approved drugs targeting epigenetic regulators (e.g., DNMTs, HDACs, and EZH2). Our analysis identified ribosome signaling pathways as significantly activated in CRPR2 compared to RPR2 (**Fig. R27**). Notably, several FDA-approved drugs, such as homoharringtonine (omacetaxine mepesuccinate), a protein translation inhibitor targeting the ribosome, are clinically available and have been used in hematologic malignancies⁶. While these findings suggest that ribosome-targeting agents may represent potential co-targets with EZH2 inhibition, this hypothesis requires further experimental validation and is beyond the current scope of this manuscript. We have briefly incorporated this perspective into the Discussion (highlighted; lines 394-402). Thank you for the insightful comment.

B8. “The scRNA-seq findings are compelling, but additional validation at the protein level is essential. Western blot or immunofluorescence for key markers identified in human SCLC samples would strengthen the conclusions.”

Thank you for the insightful comments. As suggested, we performed IHC on human SCLC samples to validate the expression of key markers (CRACD, EZH2-mediated histone modification [H3K27Me2/3], and MHC-I) identified from our studies. The IHC results confirmed differential expression of representative markers at the protein level, thereby supporting the transcriptomic findings. Briefly, CRACD downregulation is correlated with MHC-I downregulation, whereas it is inversely correlated with EZH2 histone markers (H3K27Me2/3) (**Fig. R28**). New results were added (**Supplementary Fig. 11a-b**).

Figure R28. Immunohistochemical staining of CRACD, MHC-I, H3K27Me2, and H3K27Me3 in human SCLC tissues. Tumor sections were grouped based on CRACD expression status, and corresponding changes in MHC-I and histone methylation marks were examined. Quantification was performed using H-score analysis. Statistical significance was assessed by Student's t-test. Data are presented as mean \pm SEM; $P < 0.05$ was considered significant.

B9. “Certain statistical comparisons require greater transparency regarding methodology. The manuscript should clarify whether multiple testing corrections were applied in pathway analyses.”

We thank the reviewer for this helpful comment. As suggested, we have clarified in the Methods section that multiple testing correction was applied in pathway enrichment analyses. Specifically, GSEA was performed using the *fgsea* R package with 2,000 permutations, and adjusted *p*-values (*padj*) were calculated using the Benjamini–Hochberg method to control the false discovery rate (FDR). This information is now explicitly stated in the revised Methods section under “Gene set enrichment analysis (GSEA)” (highlighted).

B10. “The study does not compare EZH2 inhibition with current immunotherapies, such as PD-1/PD-L1 blockade. Could combination therapy enhance therapeutic efficacy? Does it provide an added advantage as combination therapy or an alternative to existing therapies?”

EZH2i + ICIs. We agree that combination therapy with immune checkpoint inhibitors (ICIs) could be a rational strategy to enhance therapeutic efficacy, particularly given the likelihood of relapse with monotherapy in CRACD-deficient SCLC. In our current study, we did not carry out combination therapy for the following reasons. First, *EZH2 inhibition alone was sufficient to suppress tumor growth in animal models.* Therefore, evaluating the effect of combination therapy (EZH2i + ICIs) was *not feasible in our preclinical settings.* Second, the choice of specific ICIs lacks experimental justification in our mouse models. Third, although using human SCLC cells or PDXs is more pathologically relevant, especially to immunotherapies, humanized mice cannot fully recapitulate the human immune landscape.

Perspective. Yes, we strongly agree that combination therapy would enhance therapeutic efficacy in clinical applications, which will improve current immunotherapy strategies by detailed patient stratification, e.g., CRACD expression and MHC-I expression in addition to the expression of immune checkpoint molecules. The limited efficacy of monotherapy in overcoming resistance or relapse is a well-established challenge in clinical trials. In contrast, our preclinical models indicate that EZH2 inhibition alongside ICIs could offer a compelling solution, especially for patients whose tumors exhibit CRACD loss-of-function (LOF). Our study proposes a biomarker-guided strategy, integrating targeted EZH2 inhibition and immunotherapy (ICIs), with the potential to overcome the shortcomings of current ICI-based treatments. Such information was added to the Discussion. Thank you.

Minor points

B11. “To enhance translational impact, the graphical abstract should more explicitly emphasize the therapeutic relevance of the CRACD-EZH2-MHC-I axis.”

As suggested, we have revised the graphical hypothesis (Fig. R29, Fig. 5j).

Figure R29. Graphical abstract

B12. *“The discussion would benefit from a broader integration of findings from similar studies on tumor plasticity in other malignancies, such as prostate and pancreatic neuroendocrine tumors.”*

As suggested, we have expanded the discussion section to include relevant findings from studies on tumor plasticity in prostate^{7,8} and pancreatic neuroendocrine tumors to provide a broader context⁹. (highlighted)

B13. *“Additional references on EZH2 inhibitors in SCLC and other cancers would provide a more comprehensive contextualization of the therapeutic approach.”*

As recommended, we have included an additional reference¹⁰ (Highlighted)

B14. *“Certain sections, particularly within the results, contain dense and complex sentences that could be streamlined for clarity and conciseness.”*

The writing editor (Christine F. Wogan) at MD Anderson Cancer Center revised the results for better legibility.

B15. *“The term “immune evasion” is used broadly throughout the manuscript. Distinguishing between specific mechanisms, such as T cell exclusion versus antigen presentation loss, would improve precision in interpretation.”*

As kindly suggested, we have refined the description of immune evasion throughout the manuscript by specifying distinct mechanisms such as T cell exclusion and antigen presentation loss where appropriate (highlighted).

“In summary, this study provides a compelling contribution to SCLC research by identifying CRACD as a key tumor suppressor that constrains NE plasticity and immune evasion. The mechanistic insights into actin cytoskeletal regulation, NOTCH signaling suppression, and EZH2-driven epigenetic remodeling are highly valuable. However, the study lacks a clear mechanistic link between CRACD, NOTCH, and EZH2. Additionally, expanding immune profiling to better characterize the tumor microenvironment, functionally validating the impact of EZH2 inhibition on CD8+ T cell activation and cytotoxicity, and assessing survival outcomes while exploring combinatorial therapeutic strategies will further strengthen the findings. Addressing these critical points will enhance the manuscript’s scientific rigor.”

We appreciate all critical comments that improve the manuscript.

Reviewer 3

“This manuscript, written by Seo et al., describes the role of the actin de-capping inhibitor, CRACD, in small cell lung cancer (SCLC). CRACD promotes the actin polymerization, and is thereby involved in the plasticity and immune reactivity of SCLC. The authors conducted the single cell RNA sequencing (scRNA seq) analysis and the spatial analysis (Xenium in situ hybridization) for the mouse model of SCLC, in which RB1, TP52 and RBL2 were knocked out (RPR2 KO mice). Using this mouse model, they found that the CRCD transforms the pre-neoplastic cells into SCLC tumor-like cells, including the compaction of the cells. CRACD also induces neuroendocrine (NE) plasticity and increases the tumor cell heterogeneity via dysregulation of NOTCH1 pathway. Even more importantly, they observed that CARCS induced the EZH2-mediated H3K27 trimethylation by the nuclear actin reduction. This epigenomic change suppresses the expression of the MHC-I genes. As a result, the recruitment of cytotoxic T cells (CTLs) are reduced. As an attempt to utilize these findings for developing a new drug treatment, the authors treated the CRCD KO-RPR2 mice with an EZH2 inhibitor. They found that the MHC-I expression and the immune surveillance were actually restored. Even though further analyses directly using human material may be needed, this study paves the first step towards the combination therapy of the ICB and EZH2 inhibitor to this difficult cancer species. Overall, I think this is the well-written paper, supported by the solid and a wide variety of experimental evidence. The proposed perspective to utilize the EZH2 is timely and the one which a large number of the patients having SCLC are looking forward to.”

Major points

C1. *“While I fully understand that the scope of this particular paper should lie in the study of the pre-clinical mouse model, I’d still like to request the authors to conduct the extensive analysis to demonstrate the relevance of the obtained results directly using the human material, at least, to some extent. The presented data is, after all, from the mouse model and does not always indicate the in vivo relevance. In fact, the molecular mechanisms underlying the lung cell lineage and the cancer development are supported to occasionally quite different from humans, even though superficial appearance may look similar.”*

Thank you for the constructive comments. We agree that analyzing more human patient samples will strengthen our findings. Unfortunately, we had to admit that dissecting “the molecular mechanisms underlying the lung cell lineage and the cancer development using human samples” is barely feasible. Using SCLC PDOs might be the only option, which has not been greatly successful. Moreover, the scRNA-seq datasets from human SCLC tumor samples are very limited. As shown in Figure 5, we performed extensive analysis using the previous scRNA-seq datasets of SCLC patients’ tumors (n=19), showing the relevance of CRACD status to MHC-I downregulation and NOTCH signaling pathways.

As kindly suggested, we conducted additional analysis by IHC of SCLC tumor microarray and observed that CRACD downregulation is correlated to the loss of MHC-I, the elevated H3K27me2/3. Conversely, in CRACD-high SCLC tumors, MHC-I expression was elevated, while H3K2me2/3 levels were relatively reduced (**Fig. R31**). New results were added (**Supplementary Fig. 11a-b**).

Figure R31. Immunohistochemical staining of CRACD, MHC-I, H3K27Me2, and H3K27Me3 in human SCLC tissues. Tumor sections were grouped based on CRACD expression status, and corresponding changes in MHC-I and histone methylation marks were examined. Quantification was performed using H-score analysis. Statistical significance was assessed by Student's t-test. Data are presented as mean \pm SEM; $P < 0.05$ was considered significant.

C2. “Results of the Xenium analysis is not fully associated with those of the scRNA analysis. The spatial locations of the respective single cells should be presented. Also, I assume the data from the 379 gene Mouse Tissue Atlas Panel plus 100 gene custom should not be so much comprehensive. In addition to showing just the cellular composition or the tumor heterogeneity, please show which parts of the single cells profiles are directly represented in the Xenium data.”

Thank you for the comments. As suggested, we visualized the scRNA-seq information (Figure 3) on the Xenium results.

Xenium In Situ before 5K genesets availability. We initially planned to combine scRNA-seq with Xenium In Situ to complement each other. Our Xenium experiments were performed with the customized 479 genesets before the release of 5K genesets. With the limited number of genes, we were only able to acquire cell cluster-related information, which was used for analyzing tumor cell heterogeneity (**Fig. 2**).

Visualizing scRNA-seq information on Xenium. Cell lineages. As suggested, we projected cell clusters and cell proliferation information from scRNA-seq onto Xenium data. Consistent with the scRNA-seq results, we located new non-NE cell clusters (purple) generated in CRPR2 tumors at the core of SCLC tumor nodules. These non-NE cells were surrounded by NE cells (**Fig. R32A, B**), recapitulating NE cell plasticity of CRPR2 tumors. Additionally, CRPR2 tumor cells are more proliferative than RPR2 tumor cells. Moreover, NE cells are relatively more mitotic than non-NE cells (**Fig. R32C**). **Immune niche.** We also quantified the T cells infiltrated into SCLC tumors (RPR2 vs. CRPR2). In line with the results from scRNA-seq and immunostaining-based immune profiling (**Supplementary Fig. S6**), T cells were rarely detected in CRPR2 tumors compared to RPR2 tumors (**Fig. R32D, E**). These new results were added to the revised manuscript (**Fig.2f-h, Supplementary Fig. S6i-k**).

It should be noted that *Cracd* KO itself is sufficient to upregulate NE and SCLC markers in lung organoids and GEMMs¹ (**Fig. R32**). Therefore, regardless of RBL2 LOF, it is highly likely that CRACD loss induces NE cell plasticity, promoting SCLC tumorigenesis in a cell-autonomous manner. Similarly, we recently found that *Cracd* KO upregulates NE genes in GEMMs and lung organoids¹.

Figure R32. Spatial localization and characterization of tumor cell subpopulations and T cells in RPR2 and CRPR2 tumors. **A.** UMAP projection of scRNA-seq data from RPR2 and CRPR2 tumors. Cells are colored by unsupervised clusters; NE and non-NE populations are delineated by dashed lines. **B.** Spatial localization of NE and non-NE clusters in representative RPR2 and CRPR2 tumors. Cluster identities were assigned by correlation-based mapping of Xenium transcriptomic data to matched scRNA-seq clusters. **C.** Spatial mapping of cell cycle states projected onto Xenium coordinates. Cell cycle phase (inferred from matched scRNA-seq data) are shown as: G1 (magenta), S (aqua), and G2/M (purple). **D.** Spatial localization of T cells (Cd3d, Cd3e, Cd3g, Cd8a, Cd8b1, Cd4) within RPR2 and CRPR2 tumors, identified by Xenium transcriptomic signatures. **E.** Quantification of T cells density per unit area from Xenium spatial data.

C3. “To me, the highlight of this paper is the use of an EZH2 inhibitor. On the other hand, I think the current EZH2 inhibitors should invoke substantial effects on the epigenome statuses of indirect target sites (as a secondary effect) or on indirect target genes other than EZH2. In addition to the MHC-I genes, the sites where the relevant changes were observed should be further scrutinized. Also, I wonder if some key cancer gene regulations may be also influenced?”

Despite the promising outcome of EZH2 blockade in preclinical models, we understand the possible off-target and/or secondary effects of EZH2 inhibitors beyond MHC-I. Also, EZH2 has been shown to promote tumorigenesis by methylating non-histone proteins^{11,12}. To address this, we performed bulk RNA-seq of CRPR2 cells treated with EZH2 inhibitors ($n=3$ per group). Of note, compared to scRNA-seq, bulk RNA-seq provides relatively deeper sequencing reads. It was reproducible to see the de-expression of MHC-I transcripts in CRPR2 cells treated with EZH2 inhibitors, along with the upregulation of KRAS signaling and EMT, and the downregulation of MYC targets, DNA repair, DNA replication, ribosome biogenesis, and RNA metabolic processes from GSEA (Fig. R33). Thus, in addition to EZH2i, using Kras inhibitors, PPARi, or HHT could be considered. New results were added (Supplementary Fig. S9a-d). Respectfully, we believe that further investigation of additional EZH2 targets and their inhibitors is out of the scope of this manuscript.

It is noteworthy that, despite the additional impact of EZH2 inhibitors on such signaling pathways, we strongly believe that the anti-tumorigenic effect of EZH2 inhibitors is mainly due to the changes related to immunosurveillance, including antigen processing and presentation. This is because, unlike RPR2 cells, CRPR2 cells develop tumors relatively well in immunocompetent mice (C57BL/6) (Supplementary Fig. 8d-e), which is markedly suppressed by EZH2 inhibitors (Fig.4I-n).

Minor points

C4. “I’m not totally sure how the obtained results should be interpreted in the context of the previous molecular subtypes (ASCL1, NeuroD1, POU2F3 and YAP1) and the neuroendocrine (NE) or non-neuroendocrine subtypes. Sometimes, those particular subtypes are associated with inflammatory subtypes, to which immune check point blockage (ICB) should be effective. However, as mentioned by the authors themselves, these classifications are not always consistent with clinical phenotypes (or drug responses) of the patients. Honestly,

I'm not familiar with this mouse model. In fact, during the review. I firstly found a quite frequently employed model for this purpose. Therefore, I wonder what subtype this mouse is modeling (also related to the point 1). Please carefully discuss the limit of the present analysis."

We thank the reviewer for raising this important point.

Subtypes. ANPY subtypes were first introduced based on the "bulk" RNA-seq¹³⁻¹⁵. Later, the inflammatory subtype was claimed based on the PDXs¹⁵. Thus, directly comparing the information from bulk RNA-seq with scRNA-seq (tumor cells only in this study) might not be informative. For example, despite the popularity of ANPY subtyping, scRNA-seq datasets do not show such distinct subtypes (except for the POUF1 subtype) (**Fig. 5d**).

GEMMs. RPR2 GEMMs have been extensively used^{16,17}, displaying the feature of A subtype SCLC (ASCL1). Our scRNA-seq analysis of human SCLC tumors revealed that CRACD-low tumors (MS1) belong to the ASCL1 subtype, while CRACD-high tumors (MS2) encompass all four ANPY subtypes (**Fig. 5d**), which supports our rationale for using RPR2 to study CRACD inactivation. We have addressed this point in the revised manuscript.

Tumor cell heterogeneity. Most SCLC preSCs and GEMMs presented in this study exhibit the NE features. Not surprisingly, many groups, including us, have identified dynamic cell plasticity, a conversion between non-NE cells to NE cells in SCLC^{18,19}, along with EMT gene expression as well as Myc expression. Our results showed such cell lineages and cell plasticity at single-cell levels for the first time (**Fig. 2f-h**). Intriguingly, our recent study also showed that *Cracd* KO induces NE cell plasticity of lung adenocarcinoma¹. It is noteworthy that the outcome of CRACD LOF is the loss of epithelial integrity, which is similar to EMT.

Together, despite the conventional SCLC subtypes and classical pathological classification, single-cell analysis indicates dynamic and heterogeneous features of SCLC. *Nonetheless, we fully appreciated the limitation of the RPR2 preclinical model and its possible incompatibility with human tumors.* As kindly suggested, such information and limitations were further discussed (highlighted). Thank you.

C5. *"Relatedly, in human cancers, the RBL2 mutation does not always co-occur with the CRACD mutation. I wonder how the neoplastic phenotype which appears in the mouse model without the RBL2 mutation should be further connected to the results obtained from the RPR2 mouse model. For this purpose, I'd like to see the spatial analysis of the preSC mice. At least, please include the comparison analysis on the spatial characteristics between the RPR2 and CRPR2 mice."*

Thank you for another insightful comment. *Rbl2* KO is often used for GEMMs since it accelerates SCLC tumorigenesis. Without *Rbl2* KO, it takes around 12 months to develop SCLC tumors in RP mice. In our understanding, the reviewer's comment is to assess the impact of *Cracd* KO on SCLC tumorigenesis in the absence of *Rbl2* KO by running Xenium in preSC mice. Unfortunately, we do not have a budget (14K) for additional Xenium of preSC tumors (subcutaneous transplantation), which could still be inferior to GEMMs. Alternatively, as suggested, we further compared Xenium data with scRNA-seq data (RPR2 vs. CRPR2). Please see the response to C3.

C6. *"Innocently, I also wonder, in this mouse model, where in the lung and what cell type the preSC cells and LCSC cells are originating from."*

The primary cells of origin for SCLC are known to be lung neuroendocrine (NE) cells²⁰. This cell type accounts for approximately 0.5% of all lung epithelial cells and expresses a number of neuroendocrine markers²¹, including ASCL1, CGRP, SYP, and CHGA. PreSC cells and mouse SCLC cells continue to express the same markers, which help define them as neuroendocrine cells. Nonetheless, we do not entirely exclude any engagement of NE cell plasticity in generating NE tumor cells from non-NE cells, as we recently reported¹.

C7. *"The interpretation of the Notch-Delta signaling status is not clear to me in the context of the present paper. This axis is pivotal to the transition from NE to non-NE subtypes, as often referenced (even though it*

may not be the case of SCLC; Yan Ting Shu et al, Nat Com 2022, for example). There, REST is also involved, which might be also involved with the EZH2 inhibition. In the single cell data, I wonder if the whole picture of the regulatory network may be represented for the possible feed-back or cross-talk of the pathways (which may be also, as least to some extent, represented by the Xenium data).”

Thank you for sharing another valuable insight and literature. Briefly, our findings are somewhat consistent with the Julien Sage lab’s report, i.e., the YAP-NOTCH-HES1/REST axis inhibits NE gene expression⁴ and findings from the Trudy Oliver lab., MYC-driven non-NE cell plasticity with YAP and NOTCH signaling activation^{4,19}.

YAP-NOTCH-REST axis in inhibiting NE cell plasticity. In addition to the results in this study, our recent paper¹ and a manuscript in revision showed that CRACD LOF consistently inhibits the NOTCH signaling, leading to the activation of the secretory cell lineages, including mucinous or neuroendocrine ones, depending on the context. Yang Ting Shu et al. showed that YAP1-transactivated NOTCH2 directly activates HES1 and REST to suppress NE genes⁴.

Mechanism. Briefly, CRACD loss-dysregulated actin polymerization activates the HIPPO signaling that senses mechanical stress³, which subsequently phosphorylates and inhibits YAP1. Then, the downregulation of YAP1-transactivated NOTCH⁴ downregulates the NOTCH signaling, followed by reduced expression of NOTCH1 target genes, HES1 and REST. Finally, NE genes suppressed by HES1 and REST are de-repressed to induce NE cell plasticity (Fig. R35).

Supporting data. A. While RPR2 tumors (Cracd WT) displayed a distinct nuclear pattern of YAP1 protein expression, CRPR2 (Cracd KO) tumor cells barely showed the expression of YAP1 (both nuclear and cytosolic) (Fig. R36). We also treated RPR2 cells with verteporfin, a YAP1 inhibitor that disrupts the YAP-TEAD complex. Verteporfin reduced the cleaved Notch1 (N1ICD) protein and its downstream target HES1 (Fig. R37). These results support that CRACD influences NOTCH signaling via the actin-YAP1 axis. We have updated the discussion and added these data to the revised manuscript (Fig. 2, Supplementary Fig. S4).

Figure R35. Illustration of the mechanism of how CRACD inactivation induces NE cell plasticity, tumor cell heterogeneity, and MHC-I suppression.

Figure R36. Immunohistochemistry of SCLC tumors (RPR2 vs. CRPR2) for YAP1. Scale bars: 50 μm.

Figure R37. Immunoblot analysis of RPR2 cells treated with vehicle or verteporfin 0.5 (+) or 1.0 μM (++) 72 hours after treatment, cells were harvested for immunoblotting.

Visualizing the YAP-NOTCH-REST axis in scRNA-seq.

CRACD is a capping protein inhibitor, promoting actin polymerization². Thus, as kindly pointed out, it is highly likely that CRACD LOF-reduced actin polymerization (F-actin) (**Fig. 4h**) inhibits the HIPPO signaling and subsequently activates YAP1-mediated *NOTCHs* expression, leading to the de-repression of NE genes. This may explain how CRACD LOF induces NE cell plasticity of the NE cell clusters. Indeed, compared to RPR2, CRPR2 tumors exhibited an increase in the number and cell plasticity of non-NE tumor cells (**Fig. 2d-e**). This might be due to the activation of the YAP1-NOTCH2-REST axis, as described in Yang Ting Shu et al⁴. In the initially submitted manuscript, we observed the marked upregulation of *Hes1*, *Dll1*, *Jag1*, *Notch1*, *Notch2*, and *Notch3* in the non-NE cell clusters of CRPR2 compared to RPR2 (**Supplementary Fig. S4d**). Similarly, *Yap1*, *Taz*, *Tead1/4*, and *Rest* were upregulated in non-NE cells of CRPR2 tumors compared to RPR2 (**Supplementary Fig. S4b-c, R39**). Thus, increased non-NE cell clusters in CRPR2 tumors can be well explained by the activation of the YAP-NOTCH-REST axis.

Unfortunately, given that our customized geneset of Xenium did not include those genes, we could not combine scRNA-seq with Xenium to spatially visualize the YAP-NOTCH-REST axis.

Figure 3 (from the initially submitted manuscript)

Figure R39. Feature plots of scRNA-seq datasets of RPR2 and CRPR2 tumors. **A.** Upregulation of the Yap1/Taz-REST axis in non-NE cell clusters of CRPR2 tumors. **B.** Upregulation of *Ezh2* mainly in NE cell clusters of CRPR2 tumors, compared to RPR2.

YAP-NOTCH-REST axis in EZH2-mediated MHC-I suppression of *Cracd* KO tumors. Given that REST recruits EZH2 (PRC2) to repress gene expression via H3K27me2/3²², it is possible that EZH2-induced silencing of MHC-I could also be due to YAP-NOTCH-REST axis activation, at least in non-NE cell clusters. However, we found that *Ezh2* is mainly expressed in NE cells but not non-NE cells, where NE markers are suppressed with HES1/REST activation (**Fig. R39B**). Therefore, it is implausible that the REST-EZH2 axis is engaged in suppressing NE gene expression.

New results were added (**Supplementary Fig. S4**), and further discussion was made (highlighted).

C8. “Discussion should include the recent papers which also describe the single cell features of SCLC (Leslie Duplaquet et al, *Nature Cell Biology* 2023 and Abbie S. Ireland et al, *Cancer Cell* 2022). Those papers may be also useful to give a more comprehensive view for what is occurring in the SCLC cells and how the epigenome perturbation may make effects (either beneficial or adverse). Also, possible involvement of *Myc* could be also discussed.”

As kindly recommended, two references were added. *Myc* was discussed (highlighted). Of note, compared to RPR2, CRPR2 tumors showed *Myc* upregulation in non-NE tumor cells (**Supplementary Fig. S4d**), in line with the findings of the Trudy Oliver group’s paper¹⁹ showing non-NE cell transition by MYC.

C9. *“Please show the list of the genes used for the custom 100 probes for the Xenium analysis. I also wonder how the timing of the sacrifice was determined. The time course sampling should be also useful to validate the robustness of the analysis.”*

The list of genes used for the custom 100 probes in the Xenium analysis has been included in **Supplementary Table 9**. The 6-month post-Cre infection time point was selected because RPR2 (Rb1/Trp53/Rbl2-mutant) mice commonly reach endpoints defined by approved animal protocols and tumor progression at this stage. This timing also allows a robust evaluation of the additional impact of Cracd loss on tumor formation. This standard time point has been widely adopted in prior studies using this GEMM¹⁶. While we agree that time-course sampling could further strengthen the analysis, our current study focused on characterizing the tumor microenvironment at a fully developed stage. Accordingly, samples at earlier time points were not collected. It will certainly be followed up in our future animal experiment, where the Cre-infected lungs are collected at multiple time points and analyzed using Xenium.

C10. *“Please make sure that the Xenium data is also made publicly available in GEO. Also, for all the codes for the data process in GitHub, especially for the parts of the relatively new analyses, including the VR analysis, which I found intuitively very much helpful. (I have not visited the indicated URLs for those datasets using the reviewer token.)”*

We thank the reviewer for this suggestion and have updated the Data and Code availability statements accordingly:

Data availability

All datasets generated in this study have been deposited in the Gene Expression Omnibus (GEO) under the following accession numbers (reviewer tokens provided):

- scRNA-seq: GSE218544 (token: efsxisoijvwzhuh)
- CUT&RUN-seq: GSE280263 (token: uzqpeauszlatteb)
- Xenium spatial transcriptomics: GSE299069 (token: qvkjqkquhtynjkv)

Code availability

All custom scripts and pipelines used for data processing and analysis—including the VR analysis—are publicly available at:

https://github.com/jaeilparklab/CRACD_SCLC_scRNAseq

We appreciate very constructive comments.

Reviewer 4

“Seo et al. examined the impact of CRACD inactivation on small cell lung cancer (SCLC) tumor development, plasticity and immune evasion. The Authors examined Cracd/CRACD in SCLC in vitro and in vivo model systems, human primary SCLC samples with experimental and computational approaches and found that loss of Cracd/CRACD disrupted cytoplasmic and nuclear actin organization leading to dysregulation of NOTCH1 signaling and EZH2-mediated suppression of MHC-I pathways. Alterations to these processes in turn resulted in increased plasticity and immune evasion. The Authors’ demonstration of cellular plasticity using single cell and spatial genomics was state-of-the-art and also contributed to their hypothesis. Functional validation of computational results using animal and in vitro models of SCLC was also comprehensive. The study also suggested EZH2 inhibition as a potential strategy to improve response to immune checkpoint blockade (ICB) therapy in SCLC. Overall, this is a very comprehensive and well-executed study with high translational relevance. To strengthen the study further, the authors should consider the following questions/issues:”

Major comments

D1. “Given germline constitutive knockout of Cracd in the qKO model, is it possible that the differences in the tumor immune microenvironment and even in the cancer cells themselves in the tKO vs. qKO models also stemmed from Cracd loss in the stromal, immune and other non-cancer cells in the mouse? Has the impact of Cracd loss in various immune and other stromal compartments such as T cells been documented in vitro and in vivo models in the Cracd KO control mouse?”

Thank you for the insightful comments. We agree that the differences observed between tKO and qKO models could be influenced not only by tumor-intrinsic factors but also by changes in the stromal and/or immune compartments.

Cracd KO in tumor cells vs. non-tumor cells. To address this potential issue from germline KO models, we performed *syngeneic transplantation* of SCLC tumor cells (RPR2 [Cracd WT] vs. CRPR2 [Cracd KO]; derived from GEMMs [C57BL/6]; **Fig. 1a**) into C57BL/6 mice (Cracd WT) (**Fig. 4, Supplementary Fig. S8**). In this experimental setting, only tumor cells carry the Cracd KO. Consistent with the results from GEMMs (**Fig. 3**), tumor development from CRPR2 cells was significant compared to RPR2 (**Fig. 4I, Supplementary Fig. 8d-e**), excluding potential pitfalls in using Cracd germline KO mice. We recently demonstrated the NE cell plasticity of LUAD cells by using conditional Cracd KO¹. In this study, we utilized both Cracd KO GEMMs and somatic engineering by intratracheal instillation of adenoviruses encoding Cas9-sgCracd-Cre and observed similar results, a conversion of LUAD into NE tumor cells.

Impact of Cracd KO on tumor niche. We identified the CRACD gene as a tumor suppressor and established Cracd germline KO mice². In mice, Cracd KO is sufficient to induce intestinal hyperplasia², mucinous cell plasticity of the intestine (unpublished), NE cell plasticity of LUAD¹, and hyperpigmentation of the skin (unpublished). Cracd KO mice are viable without any discernible phenotypes. It should be noted that CRACD, as a capping protein inhibitor, promotes actin polymerization and is generally expressed in epithelial cells. In our recent study¹, we analyzed Cracd WT vs. KO lung tissues for scRNA-seq datasets. We found no outstanding differences in cell clusters and ratios of immune cells, fibroblasts, and endothelial cells in Cracd KO mice compared to WT¹.

Therefore, the impact of Cracd KO on the tumor niche, including immune cells, might be marginal or insufficient to be associated with NE plasticity. We added such information in the Discussion as an alternative interpretation as well as limitations.

D2. “The authors argued that CRACD inactivation, via actin dysregulation, inhibited NOTCH1 signaling, which in turn promoted neuroendocrine (NE) plasticity. Why would this phenotype not occur in the non-NE population with activated NOTCH1 signaling in the qKO tumors? In Figure 7j, the authors claimed that increased non-NE features in qKO tumors may occur via EMT activation, although correlations between the two were only observed. The authors should qualify these claims more or provide additional evidence of the CRACD inactivation-EMT-non-NE axis.”

Thank you for insightful comments. To some extent, our results align with the Julien Sage lab's report, i.e., the YAP-NOTCH-HES1/REST axis inhibits NE gene expression⁴ and the Trudy Oliver lab's finding., MYC-driven non-NE cell plasticity with YAP and NOTCH signaling activation^{4,19}.

YAP-NOTCH-REST axis in inhibiting NE cell plasticity. In addition to the results in this study, our recent paper¹ and a manuscript in revision showed that CRACD LOF consistently inhibits the NOTCH signaling, leading to the activation of the secretory cell lineages, including mucinous or neuroendocrine ones, depending on the context. Yang Ting Shu et al. showed that YAP1-transactivated *NOTCH2* directly activates *HES1* and *REST* to suppress NE genes⁴.

Mechanism. Briefly, CRACD loss-dysregulated actin polymerization activates the HIPPO signaling that senses mechanical stress^{23,24}, which subsequently phosphorylates and inhibits YAP1. Then, the downregulation of YAP1-transactivated *NOTCH1*²⁵ downregulates the NOTCH signaling, followed by reduced expression of NOTCH1 target genes, *HES1* and *REST*. Finally, NE genes suppressed by HES1 and REST are de-repressed to induce NE cell plasticity (**Fig. R41**).

Supporting data. A. While RPR2 tumors (*Cracd* WT) displayed a distinct nuclear pattern of YAP1 protein expression, CRPR2 (*Cracd* KO) tumor cells barely showed the expression of YAP1 (both nuclear and cytosolic) (**Fig. R42**). We also treated RPR2 cells with verteporfin, a YAP1 inhibitor that disrupts the YAP-TEAD complex. Verteporfin reduced the cleaved Notch1 (N1ICD) protein and its downstream target HES1 (**Fig. R43**). These results support that CRACD influences NOTCH signaling via the actin-YAP1 axis. We have updated the discussion and added these data to the revised manuscript (**Fig. 2l-m, Supplementary Fig. S4**).

Figure R41. Illustration of the mechanism of how CRACD inactivation induces NE cell plasticity, tumor cell heterogeneity, and MHC-I suppression.

Figure R42. Immunohistochemistry of SCLC tumors (RPR2 vs. CRPR2) for YAP1. Scale bars: 50 μm.

Figure R43. Immunoblot analysis of RPR2 cells treated with vehicle or verteporfin 0.5 (+) or 1.0 μM (++) 72 hours after treatment, cells were harvested for immunoblotting.

CRACD is a capping protein inhibitor, promoting actin polymerization². Thus, as pointed out, it is highly likely that CRACD LOF reduces actin polymerization (F-actin) (**Fig. 2k**), which inhibits the HIPPO signaling and subsequently activates YAP1-mediated *NOTCHs* expression, leading to the de-repression of NE genes. This may explain how CRACD LOF induces NE cell plasticity of the NE cell clusters. Indeed, compared to RPR2, CRPR2 tumors exhibited an increased number and cell plasticity of non-NE tumor cells (**Fig. 2f-h**). This might be due to the activation of the YAP1-NOTCH2-REST axis, as described in Yang Ting Shu et al^{26,27}. In the initially submitted manuscript, we observed the marked upregulation of *Hes1*, *Dll1*, *Jag1*, *Notch1*, *Notch2*, and *Notch3* in the non-NE cell clusters of CRPR2 compared to RPR2 (**Supplementary Fig. S3d**). Additionally, we observed the upregulation of *Yap1*, *Taz*, *Tead1/4*, and *Rest* in non-NE cells of CRPR2 tumors compared to RPR2 (**Supplementary Fig. 4b, Fig. R44**). Thus, increased non-NE cell clusters in CRPR2 tumors can be well explained by the activation of the YAP-NOTCH-REST axis. Unfortunately, given that our customized geneset of Xenium did not include those genes, we could not combine scRNA-seq with Xenium in the context of the YAP-NOTCH-REST axis.

Figure 3 (from the initially submitted manuscript)

Figure R44. Feature plots of scRNA-seq datasets of RPR2 and CRPR2 tumors. **A.** Upregulation of the Yap1/Taz-REST axis in non-NE cell clusters of CRPR2 tumors. **B.** Upregulation of Ezh2 mainly in NE cell clusters of CRPR2 tumors, compared to RPR2.

Distinct impact of *Cracd* KO on non-NE and NE cell clusters. As insightfully pointed out, it is somewhat difficult to digest why *Cracd* KO affects both non-NE and NE cell clusters, which is quite perplexing in several aspects. Until scRNA-seq-based analyses, such complicated questions could not be addressed. To challenge this, we examined cell lineage trajectories of RPR2 vs. CRPR2 tumors using the scRNA-seq datasets. First, we found that #10 NE cell cluster is the actively proliferating cell cluster in both RPR2 and CRPR2 tumors (**Fig. 1j**). Intriguingly, cell lineage trajectory analyses using multiple packages showed that, unlike RPR2, *Cracd* KO (CRPR2) increases non-NE (cell clusters #1, 4, 2), which become NE cell clusters (**Fig. 1k-n**). This explains why *Cracd* KO SCLC tumor cells show an increase in both non-NE and NE cell clusters. This is also consistent with the MYC-driven non-NE cell plasticity¹⁹. Our experimental data is further supported by analyzing tumor cell heterogeneity. Unlike such a cell lineage, spatial transcriptomics also confirmed the increased cell plasticity and tumor cell heterogeneity in *Cracd* KO SCLC tumors (CRPR2) (**Fig. 3i, j, and 4**). Next, we checked the expression of all components and target genes of the actin-HIPPO, HIPPO, YAP, and NOTCH pathways from the scRNA-seq datasets of RPR2 and CRPR2 tumors. As mentioned above, the YAP-

Figure R45. Illustration of expression of signaling pathways and target genes. The expression of each pathway component or target gene was visualized based on the scRNA-seq datasets of RPR2 and CRPR2 tumors.

NOTCH signaling axis is activated in only the non-NE cells but not NE cells, represented by specific expression of *Yap1*, *Taz*, *Notchs*, *Hes1*, and *Rest* in non-NE cells (**Fig. R44**). While genes encoding proteins involved in actin-regulated HIPPO signaling did not show differences in their expression, *Ajuba* expression is only detected in non-NE cells (**Fig. R45**). Ajuba protein directly binds to and inhibits LATS1/2, activating YAP. Given its role in inhibiting HIPPO signaling, Ajuba's specific expression in non-NE cells might explain why *Cracd* KO leads to two distinct outcomes (HIPPO activation in NE cells and YAP activation in non-NE cells). Nonetheless, we cannot exclude the involvement of other signaling or TFs because the expression level of *Ajuba* was not markedly higher compared to YAP and NOTCH signaling activation.

These new results and discussion were added to the revised manuscript.

D3. *“What correlations did the authors observe upon analysis of mouse tumor samples using the different methods (scRNA-seq, spatial transcriptomics), in terms of (1) SCLC plasticity/lineage identity and (2) the profile of immune cells within the microenvironment? In other words, are the differences in cellular composition observed with scRNA-seq reproduced in the spatial transcriptomic dataset? A deeper and full analysis of these combined datasets may be the focus of a separate paper altogether, but some superficial-level correlations is warranted for additional rigor in the current study.”*

Xenium In Situ before 5K genesets availability. We agree to these critical comments. As pointed out, we initially planned to combine scRNA-seq with Xenium In Situ to complement each other. Our Xenium experiments were performed with the customized 479 genesets before the release of 5K genesets. With the limited number of genes, we were only able to acquire cell cluster-related information, which was used for analyzing tumor cell heterogeneity (Fig. 2f-h).

Visualizing scRNA-seq information on Xenium. Cell lineages. As suggested, we projected cell clusters and cell proliferation information from scRNA-seq onto Xenium data. Consistent with the scRNA-seq results, we located new non-NE cell clusters (purple) generated in CRPR2 tumors at the core of SCLC tumor nodules. These non-NE cells were surrounded by NE cells (Fig. R46A, B), recapitulating NE cell plasticity of CRPR2 tumors. Additionally, CRPR2 tumor cells are more proliferative than RPR2 tumor cells. Moreover, NE cells are relatively more mitotic than non-NE cells (Fig. R46C). **Immune niche.** We also quantified the T cells infiltrated into SCLC tumors (RPR2 vs. CRPR2). In line with the results from scRNA-seq and immunostaining-based immune profiling (Supplementary Fig. S6), T cells were rarely detected in CRPR2 tumors compared to RPR2 tumors (Fig. R46D). These new results were added to the revised manuscript (Fig. 2 f-h, Supplementary Fig. 6n-o).

Figure R46. Spatial localization and characterization of tumor cell subpopulations and T cells in RPR2 and CRPR2 tumors. **A.** UMAP projection of scRNA-seq data from RPR2 and CRPR2 tumors. Cells are colored by unsupervised clusters; NE and non-NE populations are delineated by dashed lines. **B.** Spatial localization of NE and non-NE clusters in representative RPR2 and CRPR2 tumors. Cluster identities were assigned by correlation-based mapping of Xenium transcriptomic data to matched scRNA-seq clusters. **C.** Spatial mapping of cell cycle states projected onto Xenium coordinates. Cell cycle phase (inferred from matched scRNA-seq data) are shown as: G1 (magenta), S (aqua), and G2/M (purple). **D.** Spatial localization of T cells (Cd3d, Cd3e, Cd3g, Cd8a, Cd8b1, Cd4) within RPR2 and CRPR2 tumors, identified by Xenium transcriptomic signatures. **E.** Quantification of T cells density per unit area from Xenium spatial data.

D4. “Any difference in survival between the qKO vs tKO GEMMs? Given the changes in plasticity, did the authors observe differences in metastasis with qKO vs tKO GEMMs?”

As noted in our response to Reviewer 3 (Comment C9), we used the same time point, six months post-Cre infection, for our analysis. At this stage, we compared tumor burden between qKO vs. tKO GEMMs, although survival analysis was not performed. While not known, it is reasonably assumed that the significantly higher tumor burden in the qKO GEMMs compromises lung function faster and shortens their survival compared to tKO GEMMs. At the six-month time points, we have not found any palpable metastasis in the liver and adrenal glands.

Minor comments

D5. “*Italicize in vitro and in vivo*”
 Edited.

D6. “There was no mention of Fig. S2 in the main text.”

It was mentioned in line 112. We edited the manuscript for better legibility.

D7. “Provide in the main text brief descriptions of computational terms and analytical methods that are heavily referenced such as ‘root cell’, ‘scVelo’, ‘Dynamo’, ‘entropy’, and ‘CellChat’.”

All jargons were spread out with references.

D8. “Line 74: Potential mechanisms underlying SCLC plasticity have been explored in a number of studies such as Ireland et al. (2020). The authors should cite additional relevant literature here.”

As kindly suggested, additional references^{18,19,26} were cited.

D9. “Given the central nature of NOTCH1 signaling and MHC-I downregulation to the mechanism of their current study in SCLC, the authors should provide a brief background in the INTRODUCTION.”

Added as shown below.

“Recent studies have identified NOTCH signaling as a critical regulator of NE plasticity in SCLC. The NOTCH signaling inactivation promotes NE differentiation, whereas its activation drives transitions toward non-NE states, contributing to tumor heterogeneity in SCLC¹⁸. A key mechanism of NOTCH1 suppression in SCLC is the upregulation of DLL3 (Delta-like ligand 3), a direct target of ASCL1, which serves as a negative regulator of the NOTCH pathway²⁸.

In parallel, MHC-I downregulation has emerged as a key mechanism of cancer immune evasion, limiting CTL recognition and reducing the efficacy of ICB therapies. Unlike NSCLC, where genetic alterations frequently drive immune escape, SCLC is characterized by profound defects in antigen processing and presentation pathways²⁹, leading to widespread MHC-I suppression.

Recent evidence has implicated LSD1 (KDM1A) as a key epigenetic regulator of MHC-I repression in SCLC³⁰. LSD1 suppresses antigen presentation through histone modifications that silence the transcription of key MHC-I components³¹. Conversely, pharmacological inhibition of LSD1 using GSK-LSD1 or ORY-1001 has been shown to restore MHC-I expression, enhance antigen presentation, and reinvigorate anti-tumor immunity in preclinical SCLC models³⁰.”

D6. “Can the authors comment on the location of mutations in CRACD that are found in primary tumors? Do these mutations, especially those occurring at a high frequency, reside in functional domains of the protein?”

In the initial submission, we have shown the distribution of CRACD mutations identified in primary SCLC tumors (**Supplementary Fig. 1**). Frame-shift mutations at AA 165 are most frequent in SCLC, which is also observed in other cancer types. Thus, the truncated CRACD mutant likely loses its function (loss-of-function), as previously tested². Additional point mutations were also identified, while no specific mutational hot spots other than nonsense mutations were found. In another manuscript (in preparation), we assessed the role of the various CRACD mutants by ectopic expression of CRACD mutants in CRACD-null cells. CRACD is a capping protein inhibitor, promoting actin polymerization (increased F-actin

Figure R47. Assessing the impact of CRACD point mutations on actin polymerization. MKN45 cells carrying a nonsense mutation in the CRACD alleles were transduced with lentiviruses encoding mCherry-tagged CRACD wild-type (WT) or other point mutants. F-actin was visualized by phalloidin staining.

visualized by phalloidin). However, CRACD point mutations failed to do so (**Fig. R47**). There are no specific functional domains in the CRACD protein except for two capping protein-interacting motifs we previously identified². There is a Glu-rich domain (213–333 amino acids). It is noteworthy that, in addition to genetic alterations, the expression of *CRACD* mRNA is also downregulated in SCLC. We are investigating those point mutations in the context of the CRACD protein structure and dysfunction in another study.

D7. “There are two panels in Fig. S1 labeled with ‘D’”
It was revised. Thank you.

D8. “Line 99: please cite the specific dataset on TCGA the authors used to extract transcriptomic bulk RNA-seq data for healthy tissues and SCLC tumors.”

Bulk transcriptomic profiles of healthy lung tissues and SCLC tumors were downloaded from the TCGA repository (TCGA-SCLC cohort) and processed as described.

Reference: Cancer Genome Atlas Research Network. Comprehensive genomic characterization of squamous cell lung cancers. *Nature* 489, 519–525 (2012)³².

D9. “The authors should label the timepoint at which the lungs were collected & compared.”
The detailed information was added in the Methods. Thank you.

D10. “How did the authors select genes for the analysis of the different signaling pathways (NOTCH, Myc, EMT, etc.)? Why are there only 2 genes for the EMT program?”

Pathway analyses. Our apologies for misleading. For unbiased analysis, we do not pick and choose genes for pathway analysis. We performed GSEA, fGSEA, or pathway analysis using the well-accepted and publicly available genesets. In those figures, due to the limited space, we only showed a few representative genes for signaling or pathways. For clarification, we added the pathway analysis instead of several genes (**Supplementary Fig. S4**).

EMT scoring. For clarification, we included the EMT pathway scores and EMT-related genes (**Fig. R48**). The results remained the same.

Figure R48. Feature plots of scRNA-seq datasets of RPR2 and CRPR2 tumors. **A.** Upregulation of the Yap1/Taz-REST axis in non-NE cell clusters of CRPR2 tumors. **B.** Upregulation of Ezh2 mainly in NE cell clusters of CRPR2 tumors, compared to RPR2.

D11. “The figure legend for Figure. 3d refers to violin plots but there were none.”
Legends were edited accordingly (feature plots only).

D12. “Figure 4D: Why were there so few tKO tumors (4) compared to qKO tumors (36) in the spatial transcriptomics study? Could this discrepancy have an impact on the results?”

As observed and described in Figure 2, *Cracd* KO significantly accelerates SCLC tumorigenesis. For Xenium In Situ analysis, we used the same FFPE blocks from Figure 3 showing relatively more tumors in CRPR2 than RPR2 (six months post Cre). This is also consistent with additional finding that immune evasion of CRPR2 tumors by suppressing MHC-I (**Fig. 3I**). Therefore, such a discrepancy in tumor development (RPR2 vs. CRPR2) is well in line with overall results.

D13. “What is the mutation frequency in CRACD in MS1 vs. MS2 tumors?”

Thank you for the insightful comment. Based on the public datasets, CRACD mutation frequency was 12.9% (32/249) in SCLC patient tumors and 15.7% (16/102) in SCLC cell lines (**Supplementary Fig. S1a-b**). Unfortunately, the exome-seq datasets of those patients were not available, therefore, we were unable to directly compare mutation frequency between MS1 and MS2. However, transcriptomic analysis revealed that CRACD expression was consistently lower in MS1 compared to MS2. To better address this comment, we also performed IHC of SCLC TMA for CRACD and found consistent results (**Fig. R28**). It should be noted that CRACD LOF includes genetic alterations, transcriptional downregulation, and post-translational downregulation. We apologize for misleading. We revised the manuscript by adding new results and editing.

Figure R28. Immunohistochemical staining of CRACD, MHC-I, H3K27Me2, and H3K27Me3 in human SCLC tissues. Tumor sections were grouped based on CRACD expression status, and corresponding changes in MHC-I and histone methylation marks were examined. Quantification was performed using H-score analysis. Statistical significance was assessed by Student's t-test. Data are presented as mean \pm SEM; $P < 0.05$ was considered significant.

D14. “Does the mutational status of CRACD correlate with response to immunotherapy in patients with SCLC?”

This is another very important perspective. However, addressing this comment itself would be another study that needs large cohorts. Moreover, SCLC datasets combined with immunotherapy response are not available to the best of our knowledge. Additionally, considering several options for immune checkpoint inhibitors (anti-PD1, PD-L1, and CTLA-4) as well as patient stratification. Respectfully, this could be beyond the scope of the submitted manuscript.

D15. “Line 469. Authors mean “not with primary tumors”

Revised.

D16. “Any functional assays to evaluate cell motility or migration affected by actin dysregulation? Authors should demonstrate the impact of *Cracd* KO on cell motility/migration to confirm its role in actin dysregulation in the context of SCLC cells.”

As suggested, we performed wound healing and transwell cell migration assays using RPR2 and CRPR2 cells. CRPR2 cells showed increased cell migration and invasion (**Fig. R49**). Of note is that the *Cracd* KO colorectal cancer mouse model showed the invasive phenotype in vitro, as examined by pathologists².

Figure R49. Increased motility and invasiveness of CRPR2 cells. **A.** Representative images of wound healing assays in RPR2 and CRPR2 cells at 0 and 2 days. The wound closure area was quantified and plotted on the right. **B.** Transwell migration and invasion assays of RPR2 and CRPR2 cells. Representative images of migrated cells stained with crystal violet.

Again, we appreciate all insightful and constructive comments.

Reviewer 5

“I co-reviewed this manuscript with one of the reviewers who provided the listed reports. This is part of the Nature Communications initiative to facilitate training in peer review and to provide appropriate recognition for Early Career Researchers who co-review manuscripts.”

References

- 1 Kim, B. *et al.* CRACD loss induces neuroendocrine cell plasticity of lung adenocarcinoma. *Cell Reports* **43** (2024).
- 2 Jung, Y.-S. *et al.* Deregulation of CRAD-controlled cytoskeleton initiates mucinous colorectal cancer via β -catenin. *Nature cell biology* **20**, 1303-1314 (2018).
- 3 Seo, J. & Kim, J. Regulation of Hippo signaling by actin remodeling. *BMB reports* **51**, 151 (2018).
- 4 Shue, Y. T. *et al.* A conserved YAP/Notch/REST network controls the neuroendocrine cell fate in the lungs. *Nature Communications* **13**, 2690 (2022).
- 5 Mahmood, S. R. *et al.* β -actin dependent chromatin remodeling mediates compartment level changes in 3D genome architecture. *Nature Communications* **12**, 5240 (2021). <https://doi.org/10.1038/s41467-021-25596-2>
- 6 Gandhi, V., Plunkett, W. & Cortes, J. E. Omacetaxine: a protein translation inhibitor for treatment of chronic myelogenous leukemia. *Clin Cancer Res* **20**, 1735-1740 (2014). <https://doi.org/10.1158/1078-0432.Ccr-13-1283>
- 7 Ku, S. Y. *et al.* Rb1 and Trp53 cooperate to suppress prostate cancer lineage plasticity, metastasis, and antiandrogen resistance. *Science* **355**, 78-83 (2017).
- 8 Nouri, M. *et al.* Therapy-induced developmental reprogramming of prostate cancer cells and acquired therapy resistance. *Oncotarget* **8**, 18949 (2017).
- 9 Farrell, A. S. *et al.* MYC regulates ductal-neuroendocrine lineage plasticity in pancreatic ductal adenocarcinoma associated with poor outcome and chemoresistance. *Nature communications* **8**, 1728 (2017).
- 10 Porazzi, P. *et al.* EZH1/EZH2 inhibition enhances adoptive T cell immunotherapy against multiple cancer models. *Cancer Cell* **43**, 537-551. e537 (2025).
- 11 Zoma, M. *et al.* EZH2-induced lysine K362 methylation enhances TMPRSS2-ERG oncogenic activity in prostate cancer. *Nature Communications* **12**, 4147 (2021).
- 12 Kim, E. *et al.* Phosphorylation of EZH2 activates STAT3 signaling via STAT3 methylation and promotes tumorigenicity of glioblastoma stem-like cells. *Cancer cell* **23**, 839-852 (2013).
- 13 Rudin, C. M. *et al.* Molecular subtypes of small cell lung cancer: a synthesis of human and mouse model data. *Nature Reviews Cancer* **19**, 289-297 (2019).
- 14 Gay, C. M. *et al.* Patterns of transcription factor programs and immune pathway activation define four major subtypes of SCLC with distinct therapeutic vulnerabilities. *Cancer cell* **39**, 346-360. e347 (2021).
- 15 Chan, J. M. *et al.* Signatures of plasticity, metastasis, and immunosuppression in an atlas of human small cell lung cancer. *Cancer cell* **39**, 1479-1496. e1418 (2021).
- 16 Schaffer, B. E. *et al.* Loss of p130 accelerates tumor development in a mouse model for human small-cell lung carcinoma. *Cancer research* **70**, 3877-3883 (2010).
- 17 Gazdar, A. F. *et al.* The comparative pathology of genetically engineered mouse models for neuroendocrine carcinomas of the lung. *Journal of thoracic oncology* **10**, 553-564 (2015).
- 18 Lim, J. S. *et al.* Intratumoural heterogeneity generated by Notch signalling promotes small-cell lung cancer. *Nature* **545**, 360-364 (2017).
- 19 Ireland, A. S. *et al.* MYC drives temporal evolution of small cell lung cancer subtypes by reprogramming neuroendocrine fate. *Cancer cell* **38**, 60-78. e12 (2020).
- 20 Park, K.-S. *et al.* Characterization of the cell of origin for small cell lung cancer. *Cell cycle* **10**, 2806-2815 (2011).
- 21 Branchfield, K. *et al.* Pulmonary neuroendocrine cells function as airway sensors to control lung immune response. *Science* **351**, 707-710 (2016).
- 22 Dietrich, N. *et al.* REST-mediated recruitment of Polycomb repressor complexes in mammalian cells. *PLoS genetics* **8**, e1002494 (2012).
- 23 Reddy, P., Deguchi, M., Cheng, Y. & Hsueh, A. J. Actin cytoskeleton regulates Hippo signaling. *PloS one* **8**, e73763 (2013).
- 24 Matsui, Y. & Lai, Z.-C. Mutual regulation between Hippo signaling and actin cytoskeleton. *Protein & cell* **4**, 904-910 (2013).
- 25 Totaro, A. *et al.* YAP/TAZ link cell mechanics to Notch signalling to control epidermal stem cell fate. *Nature communications* **8**, 15206 (2017).
- 26 Wu, Q. *et al.* YAP drives fate conversion and chemoresistance of small cell lung cancer. *Science Advances* **7**, eabg1850 (2021).

- 27 Totaro, A., Panciera, T. & Piccolo, S. YAP/TAZ upstream signals and downstream responses. *Nature cell biology* **20**, 888-899 (2018).
- 28 Kim, J. W., Ko, J. H. & Sage, J. DLL3 regulates Notch signaling in small cell lung cancer. *iScience* **25** (2022).
- 29 Roper, N. *et al.* Notch signaling and efficacy of PD-1/PD-L1 blockade in relapsed small cell lung cancer. *Nature communications* **12**, 3880 (2021).
- 30 Nguyen, E. M. *et al.* Targeting LSD1 rescues MHC class I antigen presentation and overcomes PD-L1 blockade resistance in small cell lung cancer. *Journal of thoracic oncology: official publication of the International Association for the Study of Lung Cancer* **17**, 1014 (2022).
- 31 Hiatt, J. B. *et al.* Inhibition of LSD1 with bome demstat sensitizes small cell lung cancer to immune checkpoint blockade and T-cell killing. *Clinical Cancer Research* **28**, 4551-4564 (2022).
- 32 Hammerman, P. S. *et al.* Comprehensive genomic characterization of squamous cell lung cancers. *Nature* **489**, 519-525 (2012). <https://doi.org:10.1038/nature11404>